# Insights into the aging of biomass burning aerosol from satellite observations and 3D atmospheric modeling: Evolution of the aerosol optical properties in Siberian wildfire plumes

Igor B. Konovalov[1], Nikolai A. Golovushkin[1], Matthias Beekmann[2], and Meinrat O. Andreae[3,4,5]

[1]Institute of Applied Physics, Russian Academy of Sciences, Nizhniy Novgorod, Russia
[2]LISA/IPSL, Laboratoire Interuniversitaire des Systèmes Atmosphériques, UMR CNRS 7583, Université Paris Est Créteil (UPEC) et Université Paris Diderot (UPD), France
[3]Max Planck Institute for Chemistry, Mainz, Germany
[4]Scripps Institution of Oceanography, University of California San Diego, La Jolla, CA 92093, USA
[5]Department of Geology and Geophysics, King Saud University, Riyadh, Saudi Arabia

*Correspondence to*: Igor B. Konovalov (konov@ipfran.ru)

**Abstract.** Long-range transport of biomass burning (BB) aerosol from regions affected by wildfires is known to have a significant impact on the radiative balance and air quality in receptor regions. However, the changes that occur in the optical properties of BB aerosol during the long-range transport events are insufficiently understood, limiting the adequacy of representations of the aerosol processes in chemistry transport and climate models. Here we introduce a framework to infer and interpret changes in the optical properties of BB aerosol from satellite observations of multiple BB plumes. Our framework includes (1) a procedure for analysis of available satellite retrievals of the absorption and extinction aerosol optical depths (AAOD and AOD) and single scattering albedo (SSA) as a function of the BB aerosol photochemical age, and (2) a representation of the AAOD and AOD evolution with a chemistry transport model (CTM) involving a simplified volatility basis set (VBS) scheme with a few adjustable parameters. We apply this framework to analyze a large-scale outflow of BB smoke plumes from Siberia toward Europe that occurred in July 2016. We use AAOD and SSA data derived from OMI (Ozone Monitoring Instrument) satellite measurements in the near-UV range along with 550 nm AOD and carbon monoxide (CO) columns retrieved from MODIS (Moderate Resolution Imaging Spectroradiometer) and IASI (Infrared Atmospheric Sounding Interferometer) satellite observations, respectively, to infer changes in the optical properties of Siberian BB aerosol due to its atmospheric aging and to get insights into the processes underlying these changes. Using the satellite data in combination with simulated data from the CHIMERE CTM, we evaluate the enhancement ratios (EnR) that allow isolating AAOD and AOD changes due to oxidation and gas-particle partitioning processes from those due to other processes, including transport, deposition, and wet scavenging. The behavior of EnRs for AAOD and AOD is then characterized using nonlinear trend analysis. It is found that the EnR for AOD strongly increases (by about a factor of 2) during the first 20-30 hours of the analyzed evolution period, whereas the EnR for AAOD does not exhibit a statistically significant increase during this period. The increase in AOD is accompanied by a statistically significant enhancement of SSA. Further BB aerosol aging (up to several days) is associated with a strong decrease of EnRs for both AAOD and AOD. Our VBS simulations constrained by

the observations are found to be more consistent with satellite observations of strongly aged BB plumes than "tracer" simulations in which atmospheric transformations of BB organic aerosol were disregarded. The simulation results indicate that the upward trends in EnR for AOD and in SSA are mainly due to atmospheric processing of secondary organic aerosol (SOA), leading to an increase in the mass scattering efficiency of BB aerosol. Evaporation and chemical fragmentation of the SOA

species, part of which is assumed to be absorptive (to contain brown carbon), are identified as a likely reason for the subsequent decrease of the EnR for both AAOD and AOD. Hence, our analysis reveals that the long-range transport of smoke plumes from Siberian fires is associated with major changes in BB aerosol optical properties and chemical composition. Overall, this study demonstrates the feasibility of using available satellite observations for evaluating and improving representations in atmospheric models of the BB aerosol aging processes in different regions of the world at much larger temporal

scales than those typically addressed in aerosol chamber experiments.

## 1 Introduction

Open biomass burning – including wildfires, controlled fires, and agricultural burns – is an important source of aerosol particles that are a key agent in the processes controlling regional and global climate (e.g., Bond et al., 2013; Jacobson, 2014; Sand et al., 2015). Climate effects of biomass burning (BB) aerosol are associated, in particular, with the scattering of the

solar radiation by organic matter, which is typically a dominant component of BB aerosol (Reid et al., 2015a), as well as with absorption of solar radiation by black carbon (BC) and brown carbon (BrC), which are also ubiquitous components of BB aerosol (Andreae and Gelencsér, 2006), both within the atmosphere and on snow and ice surfaces (Gustafson and Ramanathan, 2016; Jiang et al., 2016; Wang et al., 2016). Acting as cloud condensation nuclei (CCN), BB aerosol particles can affect the formation and radiative characteristics of clouds (e.g., Hobbs et al., 1969; Petters et al., 2009; Lu et al., 2018).

Apart from being an important agent in the climate system, BB aerosol is a major contributor to air pollution episodes that have been caused by wildfires worldwide (e.g., Konovalov et al., 2011; Keywood et al., 2013; Gupta et al., 2018) and represent a major human health problem (Lelieveld et al., 2020). These facts highlight the importance of having adequate representations of the physical and chemical properties of BB aerosol in models of various complexities for climate and air pollution studies. However, developing such representations is an extremely challenging task, given the high diversity of BB

aerosol composition and optical properties (e.g., Reid et al., 2005a,b; Tsigaridis and Kanakidou, 2018), and the dynamic, nonlinear nature of the atmospheric evolution of its organic component (e.g., Bian et al., 2017; Konovalov et al., 2019).

The composition of primary BB smoke is strongly dependent on fuel type and highly variable burning conditions (McMeeking et al., 2009; Levin et al., 2010; Liu et al., 2014; Laskin et al., 2015). For example, smoldering fires have been reported to favor higher emissions of organic aerosol (OA) and semi-volatile organic gases, but lower BC emissions, compared to flam-

ing conditions (Jen et al., 2019). In turn, the BC-to-OA ratio has been identified as a key factor controlling BrC contribution to light absorption by BB aerosol (Saleh et al., 2014; Lu et al., 2015; Pokhrel et al., 2017; McClure et al., 2020).

Furthermore, results of laboratory and field studies show a very diverse picture of atmospheric transformations of BB aerosol, suggesting that oxidation of volatile and semi-volatile organic gases or heterogeneous oxidation of particles can lead, in different cases, to either enhancements or losses of organic matter by BB aerosol (e.g., Yokelson et al. 2009; Cubison et al., 2011; Hennigan et al., 2011; Akagi et al., 2012; Jolleys et al., 2015; Konovalov et al., 2015; Tiitta et al., 2016; Ciarelli et al.,

2017; Vakkari et al., 2018). Besides, some observational studies reported that atmospheric aging of BB aerosol did not result in changes of mass concentration (corrected for dilution) of OA (e.g., May et al.; 2015; Sakamoto et al. 2015; Zhou et al., 2017). This diversity, which is presently not well understood and is not reproduced in chemistry transport and climate models, can, at least partly, be associated with differences in fuel types, burning conditions, initial parameters of BB smoke plumes, and the dilution rate (Hennigan et al., 2011; Ahern et al., 2019; Lim et al., 2019; Hodshire et al. 2019; Konovalov et

al., 2019).

There is still no consensus on the patterns of the changes of absorptive properties associated with brownness of organics in BB aerosol in the real atmosphere, although findings from many laboratory and field studies indicate that these changes can be very significant (e.g., Kumar et al., 2018; Browne et al., 2019; Fan et al., 2019; Li et al., 2019; Sumlin et al., 2019). Based on both a BB aging experiment with an outdoor smog chamber and an analysis of field observations, Zhong and Jang (2014)

reported an initial increase in light absorption by BrC under natural sunlight, followed by a decrease. The same BrC aging effects, that is, an initial photo-enhancement of BrC absorption and eventual bleaching of OA under the UV irradiation, have also been found in more recent laboratory experiments with water-soluble organic extracts from BB aerosol (Wong et al., 2017; 2019). However, only rapid bleaching of BrC (on the time scale of 9 to 15 hours) has been detected in age-resolved aircraft observations of BB aerosol (Forrister et al., 2015). Furthermore, while these aircraft observations suggest that BrC

almost fully disappears from BB aerosol after about 40 hours of atmospheric aging, an analysis of the AErosol RObotic NETwork (AERONET) data by Wang et al. (2016) indicates that a sizeable fraction of BrC "survives" during a much longer atmospheric exposition. Laboratory data (Di Lorenzo et al., 2017; Wong et al., 2017; 2019; Fleming et al., 2020) indicate that this fraction is likely formed by high-molecular-weight chromophores (while low-molecular-weight chromophores are more rapidly destroyed), suggesting that the atmospheric fate of the BrC fraction of BB aerosol is dependent on fuel type and

burning conditions. Finally, it should be noted that the atmospheric relevance of various processes, which, according to laboratory studies, can affect formation and bleaching of chromophores in BB aerosol, is so far not sufficiently clear (Fleming et al., 2020); this fact constitutes an additional obstacle for developing adequate modeling representations of BB aerosol aging processes.

The discussion above indicates that the sparse and often contradictory results of field and laboratory studies available so far

can hardly provide consistent observational constraints for representations of the effects of atmospheric aging of BB aerosol in chemistry-transport and climate models. The general objective of the present study is to find a way to infer statistically reliable information on the impact of aging processes on the optical properties of BB aerosol from available satellite measurements. To this end, we developed an analytical framework enabling the synergetic combination of satellite and model

data on the BB aerosol optical properties and composition. Unlike the inevitably sparse field observations that are available only for a tiny fraction of wildfires occurring in the world each year, satellite instruments provide abundant observations of BB plumes associated with virtually any major vegetation fire worldwide. Accordingly, satellite observations of BB plumes can enable statistical characterization of the BB aerosol aging effect in a given region, encompassing a wide range of BB

aerosol sources and burning conditions. A similar concept was initially exploited in our previous study (Konovalov et al., 2017a), focusing on the analysis of the MODIS (Moderate Resolution Imaging Spectroradiometer) observations of extinction aerosol optical depth (AOD) (Levy et al., 2013) in Siberia. We found, in particular, that after being corrected for dilution, AOD increases almost two-fold in BB plumes aged between 15 and 20 hours, compared to fresher plumes, indicating strong secondary organic aerosol (SOA) formation. This study substantially extends the scope of the previous one by analyzing

satellite observations of both absorption and extinction characteristics of BB aerosol. Specifically, along with the MODIS AOD observations, we analyze near-UV retrievals of the absorption aerosol optical depth (AAOD) and single scattering albedo (SSA) from the OMI (Ozone Monitoring Instrument) measurements (Torres et al., 2007; 2013). Taking into account that the OMI observations of AAOD at the 388 nm wavelength are sensitive to the BrC content in the BB aerosol (Jethva et al., 2011; Mok et al., 2016), these observations can be expected to provide useful insights into the changes of the absorptive

properties of the organic fraction of BB aerosol. At the same time, the MODIS AOD observations are expected to bring light to a possible role of SOA formation (Saleh et al., 2013) in these changes. Note that the OMI AAOD and MODIS AOD observations were earlier found to provide consistent information on BB aerosol optical properties, enabling their use as observational constraints on BB emissions of BC and OC (Konovalov et al., 2018). The OMI AAOD retrievals were also found to provide useful constraints on anthropogenic BC emissions over southeastern Asia (Zhang et al., 2015). Similar to Konovalov

et al. (2017a), we additionally use CO column amounts retrieved from IASI (Infrared Atmospheric Sounding Interferometer) satellite observations (Clerbaux et al., 2009) to account for the effects of dilution.

To analyze and interpret the satellite observations, we use the CHIMERE chemistry transport model (Mailler et al., 2017) in combination with an external module, OPTSIM (Stromatas et al., 2012), enabling Mie calculations of the light absorption and extinction for both core-shell and homogeneous morphologies of particles. Typically, BrC has been introduced in previ-

ous modeling studies as an absorbing fraction of otherwise non-absorbing OA (e.g., Park et al., 2010; Saleh et al., 2015; Jo et al., 2016), with pre-defined absorptive properties that are not affected by atmospheric aging. More recently, Wang et al. (2018), Brown et al. (2018), and Zhang et al. (2020) assigned empirical estimates of absorption properties for all OA in their global modeling studies, and also assumed that the absorption of OA is decreasing with a constant rate as a function of the BB aerosol photochemical age, irrespective of the simulated evolution of the chemical composition of the particles. In this

study, we largely follow the latter approach but take a step further by considering evolution of BrC consistently with simulated changes of both primary organic aerosol (POA) and SOA within the volatility basis set (VBS) framework (Robinson et al., 2007). Several parameters of our model are adjusted to meet the observational constraints. Nonetheless, because our representation of BB aerosol aging is highly simplified and associated with large uncertainties, we use our simulations main-

ly to get insights into qualitative patterns of the BB aerosol transformations, rather than for quantitative characterization of any processes.

Aerosol particles emitted from vegetation fires in Siberia can be transported in the atmosphere over thousands of kilometers and significantly affect atmospheric composition in receptor regions, such as, e.g., East Asia (Agarwal et al., 2010; Matsui et al, 2013, Ikeda et al., 2015; Yasunari et al., 2018 ), the Western USA (Laing et al., 2016), and Alaska (Warneke et al., 2009). Siberian fires have been estimated to contribute, on average, up to 40 % of total BC deposited annually in the Arctic (Evangelio et al., 2016), where BB aerosol can significantly affect the climate processes (Sand et al., 2015) associated with an observed rapid increase of the annual surface temperature in this region (Bekryaev et al., 2010). As the role of fires in boreal regions is projected to increase in a warmer climate (Oris et al., 2014), there is a need for better quantitative understanding of the climate impact of BB aerosol emitted in these regions, which can be achieved, in particular, through the use of stronger observational constraints to represent of BB aerosol optical properties and their evolution in chemistry transport and climate models. A way to obtain and exploit such constraints is proposed in this paper.

In this study, we analyze aging effects on the evolution of BB aerosol from Siberian fires by considering a relatively short period – from 15 to 31 July 2016. This analysis period is chosen to include a spectacular event of an anomalous outflow of optically dense BB plumes from Siberia to the European part of Russia. Different aspects of this event have already been addressed in the scientific literature (e.g., Sitnov et al., 2017; 2020; Gorchakov et al., 2017). It has been shown, in particular, that this event was associated with the formation of a so-called dipole structure of the atmospheric circulation, with the westward transport of air masses being a result of anticyclonic and cyclonic circulations to the north and the south of the air transport trajectory, respectively. The outflow of the BB plumes took place in a relatively dry and cloudless atmosphere, providing favorable conditions for observing the evolution of BB smoke plumes from space. Importantly, the geographical isolation of aged and young plumes during this event allows us to minimize the effects of possible uncertainties associated with the estimation of photochemical age.

To the best of our knowledge, this is the first study attempting to constrain simulations of the aging behavior of BB aerosol with satellite observations of both absorption and extinction AODs. Using the constrained simulations, we attempt to get insights into the key physical and chemical processes driving changes of the optical characteristics of BB aerosol from Siberian wildfires during its long-range transport. By examining the changes in the absorption properties of BB aerosol through the joint analysis of the corresponding satellite retrievals and Mie calculations, this study significantly extends our previous analysis of the evolution of BB aerosols in Siberia (Konovalov et al., 2017a), which had been focused on the evolution of only the extinction AOD. Although this study addresses a relatively short period and a specific region, we believe that the analytical framework developed here, as well as insights into the effects of atmospheric aging of BB aerosol during a period spanning its typical lifetime, will be beneficial for other studies involving modeling of aerosol processes and their climate impacts, especially in Northern Eurasia and in the Arctic.

## 2 Method and data

### 2.1 Analysis method

We analyze the atmospheric evolution of BB aerosol by considering the enhancement ratio (EnR), which is defined as the ratio of the enhancement of actual AAOD or AOD due to fire emissions to the corresponding enhancement of an aerosol tracer that represents the atmospheric evolution of a hypothetical BB aerosol in which the organic fraction is composed of inert, non-volatile and non-hygroscopic matter. Under this definition, the enhancement ratio for AAOD (EnR$_{abs}$) is evaluated as follows:

$$EnR_{abs} = \eta_0 \frac{\tau_{abs} - \tau_{abs}^{bgr}}{\tau_{abs}^{t}}, \tag{1}$$

where $\tau_{abs}$ is the AAOD retrieval, $\tau_{abs}^{bgr}$ is the estimate of the background part of AAOD, $\tau_{abs}^{t}$ is the estimate of AAOD associated with the inert aerosol tracer, and $\eta_0$ is a constant normalization factor defined below. The enhancement ratio for extinction AOD (EnR$_{ext}$) is evaluated similar to EnR$_{abs}$, by substituting the observed and estimated AOD values ($\tau_{ext}$, $\tau_{ext}^{bgr}$, and $\tau_{ext}^{t}$) into Eq. (1) instead of $\tau_{abs}$, $\tau_{abs}^{bgr}$, and $\tau_{abs}^{t}$. The objective of our analysis involving these EnRs is to identify the differences between changes in AAOD or AOD due to aging of BB aerosol in the real BB plumes (including the changes that occur due to oxidation and condensation/evaporation and can also be indirectly induced by the dry and wet deposition of organic gases and particles) and those in a hypothetical simulation in which the organic fraction of BB aerosol is composed of only non-volatile, inert, and hydrophobic material and SOA formation processes are negligible. In our analysis involving EnRs, we largely follow Konovalov et al. (2017a) and Konovalov et al. (2019), but it should also be noted that the meaning of EnR defined by Eq. (1) is very similar to that of the normalized excess mixing ratio (NEMR), which was employed in previous studies (e.g., Yokelson et al., 2009; Vakkari et al., 2018; Junghenn Noyes et al., 2020) to analyze the evolution of mass concentrations (or mixing ratios) of aerosol species. In the framework of this study, Eq. (1) is also used to analyze the simulated evolution of BB aerosol. To this end, AAOD (or AOD) retrievals are replaced in Eq. (1) by the corresponding simulated values.

The BB aerosol tracers ($\tau_{abs}^{t}$ and $\tau_{ext}^{t}$) are evaluated using AAOD and AOD values obtained from simulations with a chemistry transport model (CTM) as explained in Sect. 2.3. An advantage of the BB aerosol "optical" tracers in comparison with more traditional tracers based on CO concentration is that they are affected not only by transport and eddy diffusion but also by dry deposition, wet scavenging, and coagulation, thereby representing BB aerosol in the aforementioned hypothetical simulation. Note that coagulation is not considered as one of the BB aerosol aging processes addressed in this study.

However, the BB aerosol simulated tracers can be affected by model transport errors that can lead to biases in EnRs for both fresh and aged BB plumes observed from satellites (Konovalov et al., 2017a). To address this potential issue, we employ a simple correction procedure involving satellite observations of CO columns. Specifically, we introduce a correction factor, $f_c$, defined as follows:

$$f_c = ([CO]^{obs} - [CO]^{bgr}) \, ([CO]^{mod})^{-1}, \tag{2}$$

where $[CO]^{obs}$ and $[CO]^{mod}$ are the observed total CO column amounts and the corresponding modeled contribution of fire emissions to the CO columns, respectively, and $[CO]^{bgr}$ is the background value of $[CO]$ in the absence of fires. This factor is applied to the simulated tracers. In particular, the simulated values of $\tau^t_{abs}$ are corrected as follows:

$$\tau^t_{abs}[corrected] = \tau^t_{abs}[simulated] \times f_c \tag{3}$$

This correction implies that the simulated CO column amounts in a BB plume are affected by the same multiplicative transport errors as the BB aerosol components determining AAOD and AOD. Note that a similar procedure was used in Konovalov et al. (2017a), where advantages of using the observed and simulated CO columns in the estimation of EnR for AOD are discussed in detail; similar arguments apply for our estimations of EnRs for AAOD and AOD in this study. Estima-

tion of the background values of $\tau_{abs}$, $\tau_{ext}$, and $[CO]$ in this study is described in Sect. 2.3.

The normalization factor $\eta_0$ does not affect the results of our analysis and is introduced mainly for illustrative purposes. Here, we define it such that the average of all values of $EnR_{abs}$ (or $EnR_{ext}$) over the ensemble of the data considered is equal to one:

$$\eta_0 = \langle \frac{\tau_{abs} - \tau^{bgr}_{abs}}{\tau^t_{abs}} \rangle^{-1}, \tag{4}$$

where the angular brackets denote averaging over the data ensemble.

As part of our analysis, we infer trends of EnRs for AAOD and AOD in the process of BB aerosol aging from the ensemble of available estimates of $EnR_{abs}$ and $EnR_{ext}$. To characterize the BB aerosol age, we evaluate the time, $t_e$, of exposure of a given BB plume to solar irradiation. In other words, $t_e$ is the integral time of the transport of a given BB plume since emission, during daytime hours. Unlike the BB aerosol photochemical age evaluated in Konovalov et al. (2017a), the solar irradi-

ation exposure time – which has been previously employed by Forrister et al. (2015) to evaluate the photo-degradation of BrC in observed BB plumes – is not affected by potential major uncertainties in the simulated OH concentration. At the same time, $t_e$ also quantifies the exposure of BB aerosol to the photochemical processing through oxidation by OH under the assumption that the OH concentration inside the plumes is constant during daytime and zero during nighttime. We believe that this is a reasonable simplifying assumption in our case, where observations of OH are unavailable and the accuracy of

the simulated OH concentration in the BB plumes is unknown. Accordingly, for brevity, we refer below to $t_e$ as the photochemical age of BB aerosol. Using a photochemical age defined in this way as a predictor for the atmospheric evolution of BB aerosol in an analysis such as ours has the potential drawback that it can result in losing some observational information about the oxidation processes controlled by OH and that it disregards possible effects of the nighttime oxidation processes (specifically involving $NO_3$ and $O_3$ as oxidation agents). A concrete method which is used to evaluate $t_e$ in this study is de-

tailed in Sect. 2.3.

The trends in EnR$_{abs}$ and EnR$_{ext}$ (that is, the corresponding dependences on $t_e$) are fitted with a nonlinear function, $y\,(t_e)$, which is constructed as a sum of sigmoids:

$$y(t_e) = \sum_{k=1}^{N} \frac{w_{k1}}{(1+\exp(w_{k2}t_e+w_{k3}))} + w_0,$$ (5)

where $w_{k1}$, $w_{k2}$, $w_{k3}$, and $w_0$ are the weight coefficients, and $N$ is the number of sigmoids. In essence, the function $y(t_e)$ repre-
sents a neural network (of the perceptron type), which is known as a universal approximator (Hornik et al., 1989). The opti-
mal values of the weight coefficients are found by minimizing the root mean square difference between the individual values
of EnR$_{abs}$ or EnR$_{ext}$ and the corresponding values of $y\,(t_e)$. Following Konovalov et al. (2010), the optimization of the weight
coefficients is achieved using the Nelder-Mead simplex algorithm (Press et al., 1992). Although the number of neurons ($N$)
used for the trend approximation can also be optimized in a regular (albeit more complicated) way (Konovalov et al., 2010),
here we simply tried the fits with $N=1$, $N=2$, and $N=3$. We found that while the fits with $N=2$ manifested statistically signif-
icant features that are missing in the approximations with $N=1$, the use of three neurons instead of two did not reveal any
new statistically significant features of the analyzed trends. Therefore, only results obtained only with $N=2$ are reported in
this paper. The nonlinear fit function given by Eq. (5) is also applied to the SSA retrievals described in Sect. 2.2 as well as to
model data described in Sect. 2.3. Similar to Konovalov et al. (2017a), the confidence intervals for the approximations $y(t_e)$
were evaluated with the bootstrapping method (Efron and Tibshirani, 1993) involving a random selection of sample esti-
mates. To ensure the reliable evaluation of the confidence intervals in terms of the 95[th] percentile, the analysis involved 300
random samples. Note that the method employed in this study to approximate the tendencies in the optical properties of BB
aerosol is more general and objective than the one used in our previous analysis of the AOD observations (Konovalov et al.,
2017a). The main technical differences between the analysis procedures used in Konovalov et al. (2017a) and this study are
outlined in Supplementary Material, Sect. S1.

## 2.2 Satellite data

Our analysis described in the previous section makes use of aerosol optical properties and total CO column amounts, which
are retrieved from satellite observations performed by the three satellite instruments: OMI (Levelt et al., 2006), MODIS
(Salomonson et al., 1989), and IASI (Clerbaux et al., 2009). Various combinations of these satellite retrievals were used in
our previous studies of the atmospheric impacts of Siberian fires (Konovalov et al., 2014; 2017a; 2018) and are only briefly
described below. Pre-processing and harmonization of the different satellite data sets are explained in Sect. 2.5.

Specifically, we used the AAOD and SSA retrievals for the 388 nm wavelength, which are available as part of the Level-2
OMAERUV (v. 1.8.9.1) data product (Torres et al., 2007, 2013) derived by NASA from the OMI observations onboard the
EOS Aura satellite. The OMAERUV algorithm exploits the observed departure of the spectral dependence of the near-UV
upwelling radiation at the top of the atmosphere from that of a hypothetical pure molecular atmosphere and derives AAOD,
AOD, and SSA following a look-up table approach with assumed aerosol models, aerosol layer height and surface albedo.
Importantly, the OMAERUV data product used in this study accounts for the wavelength-dependent aerosol absorption asso-

ciated with BrC (Jethva et al. 2011; Torres et al., 2013). While the OMAERUV algorithm identifies one of the three assumed aerosol types (biomass burning, desert dust, and urban/industrial), we consider the AAOD and SSA data only for the first aerosol type. The data are retrieved assuming a set of five different vertical locations of the aerosol center mass: at the surface and 1.5, 3.0, 6.0, and 10 km above the surface. We also used the "final" AAOD OMAERUV product, which is derived using the OMI-CALIOP (Cloud-Aerosol Lidar with Orthogonal Polarization) joint dataset (Torres et al., 2013) for the monthly climatology of the aerosol layer height. An important feature of the OMAERUV data product is that the AAOD data are less affected by sub-pixel cloud contamination than the AOD and SSA data, due to partial cancellation of errors in the AOD and SSA retrievals. Accordingly, the quality assured AOD and SSA data (which are associated with the quality flag "0") are much less abundant than the quality assured AAOD data (which are associated with the quality flag "0" and "1").

Both AOD and SSA data were found to be in good agreement with corresponding AERONET data worldwide (Ahn et al., 2014; Jethva et al., 2014), indicating that the AAOD data associated with the quality flag "0" are also sufficiently accurate. Besides, in our previous study (Konovalov et al., 2018), CTM simulations based on BB black carbon (BC) and organic carbon (OC) emissions in Siberia, which were inferred from the OMI AAOD data associated with the quality flag of both 0 and 1 and from the MODIS AOD observations, were found to be consistent with AERONET, aircraft, and in-situ aerosol measurements. Furthermore, the relationship between AAOD and AOD values retrieved from, respectively, the OMI and MODIS satellite observations was found to be virtually the same as that between AAOD and AOD values derived from the AERONET measurements in Siberia, confirming that sub-pixel cloud contamination, which can affect the less reliable retrievals (with the quality flag "1"), is not likely to result in serious biases in the OMI AAOD retrievals. Therefore, the previous results provide evidence for the reliability of the OMAERUV AAOD data corresponding to both values of the quality flag.

We also used AOD at 550 nm derived from the MODIS observations onboard the Aqua and Terra satellites (Levy et al., 2013). The AOD data are obtained as the merged "dark target" and "deep blue" AOD retrievals from the Collection 6.1 MYD04 and MOD04 Level-2 data products (Levy et al., 2015). Validation studies (e.g., Levy et al., 2010) indicated the high quality of the MODIS AOD data, with the multiplicative and additive errors of MODIS-retrieved AOD of ~15% and 0.05 (or less), respectively.

In this study, the OMI AAOD and MODIS AOD retrievals are analyzed independently as described in Sect. 2.1. Besides, a similar analysis was performed for the SSA estimates obtained by combining the AAOD and AOD data as follows:

$$\omega_0 = \frac{\tau_{ext}^{inferred} - \tau_{abs}}{\tau_{ext}^{inferred}}, \tag{6}$$

where $\omega_0$ is an estimate of SSA at 388 nm, $\tau_{abs}$ is AAOD retrieved from the OMI measurements, and $\tau_{ext}^{inferred}$ is AOD at 388 nm inferred from the MODIS observations at 550 nm using a power-law approximation. The extinction Ångström exponent necessary for this calculation was estimated in two ways. First, it was evaluated as the average ratio of the logarithms of the 388-nm AOD and 550-nm AOD retrievals from the matching OMI and MODIS satellite observations, respectively. Since

the OMI AOD data correspond to the quality flag "0" and are therefore very sparse, the Ångström exponent could be estimated only as a constant number being independent of the BB aerosol age. The different AOD retrievals were matched in space and time as described below in Sect. 2.5. Second, the Ångström exponent was calculated using the simulated AOD values (see Sect. 2.3) matching both the AAOD and AOD observations. The advantage of the second approach is that it allowed us to take into account the dependence of the Ångström exponent on the BB aerosol photochemical age, but an obvious drawback of the model estimates is that they can be affected by various model errors.

The total CO columns used in our analysis were retrieved using the FORLI (Fast Optimal Retrievals on Layers for IASI) algorithm (Hurtmans et al., 2012) from the IASI observations onboard the METOP-A satellite (Clerbaux et al., 2009) and are provided by LATMOS/CNRS and ULB (ESPRI, 2020). Following Konovalov et al. (2014; 2016; 2017a), the IASI CO data were pre-selected based on the degree of freedom of the signal (DOFS), which is considered as an indicator of the IASI sensitivity to CO in the boundary layer (George et al., 2009): we only considered the CO observations with DOFS exceeding 1.7.

In addition to the satellite observations of aerosol and CO, we use MODIS retrievals of the fire radiative power (FRP), which are available as the Collection 6 Level-2 MYD14/MOD14 data products (Giglio and Justice, 2015a, b). Following Konovalov et al. (2018), these FRP data were used to evaluate fire emissions, as briefly explained in the next section.

**2.3 Chemistry transport model simulations: basic configuration and fire emissions**

The chemistry transport model (CTM) is a key component of our analytical framework, serving three main objectives. First, we use the model to evaluate the BB aerosol tracers and photochemical age. Second, it is used to characterize the background conditions for aerosol and CO. Third, simulations of BB aerosol evolution help interpreting the results of the satellite data analysis.

The simulations are performed with the CHIMERE CTM, version 2017 (Mailler et al., 2017). The same version of this model was successfully used in our previous analysis of satellite observations of BB aerosol (Konovalov et al., 2018). Previous versions of the CHIMERE model proved to be useful in other studies of BB aerosol and fire emissions (e.g., Hodzic et al., 2010; Konovalov et al., 2012; Péré et al., 2014; Majdi et al., 2019). In this study, the configuration of the model was almost the same as that in Konovalov et al. (2018), although there are some differences mainly concerning the representations of the organic fraction and optical properties of BB aerosol. These model representations are described in Sect. 2.4. The basic features of the model configuration used in this study are outlined later in this section.

Specifically, the chemical transformations of gaseous species, including OH, are simulated with the MELCHIOR2 chemical mechanism (Schmidt et al., 2001). Photolysis rates are computed with the FAST-JX module (Wild et al., 2000; Bian and Prather, 2002), taking into account the attenuation of solar radiation by both clouds and aerosols. The representation of dry deposition follows Zhang et al. (2001). Wet scavenging and coagulation of aerosol particles are simulated using the standard parameterizations described in Mailler et al. (2017) and Menut et al. (2013). The formation of SOA from biogenic emissions

of isoprene and terpenes is simulated with an oxidation scheme based on Kroll et al. (2006) and Zhang et al. (2007). Anthropogenic SOA is represented as a result of the oxidation of a few specific volatile SOA precursors (Bessagnet et al., 2008). The anthropogenic emissions of gaseous and aerosol species are specified using the Hemispheric Transport of Air Pollution (HTAP) v2 global emission inventory (Janssens-Maenhout et al., 2015). Biogenic emissions of several organic gases and NO

are calculated using the Model of Emissions of Gases and Aerosols from Nature (MEGAN v2.04) (Guenther et al., 2006).

Fire emissions are derived from satellite FRP data following the methodology detailed in Konovalov et al. (2014; 2017a; 2018). Briefly, the rate of emissions of a given aerosol or gaseous species from fires per unit area is assumed to be proportional to the FRP density. The proportionality coefficients include, in particular, the emission factor (depending on a vegetation land cover type), the empirical factor relating FRP to the BB rate, and the adjustable correction factor ($F_s$), which is

allowed to take different values for different species $s$. The emission factors for gaseous species are specified using the data from Andreae (2019). However, to simplify the auxiliary – for the given study – task of optimizing BB aerosol emissions, the BC and organic carbon (OC) emission factors are taken to be the same as in Konovalov et al. (2018; see Table 1 therein). The emission rates are modulated with the diurnal profile of BB emissions, which was derived directly from the MODIS FRP data (Konovalov et al., 2014, 2015). The correction factors are adjusted against the satellite data as explained in Sect.

15    2.6.

Following Konovalov et al. (2018), the maximum injection heights of BB emissions are calculated for each FRP pixel using the "two-step" parameterization of Sofiev et al. (2012) as a function of the observed FRP, the boundary layer height, and the Brunt–Väisälä frequency. The calculations of the injection heights of the BB emissions were validated by Sofiev et al. (2012; 2013) against vertical profiles of BB plumes derived from satellite observations by the Multi-angle Imaging SpectroRadiom-

eter (MISR) and CALIOP instruments (Mazzoni et al., 2007; Omar et al., 2009) and were found to be in reasonable quantitative agreement with these observations. In our simulations, the emissions for the given model layer are computed proportionally to the weighted number of pixels yielding the maximum injection height corresponding to this layer, with the weight of each pixel evaluated proportionally to the corresponding FRP. Konovalov et al. (2018) found that BB aerosol simulations that were constrained by the OMI and MODIS satellite observations of AAOD and AOD were consistent with in situ meas-

urements of BC and OC (Mikhailov et al., 2017) at the 300m tall mast at the Zotino Tall Tower Observatory (ZOTTO) situated at a remote location in central Siberia (within our study region). Taking into account that the AAOD and AOD observations provide constraints to only columnar concentrations of BC and OC, this finding can be regarded as indirect evidence that the calculated BB emission vertical profiles within the boundary layer are reliable.

To estimate the BB aerosol photochemical age (or, more strictly speaking, the solar radiation exposure time as defined in

Sect. 2.1), the chemical mechanism in CHIMERE is extended to include two gaseous tracers that have the same emissions as BB OC. One of the tracers, $T_1$, is treated as an inert gas, while another tracer, $T_2$, decays with time, but only when the local solar zenith angle is less than 90°. The decay rate, $k_T$, is set to be equal to $2.3 \cdot 10^{-5}$ s$^{-1}$ to represent – for definiteness – the mid-range of the half-lives for the decay of BrC in BB aerosol (12 h) according to the observational estimates by Forrister et

al. (2015). Using the columnar mass concentrations of the tracers, $[T_1]$, and $[T_2]$, the BB aerosol photochemical age (or, more strictly speaking, the solar radiation exposure time as explained in Sect. 2.1), $t_e$, is estimated as follows:

$$t_e = k_T^{-1} \ln([T_1]\,[T_2]^{-1}). \tag{7}$$

The simulations are performed using a 1 by 1 degree model mesh covering a major part of Northern Eurasia (22.5–136.5 ° E; 38.5–75.5 ° N) with 12 non-equidistant model layers in the vertical. The top of the upper layer is fixed at the 200 hPa pressure level. The model was run for the period from 28 May to 15 August 2016, but the first 18 days (until 15 June) are withheld as the spin-up period. Note that BB aerosol evolution in 2016 had not been addressed in our previous studies mentioned above. The CHIMERE simulations are driven with the meteorological data from the WRF (Weather Research and Forecasting; version 3.9) model (Skamarock et al., 2008) simulations with nudging toward the FNL reanalysis (NCEP, 2017).

Similar to Konovalov et al. (2015; 2017a), the simulations considered in this study are performed for three main scenarios. The first and second main scenarios (labeled here as 'bb_vbs' and 'bb_trc', respectively) represent the conditions with predominating air pollution from fire emission. For these scenarios, all sources of gases and aerosol, except for those due to fire emissions inside the CHIMERE domain, were disregarded. The 'bb_vbs' scenario corresponds to the BB OA representation formulated within the VBS framework as described in Sect. 2.4. In the 'bb_trc' scenario, which is ancillary in the framework of the given study, the BB OA representation largely follows that for anthropogenic OA in the "standard" version of CHIMERE (Menut et al., 2003), presuming, in particular, that primary OA consists of non-volatile material. In this study, this representation is further simplified by entirely neglecting SOA formation, which turned out to be negligible anyway in our previous BB aerosol simulations with the standard version of CHIMERE (Konovalov et al., 2015; 2017a; 2018). The simulations for the 'bb_trc' scenario are used to evaluate the BB aerosol tracers involved in Eq. (1). For both scenarios, particulate fire emissions of OC were distributed following a fine-mode log-normal distribution with a mass median diameter (MMD) of 0.3 µm and a geometric standard deviation of 1.6 (Reid et al., 2005a) among 10 size sections. A smaller value of MMD of 0.25 µm was assumed for BC emissions, partly based on the recent measurements by Morgan et al. (2020). Taking into account that a contribution of the coarse mode of aerosol particles to the optical properties of BB aerosol at the UV and visual wavelengths is likely negligible (Reid et al., 2005b), this mode was disregarded in our simulations. This simplification allows us to avoid large uncertainties associated with the representation of emissions and the evolution of coarse particles.

The third scenario (labeled below as 'bgr') represents the hypothetical conditions that are background with respect to the air pollution caused by fire emissions. The corresponding simulations were made with anthropogenic and biogenic emissions of particles and gases but without fire emissions. Monthly climatology for gas and aerosol concentrations from the LMDZ-INCA chemistry transport model (Folberth et al., 2006) was used as the boundary conditions. The AOD values in our analysis were computed in CHIMERE under the assumption that particle components are homogeneously mixed. The AAOD background values have not been evaluated in this study. As discussed in Konovalov et al. (2018), the main reason is that it is not known how the background part can be accounted for in the OMI AAOD retrievals for the "biomass burning" type of aerosol. Furthermore, the CHIMERE simulations of AAOD at 388 nm for the scenes dominated by anthropogenic pollution

are probably highly uncertain particularly due to poorly known and variable values of the imaginary refractive index for OA. Hence, a contribution of the background part of AAOD to the OMI AAOD retrievals is effectively neglected and is disregarded (unless indicated otherwise) in the analysis involving Eq. (1). To evaluate the total CO columns and AOD representative of all sources (anthropogenic, biogenic, and pyrogenic) of gases and aerosol, the corresponding values for the 'bb_vbs' or 'bb_trc' scenarios are summed up with values for the 'bgr' scenario. Note that by doing so we disregard any effects of anthropogenic emissions on the evolution of BB aerosol: according to our simulations, concentrations of aerosol and reactive gases in the analyzed situation are predominantly determined by fire emissions. Our computations of the optical properties of BB aerosol are explained in the next section.

In addition, we performed supplementary simulations for a test scenario (referred below to as 'bb_poa') in which all the oxidation reactions that were taken into account in the simulations for the 'bb_vbs' scenario were turned off but POA was still assumed to be composed of semi-volatile species. Some further details on this scenario are provided below in Sect. 3.3.

Figure 1 shows the model domain and illustrates the fire emissions used in our simulations by presenting the density of the total BB emissions of particulate matter (PM) according to the 'bb_vbs' scenario in the analysis period of 15-31 July 2016 (see Introduction). The same figure also introduces the study region that covers western and central parts of Siberia – where (as evident from the figure) major fires occurred during the analysis period – and also a part of Eastern Europe, including the European territory of Russia and territories of several Eastern European countries.

## 2.4 Model representation of the evolution and optical properties of BB aerosol

As noted above, one of the goals of our BB aerosol simulations is to interpret the evolution of BB aerosol optical properties that can be inferred from satellite observations as described in the previous section. However, there is currently no uniform way to represent atmospheric transformations of BB organic aerosol (OA) and its optical properties in CTMs. Typically, model representations of the evolution of OA aerosol involve organic aerosol (OA) oxidation schemes that are designed using the volatility basis set (VBS) framework (Donahue et al., 2006; Robinson et al., 2007), but it is challenging to choose an appropriate VBS scheme among those suggested in the literature. Furthermore, differences between the available VBS schemes (e.g., Shrivastava et al., 2015; Konovalov et al., 2015; Ciarelli et al., 2017; Tsimpidi et al., 2018), which have been proposed for simulations of BB OA aerosol and are, to some extent, constrained by observational or experimental data, have been shown to result in major quantitative and even qualitative differences in the simulated multi-day evolution of BB OA mass concentration (Konovalov et al., 2019). Therefore, it is not given that any of the available schemes can adequately describe the BB aerosol evolution specifically in Siberia.

Taking into account that the real BB aerosol evolution depends on numerous variable factors, most of which cannot presently be constrained by available observations in Siberia, we do not attempt to represent the many inter-related aging processes that can affect the BB aerosol evolution in a quantitatively accurate way. Instead of choosing any concrete OA oxidation scheme among those previously described in the literature, we designed a highly simplified VBS scheme, which takes into

account only basic processes driving the OA evolution and can be fitted to the satellite observations by adjusting a few parameters. This scheme was used in our simulations for the 'bb_vbs' scenario.

As the basis for our OA oxidation scheme, we used a relatively simple "1.5-dimensional" VBS scheme proposed and described in detail by Ciarelli et al. (2017) (abbreviated below as C17). The original scheme addresses the evolution of several surrogate organic species, each of which is given the volatility, molecular weight, and oxidation state. More specifically, all semi-volatile organic compounds (SVOCs) are split into three sets, such as (1) the POA set, (2) the set of SOA species formed as a result of oxidation of the POA species, and (3) the SOA set containing products of oxidation of volatile organic compounds (VOCs) or intermediate-volatile organic compounds (IVOCs). Real VOC and IVOC species are represented by surrogate species referred to as NTVOCs ("non-traditional" VOCs). Some parameters of the C17 scheme, including the mass ratio of NTVOC and SVOC emissions and the enthalpies of vaporizations for SVOCs, were constrained with mass-spectrometric measurements of organic gases and particles from combustion of beech logs in a residential wood burner. The SVOC species are split into five volatility classes, with the volatilities ($C*$) ranging from $10^{-1}$ to $10^3$ μg m$^{-3}$. The constrained VBS scheme yielded a good agreement of the measurements with corresponding box-model simulations under OH exposures that are equivalent to about 10 to 15 hours of evolution under typical atmospheric conditions (Ciarelli et al., 2017).

Our simplified oxidation scheme includes only two aggregated volatility classes of the POA species, one of which aggregates the three original classes of organic compounds with relatively low volatilities (LV) ($C* \in [10^{-1}; 1]$ μg m$^{-3}$), and another represents mostly medium-volatility (MV) primary compounds with volatilities ranging from $10^1$ to $10^2$ μg m$^{-3}$. The corresponding model species are denoted below as LV-POA and MV-POA. The volatilities of LV-POA and MV-POA were set in our simulations at 1 and $10^2$ μg m$^{-3}$, respectively. Note that the gas-particle partitioning of MV-POA emissions in our simulation may be similar to that of more volatile organic species since small but dense BB plumes are effectively diluted over the size of a grid cell. Box model simulations performed with varying initial size and density of the BB plume indicated that such artificial dilution can unrealistically enhance SOA formation (Konovalov et al., 2019). For these reasons, the BB emissions of SVOCs species with a volatility of ~$10^3$ μg m$^{-3}$ are assumed to be also represented by MV-POA. The fire emissions of POA were split between LV-POA and MV-POA species based on a ratio of 3 to 4 following the estimates by May et al. (2013).

In our simulations, we merged the two SOA sets that had been originally introduced in C17. Similar to primary organic compounds, secondary compounds with volatilities lower than or equal to 1 μg m$^{-3}$ are aggregated as one model species, LV-SOA. To allow for a more realistic representation of the multi-generation aging of SOA, we retained a distinct SOA species with relatively high volatility (HV) of $10^3$ μg m$^{-3}$. Our scheme also involves an MV-SOA species with a volatility of $10^2$ μg m$^{-3}$. The POA and SOA species were given the same molecular weight as their respective analogs from the second, fourth, and fifth volatility classes of the SVOC sets 1 and 3 from the original C17 scheme. The properties of the VBS species in our model are listed in Table 1.

The reactions representing the evolution of BB OA within our simplified VBS scheme are specified in Table 2. Specifically, the POA species in the gas phase are assumed to undergo reactions (R1) and (R2), where the stoichiometric factor of 1.3 is introduced to account for the difference between the molecular weights for SOA from the second volatility class of set 2 of the C17 scheme and the merged SOA set in our model (189 and 144 g mol$^{-1}$, respectively). Following C17, we assume that apart from the oxidation of POA, SOA species are produced from the oxidation of NTVOCs (see reaction (R3)). However, we simplified the SOA yields: the direct oxidation of NTVOCs is assumed to result in the formation of only HV-SOA and LV-SOA, while the formation of MV-SOA is disregarded. The SOA yields for HV-SOA and LV-SOA are defined to retain the same total yield of SOA mass from the oxidation of NTVOCs as in C17 (~ 43 g mol$^{-1}$). The stoichiometric coefficients for these yields are determined by the factors (0.33 and 0.30) accounting for the difference between the molar masses of HV-SOA and LV-SOA ($131/144 \cong 0.30/0.33$) and by the adjustable parameter $\xi_1$ which was varied in our test simulations to enable consistency of our simulations with the satellite observations considered (see Sect. 2.6). The simulation results presented in this paper were obtained with $\xi_1$ equal to 0.85.

Oxidation of gas-phase HV-SOA is assumed to yield MV-SOA (reaction R4). Since the molecular weight of MV-SOA is larger than that of HV-SOA (see Table 1), functionalization is presumed to effectively dominate – although only slightly – over fragmentation in the oxidation of "high-volatility" organic species. However, fragmentation is assumed to dominate over functionalization in the oxidation of both MV-SOA and LV-SOA in (R5) and (R6): the reaction yield is controlled by another adjustable parameter, $\xi_2$, which is set in our simulations at the small value of 0.15 (see Sect. 2.6 for the adjustment procedure). This means that OH oxidation of MV-SOA and LV-SOA yields 15% of LV-SOA by a functionalization pathway, while the fragmentation pathway accounts for 85% of products, which are considered as volatile and do not contribute to SOA formation. Lower volatility of the SOA species is associated with a higher O/C ratio (see C17), thus making less volatile species more prone to fragmentation. The reaction rates ($k_{OH}$) for reactions (R1)-(R4) are taken to be the same as those for similar reactions in C17, but a smaller rate is set for reactions (R5) and (R6), presuming that they represent multi-generation fragmentation processes.

One more adjustable parameter – $\xi_3$ – in our simplified VBS scheme is the emission ratio of the mass concentration of NTVOCs to the sum of mass concentrations of the POA species, which was set at 14.2. The same parameter was estimated in C17: its optimal value was found to be 4.75 (that is, three times less than in our case), although a much larger value (up to 9.8) was found to be also consistent with the measurements analyzed in C17. As noted above, the results of the experiments performed in C17 do not directly apply to the case of BB aerosol from Siberian fires. Hence, it does not seem infeasible that oxidation of VOC emissions from Siberian fires indeed effectively yields more SOA than oxidation of VOCs from the combustion of beech logs.

Enthalpies of vaporization ($\Delta H_{vap}$) of the surrogate OA species are evaluated using the estimates derived by May et al. (2013) from thermodenuder measurements of fresh BB aerosol. The same $\Delta H_{vap}$ values (depending on the volatility) are used both for POA and SOA species. Following C17, we assume, for definiteness and simplicity, that all organic species within

particles form a well-mixed, liquid, and inviscid solution. This assumption implies, in particular, that POA species can affect partitioning of SOA species according to Raoult's law, and vice versa.

To get a tentative idea about whether or not our simplified VBS scheme enables a sufficiently realistic representation of the BB aerosol evolution (especially during the first few hours, which, as is discussed below, are not represented in satellite

observations), we simulated the BB OA evolution under the conditions of chamber experiments reported in C17 by using the microphysical box model described in Konovalov et al. (2019) with both the simplified and original (C17) VBS schemes. The simulations are described in Supplementary Material, Sect. S2, and the results are presented in Fig. S1 in comparison with available chamber measurements. These results indicate that the simulations of the BB OA evolutions with our simplified scheme are physically reasonable (see Sect. S2 for details).

The optical properties of BB aerosol particles (for both 'bb_vbs' and 'bb_trc' scenarios) are evaluated using the OPTSIM software (Stromatas et al., 2012) under the assumption that any particle is composed of a spherical BC core surrounded by a concentric shell consisting of homogeneously mixed organic components, inorganic ions, and water. Given the simulated size-resolved mass concentration and composition of BB aerosol particles, as well as the complex refractive index for each particle component, the scattering and absorbing efficiencies for the particles are computed for the core-shell mixing scenar-

io with a Mie code based on the formulations proposed by Toon and Ackerman (1981). The scattering and absorption efficiencies are then used to compute both the AOD and AAOD values. Note that Mie core/shell models in which a BC core is represented as a perfect sphere are prone to underestimation of the fractional contribution of BC to absorption at near-UV wavelengths in cases where the coating is relatively thin (with the shell/core mass ratio less than 3) as a result of the fact that actual BC particles are typically non-spherical aggregates (Liu et al., 2017; Taylor et al., 2020). However, this potential un-

derestimation is unlikely to be significant in our simulations representing typical Siberian BB aerosol, which features typical OC/BC mass ratios as large as almost 30 (Mikhailov et al., 2017).

The real part of the refractive index for all of the BB aerosol components resolved in our simulation (see Sect. 2.3), as well as the imaginary part of the refractive index for the inorganic components (including BC), were taken in our computations to be the same as those given in Stromatas et al. (2012) for the 532 nm wavelength (see Table 3 therein) and are assumed for

simplicity to be the same for both the 388 and 550 nm wavelengths considered here. In the 'bb_trc' simulation, BB aerosol absorption was calculated under the assumption that the organic shell is non-absorbing. Evaluation of the imaginary part of the refractive indexes for the POA and SOA species, $k_{poa}$ and $k_{soa}$, in our simulations for the 'bb_vbs' scenario involved the following assumptions.

First, based on the experimental findings (Saleh et al., 2014) that almost all BrC absorption of fresh BB aerosol is associated

with extremely low-volatility organic compounds, we assumed that MV-POA is not absorbing. Based on the estimates of the imaginary refractive index of fresh OA, $k_{OA}$, and their uncertainties, which have been reported by Lu et al. (2015) as a function of the BC-to-OA ratio, and assuming average values of the BC-to-OC and POA-to-OC ratios for Siberian BB aerosol to be about 0.04 and 1.8, respectively (Mikhailov et al., 2017; Konovalov et al., 2017b), we conservatively estimated $k_{OA}$ to be

0.013. This estimate, which is in the lower range of the corresponding $k_{OA}$ values ($k_{OA} \approx 0.025 \pm 50\%$) given by Lu et al. (2015), is expected to ensure that the contribution of POA to BB aerosol absorption in our simulations is not overestimated. Accordingly, taking into account the assumed MV-POA / LV-POA emission ratio (see above), we estimated $k_{poa}$ (for LV-POA) to be 0.03.

Second, we assumed that, as a result of UV photodegradation, $k_{poa}$ decreases exponentially with the BB aerosol photochemical age ($t_e$). Taking into account the laboratory results by Fleming et al. (2020), we assumed the lifetime of BrC chromophores in the LV-POA species to UV photodegradation to be 82 h. This lifetime represents the lowest part of the range of the experimental values reported by Fleming et al. (2020, see Table 3 therein).

Third, we assumed that in contrast to POA, the low-volatility SOA, LV-SOA, is not absorbing. This assumption does not contradict the aforementioned experimental findings (Saleh et al., 2014) that indicated high absorptivity of low-volatility organic compounds, since the contributions of POA and SOA species to the absorption were not isolated in these experiments, and the effect of SOA addition was found to be comparable to measurement uncertainties. Furthermore, if low-volatility SOA remaining after more rapid atmospheric processing of higher-volatility SOA would be strongly absorbing, it would be difficult to explain an almost total loss of BrC absorption of aging BB aerosol (Forrister et al., 2015), in which POA is typically replaced by oxidized species (e.g., May et al., 2015). While oxidation of low-volatility POA is likely to yield absorptive components (Wong et al., 2017), chemical processing of LV-POA in our simulations is very slow (as shown below), and so the corresponding source of BrC is disregarded. For simplicity, the HV-SOA and MV-SOA species are assumed to have the same constant values of the imaginary refractive index, $k_{soa}$. This value was adjusted in our simulations for the 'bb_vbs' scenario (see Sect. 2.6) and is taken to be equal to 0.009 for the 388 nm wavelength. Photochemical processing of HV-SOA and MV-SOA resulting in the formation of non-absorbing LV-SOA implicitly accounts in our simulations for the destruction of BrC chromophores through all possible mechanisms, including gas-phase and heterogeneous oxidation and UV photodegradation. To the best of our knowledge, estimates of the imaginary refractive index for SOA formed from photo-oxidation of BB emissions have so far been reported only by Saleh et al. (2013) based on the analysis of smog chamber experiments. According to these estimates, the imaginary refractive index (at 388 nm) for SOA is at least 0.04 (as in the case of SOA from pocosin pine) or even much larger. However, these estimates correspond to a very initial stage of BB aerosol aging (when the UV exposure time is less than 1.5 hours), which is not addressed in our analysis, and therefore maybe not applicable to the more oxidized SOA compounds that are probably dominating aged BB aerosol particles.

Overall, except for a slow degradation of BrC by UV in primary aerosol particles, we assume that each molecule of any given SVOC species contains a constant fraction (that can be different for different species) of chromophores and that atmospheric evolution of BrC within BB aerosol particles is determined by the condensation or evaporation of the different primary and secondary organic compounds forming BB aerosol particles. This approach allows us to parameterize the evolution of the optical properties of BB aerosol within the CTM and the evolution of its chemical composition consistently.

To characterize the effects of water uptake by BB aerosol particles on our computations of their optical properties, we evaluated the hygroscopicity parameter $\kappa_{org}$ (Petters and Kreidenweis, 2007) for BB OA by assuming that all the SOA species feature a constant hygroscopicity parameter, $\kappa_{org}$ (Petters and Kreidenweis, 2007), of 0.2 and that the POA species are hydrophobic. The chosen value of $\kappa_{org}$ for SOA is representative of the mid-range of the set of $\kappa_{org}$ measurements for oxidized

organic compounds generated in a flow reactor (Lambe et al., 2011). Also, we took into account the water uptake by inorganic species using the equilibrium concentrations calculated in CHIMERE with the ISORROPIA module (Nenes et al., 1998). The main features of our simulations for the different modeling scenarios defined in Sect. 2.3 are summarized in Table 3.

Finally, we would like to emphasize once again that our simplified parameterization of the physical, chemical and optical

properties of BB aerosol is not aimed at a quantitatively accurate representation of the actual very complex processes. We believe that a quantitatively accurate representation of the BB aerosol properties and their evolution in CTMs is presently not feasible in a general case due to the lack of the necessary observational constraints. However, we consider our study as a step forward towards the development of a simple and robust but yet physically sound parameterization ensuring adequate simulation of BB aerosol properties and evolution in chemistry transport and climate models.

**2.5 Pre-processing and harmonization of the satellite and model data**

At the preparatory stage of our analysis, the Level-2 (orbital) AAOD, AOD, and CO satellite data were projected onto the model grid with an hourly temporal resolution (corresponding to the temporal resolution of the output data from CHIMERE). Different pixels falling into the same grid cell were averaged and matched to the corresponding simulated value. To select a particular AAOD retrieval among those corresponding to the different altitudes of the aerosol center mass (see Sect. 2.2), the

altitude of an observed BB plume was estimated using the mass concentration of the total particulate matter from the CHIMERE simulations for the "bb_vbs" scenario. In this way, we harmonized the OMI AAOD retrievals with the corresponding simulations (similar to Zhang et al. (2015) and Konovalov et al. (2018)). As noted in Sect. 2.2, we also used the "final" AAOD retrieval product, which is not affected by possible errors of our model, but still maybe not free of biases due to differences between the actual and assumed "climatological" heights of the BB plumes.

For our analysis of $EnR_{abs}$ or $EnR_{ext}$, the MODIS and IASI observations collocated – at the scale of a model grid cell – with the OMI observations were selected by requiring that the absolute value of the time difference between the measurements taken by OMI and MODIS as well as by OMI and IASI is as small as possible and does not exceed two hours. The observed BB plumes are assumed to be mostly of a large spatial scale (tens of km) and slowly evolving at a scale of a few hours. Accordingly, the temporal and spatial inconsistencies between the different satellite data are expected to result mostly in ran-

dom uncertainties in our estimates of the $EnR_{abs}$ or $EnR_{ext}$, which can be taken into account by the corresponding confidence intervals. For the ancillary analysis that does not involve the OMI AAOD observations (which are much sparser than the MODIS AOD and IASI CO observations), we matched only the available AOD and CO observations, again requiring that the absolute value of the time difference between these observations does not exceed 2 hours.

The simulated AAOD, AOD, and CO data were matched in both space and time to the corresponding satellite data. The simulated 3-D concentration fields of CO were first processed to compute the total CO columns using the IASI averaging kernels as described in Konovalov et al. (2014; 2016). As an additional step towards the harmonization of the satellite and simulation data, we introduced a selection criterion for the simulated data, which accounts for the fact that the OMI AAOD data considered in this study are representative only of the scenes strongly affected by fire emissions. In the framework of the OMAERUV retrieval algorithm (Torres et al., 2013), such scenes were selected based on real-time AIRS (Atmospheric Infrared Sounder) retrievals of CO. Accordingly, the grid cells for which the OMAERUV data are available correspond to larger magnitudes of the retrieved CO columns than the grid cells for which the OMAERUV are not provided. However, due to errors in the fire emissions and computations of the air pollution transport, the observed "hot spots" in the CO columns do not necessarily correspond to similarly elevated values in the simulated CO columns, AAOD, and AOD. Such a mismatch between the OMAERUV data and simulations can result in systematic biases in estimates of $EnR_{abs}$. To avoid these possible biases, we first arranged (ranked) the retrieved CO columns with respect to their magnitudes. A similar ranking was performed for the simulated CO columns. Then we required that the simulated CO data (and the simulated AOD and AAOD data matching the CO columns in space and time) selected for our analysis have the same ranks as the retrieved CO columns. Using this criterion, we tried to mimic the selection procedure which was applied to the OMI observations in the framework of the OMAERUV algorithm in the case of the simulated data. An advantage of this selection procedure is that it does not involve any subjective quantitative criteria and its outcome is rather insensitive to the mean level of the CO fire emissions in the model. Finally, some outliers (too large and too small values of $EnR_{abs}$ and $EnR_{ext}$) were removed using the three-sigma rule.

Applying all the selection criteria left us with a sufficiently large number (1156) of data points (in the main data set) suitable for our statistical analysis using the method outlined in Sect. 2.2. As the OMI SSA retrievals are very sparse compared to the AAOD retrievals, we did not apply any selection criteria to the SSA data except for the common spatial and temporal windows.

Note that the harmonization procedure described above presumes that the satellite retrievals which are available only for clear sky conditions are sufficiently representative of a whole grid cell, part of which can be covered by clouds. The underlying assumption is that in the analyzed period in the middle of summer, when there was little precipitation in the study region, the fire emissions and the corresponding atmospheric loading of BB aerosol were independent, in the statistical sense, of the cloud coverage in a given location. To ensure that the masking out of cloudy scenes in the satellite retrievals does not entail any major systematic discrepancies between the retrievals and simulations, we examined the difference of the observed and retrieved values of AOD in a given grid cell in the study region and period as a function of the number of the available retrievals per grid cell and did not find any considerable relationship between the two characteristics.

Figure 2 illustrates the "ancillary" (see above) data sets of AOD observations and simulations. In addition, Figure S2 in the Supplementary Material shows similar data for the CO columns. More specifically, these figures show the AOD and CO

spatial distributions averaged over the analysis period (15-31 July 2016) in comparison to the similar distributions for the preceding period (15 June -14 July 2016) when there were no strong fire emissions in the study region. These figures also introduce two special regions used in our analysis, one of which (referred to below as the "source" region) includes the locations of major Siberian fires that emitted BB plumes transported afterward to the Eastern European part of Russia and is expected to contain relatively fresh BB aerosol, and another (referred to below as the "receptor" region), which is expected to represent aged BB aerosol. The spatial distributions shown indicate that the Siberian fires caused strong enhancements of the concentrations of BB aerosol and CO in the troposphere over a big part of Eastern Europe, which is further discussed in Sect. 3.

A final step of our procedure aimed at harmonizing the satellite and simulation data involved the estimation of probable biases in the background values of CO columns and AOD. (It may be useful to recall that the background part of AAOD is neglected in our analysis.) Similar to our previous studies mentioned above, these biases were estimated by averaging the differences between the simulated and observed data representative of background conditions. A given scene was assumed to be representative of the background conditions with respect to a given characteristic (CO columns or AOD) if a corresponding simulated value for the 'fires' scenario did not exceed 10 % of the respective value for the 'bgr' scenario. In this study, the averaging is done over the period from 15 June to 15 August 2016 separately for the source region, the receptor region, and the rest of the study region. The estimated biases are then applied to the simulations of AOD and CO columns for the corresponding regions under the 'bgr' scenario.

### 2.6 Adjustment of the model parameters

As explained above (see Sect. 2.4), our model representation of the BB aerosol aging processes for the 'vbs' scenario involves the three adjustable parameters, $\xi_1$-$\xi_3$, controlling the evolution of the POA and SOA species, and the imaginary refractive index, $k_{soa}$, controlling the absorptive properties of SOA. Besides, we need to adjust correction factors, $F_s$ (see Sect. 2.3), specifically those controlling the fire emissions of BC, POA and CO. Note that the correction factor for CO emissions, $F_{CO}$, is applied to the emissions of all other gaseous species, except for those of NTVOCs (which were determined by scaling the POA emissions). Taking into account that AAOD for BB aerosol in Siberia is typically an order of magnitude smaller than AOD (e.g., Konovalov et al., 2018), we expected that the modeled evolution of AOD is not sensitive to $k_{soa}$ and also not sensitive to the correction factor for BC emission, $F_{BC}$. Therefore, the parameters $\xi_1$-$\xi_3$, as well as the correction factors for the POA and CO emissions from fires, can be adjusted independently of $k_{soa}$ and $F_{BC}$. Specifically, we required that the nonlinear trend (see Eq. 5) in EnR for the simulated AOD be consistent with the corresponding nonlinear trend in the MODIS observations:

$$\left| y_{ext}^{obs}(t_e) - y_{ext}^{sim}(t_e) \right| < \Delta_{ext}^{obs}(t_e) + \Delta_{ext}^{sim}(t_e), \ \ t_e \in [t_e^{min}; \ t_e^{max}], \tag{8}$$

where $y_{ext}^{obs}(t_e)$ is the approximation of $EnR_{ext}$ for the AOD observations, $y_{ext}^{sim}(t_e)$ is a similar approximation for the AOD simulations, $\Delta_{ext}^{obs}$ and $\Delta_{ext}^{sim}$ are the 95% confidence intervals for $y_{ext}^{obs}$ and $y_{ext}^{sim}$, respectively, and $t_e^{min}$ and $t_e^{max}$ are the minimum and maximum values of the photochemical age in the selected dataset.

In the presence of significant nonlinear variations of $y_{ext}^{obs}$, the condition given by Eq. (8) can provide sufficiently strong observational constraints to the three parameters $\xi_1$-$\xi_3$ of our VBS scheme. To optimize the fire emissions for POA by adjusting the corresponding correction factor ($F_{POA}$), we further required (following Konovalov et al., 2018) that the average values of the AOD simulated using an optimal value of $F_{POA}$, $\tau_{ext}^{sim}$, and of the AOD derived from MODIS observations, $\tau_{ext}^{obs}$, be approximately equal:

$$\left| \langle \tau_{ext}^{sim} \rangle - \langle \tau_{ext}^{obs} \rangle \right| \langle \tau_{ext}^{obs} \rangle^{-1} < o, \tag{9}$$

where the angular brackets denote the averaging performed over (only) the source region and analysis period, and $o$ is the relative error, which is set to be equal to 0.05 in this study.

Similar conditions providing observational constraints to $k_{soa}$ and $F_{BC}$ apply to the simulations of AAOD:

$$\left| y_{abs}^{sim}(t_e) - y_{abs}^{obs}(t_e) \right| < \Delta_{abs}^{sim}(t_e) + \Delta_{abs}^{obs}(t_e), \ \ t_e \in [t_e^{min}; t_e^{max}], \tag{10}$$

$$\left| \langle \tau_{abs}^{sim} \rangle - \langle \tau_{abs}^{obs} \rangle \right| \langle \tau_{abs}^{obs} \rangle^{-1} < o. \tag{11}$$

Note that the trend $y_{abs}^{sim}(t_e)$ involved in Eq. (10) is expected to depend on both the correction factor $F_{BC}$ and the imaginary refractive indexes $k_{poa}$ (which is fixed) and $k_{soa}$. Taking into account Eq. (11), larger values of $F_{BC}$ would require smaller values of $k_{soa}$. In turn, smaller values of $k_{soa}$ would suppress the variability of $y_{abs}^{sim}(t_e)$ (which is expected to depend also on parameters $\xi_1$-$\xi_3$) and, therefore, the condition (10) would require a smaller value of $F_{BC}$. Such reasoning shows that if $y_{abs}^{obs}$ exhibits significant changes with $t_e$, then the conditions (10) and (11) allow constraining each of the two parameters, $k_{soa}$, and $F_{BC}$, independently.

It should be noted that Eqs. (8)-(11) do not imply a rigorous minimization procedure but impose only approximate constraints on the parameters. Indeed, constraining the six parameters controlling our simulations of AAOD and AOD within the VBS framework is a challenging (even though not infeasible) computational task. On the other hand, the relatively "loose" constraints defined above allowed us to adjust the parameter values with a reasonable accuracy manually by running the model multiple times and iteratively varying the parameter values using the trial-and-error method. More specifically, starting from a priori estimates of $F_{POA}$ and $F_{BC}$ based on our previous studies and estimates of $\xi_1$-$\xi_3$ based on Ciarelli et al. (2017), we first adjusted the parameters $\xi_1$-$\xi_3$ and $F_{POA}$ in an iterative process. At the next iteration cycle, using the optimized values of $\xi_1$-$\xi_3$ and $F_{POA}$, we adjusted the parameters $k_{soa}$ and $F_{BC}$. These iteration cycles were repeated to ensure the consistency of all the parameters. The optimized values for $\xi_1$-$\xi_3$ and $k_{soa}$ are indicated above in Sect. 2.4 and in Tables 2 and 3.

A similar but much simpler optimization procedure was realized in the case of the simulations with the standard version of CHIMERE (that is, for the 'bb_trc' scenario). Specifically, the parameters of the VBS scheme did not need to be adjusted by

definition, and only a pair of iterations were needed to adjust the correction factors $F_{POA}$ and $F_{BC}$ under conditions (9) and (11). The CO emissions from fires were optimized in the same way as the POA and BC emission for the 'bb_trc' scenario.

For the 'bb_vbs' scenario, the optimized values of $F_{POA}$ and $F_{BC}$ were found to be 0.9 and 1.5, respectively. With these factors, the 'bb_vbs' simulation yields a mean ratio of BC and OC mass concentrations of 0.035, which is a quite realistic value, given the estimate (0.036± 0.009) previously derived by Konovalov et al. (2017b) from the AERONET measurements as well the average value (0.038) of the highly variable BC/OC ratios observed for BB aerosol in central Siberia (Mikhailov et al., 2017). This finding indicates that the absorption closure carried out in our numerical experiments is sufficiently adequate. The optimized value of $F_{BC}$ is consistent with the $F_{BC}$ estimate (1.5±0.5) obtained in Konovalov et al. (2018) for the Siberian fires that occurred in July 2012. However, $F_{POA}$ is much smaller than the corresponding estimate of $F_{OC}$ (2.3±0.6), which was also reported in Konovalov et al. (2018). This difference accounts for the strong SOA formation in our simulations for the 'bb_vbs' scenario, which was essentially disregarded in the simulations performed in Konovalov et al. (2018) (where BrC absorption, on the other hand, was taken into account implicitly using an empirical parameterization). For the 'bb_trc' scenario, the optimized values of $F_{POA}$ and $F_{BC}$ were found to be 2.8 and 2.9, respectively. The difference between the optimal values of $F_{POA}$ for the two scenarios is qualitatively consistent with similar findings from our previous studies (Konovalov et al., 2015, 2017a) and with underestimation of AOD by simulations in which SOA formation was treated as a minor process (e.g., Petrenko et al., 2012; Tosca et al., 2013; Reddington et al., 2016). A much larger value of $F_{BC}$ in the 'bb_trc' scenario compared to that in the in 'bb_vbs' is indicative of a major contribution of BrC to BB aerosol absorption in the 'bb_vbs' scenario, since the organic matter was assumed to be non-absorbing in the 'bb_trc' simulation. The $F_{CO}$ factor was set to 1.9 for both 'bb_vbs' and 'bb_trc' scenarios.

The optimized correction factors allow obtaining "top-down" estimates of the total amounts of OC and BC emissions from fires in the study region. Specifically, the total emissions of OC (in particles) in the study region in July are found to be 1.0 and 2.8 Tg, respectively, for the 'bb_vbs' and 'bb_trc' scenarios. The big difference between the estimates reflects a strong dependence of the top-down estimates of the BB OA emissions on a model representation of the OA processes (Konovalov et al., 2015). The corresponding BC emissions are estimated as 70 and 136 Gg. As noted above, the difference between the BC emission estimates is mainly due to a missing contribution of BrC to the absorption in the simulations for the 'bb_trc' scenario. For comparison, according to the GFED4.1s inventory (van der Werf et al., 2017), the same fires emitted 0.96 Tg OC and 51 Gg BC. These "bottom-up" estimates are evidently much closer to our estimates for the 'bb_vbs' scenario than for the 'bb_trc' one. The good agreement of the bottom-up and top-down estimates of the OC emissions is not necessarily meaningful: as argued in Konovalov et al. (2015), the consistency of top-down and bottom-up estimates of the BB OA emissions should be examined by taking into account the partitioning of the measured OC emissions between gases and particles, but the necessary data are not provided as part of emission inventories. Our estimate of the BC emissions for the 'bb_vbs' scenario is indicative of the underestimation of the Siberian BB BC emissions in the GFES4.1s emission inventory, consistent with the findings of previous analyses (Hao et al., 2016; Konovalov et al., 2018). Note, however, that the present

study is not designed to evaluate the uncertainties in the top-down emissions estimates. As indicated by the more detailed analysis in Konovalov et al. (2018), the uncertainty in our estimate of the BC emissions is likely at least 35 %. Accordingly, the top-down emission estimates reported above do not allow us to make any certain conclusions about the accuracy of the BC or OC emission estimates provided by the GFED4.1s inventory.

## 3 Results

### 3.1 Analysis using the two-region approach

In this section, we provide some preliminary characterization of the input data for our analysis, specifically by considering the spatial distributions and time series of both the satellite observations and corresponding simulations. We also examine the aging-driven changes in the optical properties of BB aerosol by considering the satellite and simulated data for the two regions ("source" and "receptor") introduced above (see Sect. 2.5 and Fig. 2). The source region includes locations of major Siberian fires, and the receptor region was affected by the aged BB plumes transported from the source region (see Fig. 2 and S2). Spatial averaging of the observed characteristics over a big region suppresses the variability associated with individual air masses.

The analysis presented in this section allows us to get some preliminary insights into the effects of atmospheric aging on the BB aerosol optical properties. The advantage of this analysis is that it does not rely on any quantitative estimates (which may be inaccurate) of the BB aerosol photochemical age and is more "transparent" than the more general analysis presented in Sect. 3.2. However, such an analysis has serious limitations, as it can provide only crude snapshots of the evolving BB plumes and does not involve the characterization of statistical uncertainties.

The spatial distributions of the simulated average values of AOD and CO columns in the study region during the analyzed period are in a reasonable (although not perfect) agreement with the corresponding fields of the satellite data, both in Siberia and Europe (see Fig. 2a, c, and Fig S2a, c). Specifically, the satellite and model data (both for AOD and CO) exhibit similar big enhancements in the source region and also similar smaller but still considerable enhancements in the receptor region (compared to the period without strong BB emissions in the study region, see Fig. 2b, d and Fig. S2b, d). It is noticeable, however, that CHIMERE yields too high AOD values compared to MODIS in the eastern and south-eastern parts of the model domain (cf. Figs. 2a and 2c), where, on the contrary, the simulated CO columns tend to be lower than those retrieved from the IASI observations in the same parts of the domain. We suppose that these differences (which do not significantly affect the results of our analysis because the affected grid cells are mostly outside of the study region) may be caused by the spatial variations in several factors and parameters involved in our simulations, including the emission factors for aerosol and gaseous species, volatility distributions of the POA emissions, and the relationship between FRP and the BB rate. In turn, these variations may be due to the inhomogeneity in the spatial distribution of the vegetation species across the region covered by the model domain: in particular, while the dominant tree species in the forest of western Siberia are pine, spruce,

fir, birch, and aspen, the most abundant tree species in eastern Siberia (including the eastern part of the study region) is larch (Schepaschenko et al., 2011). According to available measurements (May et al., 2014; Hennigan et al., 2011), both the BB OA emission factors and SOA formation for different tree species exhibit strong diversity, while the relationship between FRP and the BB rate can be affected by relative prevalence of crown and ground fires, which differs in forests dominated by

different tree species (Schulze et al., 2012).

The time series corresponding to the spatial distributions shown in Figs. 2 and S2 are presented and discussed in the Supplementary Material (Sect. S3, Figs. S3 and S4). Similar to the spatial distributions, the time series of the simulations taking fire emissions into account are found to be in good agreement with the observations both in the source and receptor regions, with the correlation coefficient for the spatially averaged daily values of the observations and simulations exceeding 0.85.

Taking into account that CO has a relatively long lifetime typically exceeding 15 days in the troposphere over continental regions in summer (Holloway et al., 2000), we consider the results of the comparison of the simulated and retrieved CO columns as an indication of the good performance of CHIMERE in capturing both the emissions and transport of the BB plumes during the studied period. Furthermore, taking into account that satellite retrievals of CO columns are known to be sensitive to the CO vertical distribution (e.g., George et al., 2009) (and, as noted in Sect. 2.5, this sensitivity has been ad-

dressed in our simulations by applying the averaging kernels), this comparison is also indicative of the adequacy of the simulated vertical profiles of the BB plumes.

The spatial distributions of the OMI AAOD observations and corresponding simulations (for the 'bb_vbs' scenario) for the study region and analysis period are shown in Fig. 3. Note again that the AAOD data analyzed in this study represent only BB aerosol. Qualitatively, these distributions are similar to the corresponding distributions of AOD and CO (see Figs. 2a, c,

and S2a, c), but the data are much more sparse, especially in the receptor region. Nonetheless, both the AAOD observations and simulations show numerous "hot spots" of AAOD in the source region and are also indicative of the major outflow of BB plumes from Siberia into the European territory of Russia.

Figure 4 shows the time series of the spatially averaged AAOD values according to the OMI observations and our simulations for the 'bb_vbs' and 'bb_trc' scenarios. It should be recalled that the background AAOD values were not computed in

our simulations and thus are not shown. As the OMI AAOD data representing BB aerosol are almost absent for both regions outside of the analysis period, only the data for the analysis period are presented in Fig. 4. Note that for the receptor region, there is not enough data to cover even the whole analysis period. The time series for the corresponding AOD and CO values (that were selected consistently with the AAOD observations) are also shown in Fig. 4.

It can be seen that the simulations for both 'bb_vbs' and 'bb_trc' scenarios are in good agreement with the AAOD observa-

tions in the source region. In the receptor region, however, the simulations for the 'bb_trc' scenario overestimate AAOD for all four days for which the data are available and are biased high by ~22 % on average. The AOD simulations for the same scenario also tend to overestimate the observations, although the bias is smaller (~16 %). In contrast, the corresponding simulations of the CO columns are practically not biased. These results indicate that the decrease of the observed values of both

AAOD and AOD in the receptor region compared to the source region cannot be fully explained by the processes (such as dilution, dry and wet deposition, and coagulation) included in the simulations for the 'bb_trc' scenario. As argued below (see Sect. 3.3), these additional changes in AAOD and AOD are primarily caused by losses of the medium-volatility fraction of SOA due to fragmentation. It is noteworthy that the simulations for the 'bb_vbs' scenario, which take into account BB aero-
sol aging, do not exhibit any bias in the case of AOD and show only a minor negative bias of ~4 % in the case of AAOD. Note that the magnitude of the biases is not quite properly illustrated in Fig. 4a, because different "daily" points represent significantly different numbers of spatial grid cells. Overall, the analysis presented in this section indicates (i) that BB aerosol transported from the source region to the receptor region was affected by aging processes that resulted in reductions of both AAOD and AOD, and (ii) that our simulations, which have been designed to account for these processes, reproduce
these reductions rather adequately.

## 3.2 Analysis of EnRs for the AAOD and AOD observations as a function of the BB aerosol photochemical age

The results of the application of our analysis method, which is described in Sect. 2.1, to the AAOD, AOD, and SSA data retrieved from the OMI and MODIS measurements are presented in Figure 5. Different panels show the nonlinear approximations (trends) of EnRs for AAOD and AOD (Figs. 5a and 5b) as a function of the BB aerosol photochemical age, as well
as similar nonlinear approximations for the SSA values that are either inferred from the OMI AAOD and MODIS AOD observations (Fig. 5c) or available directly from the OMAERUV data product (Fig. 5d). Along with the nonlinear trends for EnRs and SSA, we show the running averages over each corresponding 15 data points (for EnRs or SSA) that were preliminarily arranged with respect to the BB aerosol photochemical age. The averaging is done for illustrative purposes, as the scatter of the original data points is typically large at the scale of the trends. Note that in the special situation addressed in
this study, there is a strong association between the photochemical age of BB aerosol and the geographical location (specifically, the longitude) of the BB plumes transported westward (see Fig. 6). In particular, the BB aerosol photochemical age ranges from 6 to 42 hours in the source region and from 70 to 106 hours in the receptor region. This association facilitates the interpretation of the results of our analysis and also confirms the reliability of our model estimates of the BB aerosol photochemical age.

The nonlinear trend for AAOD (Figs. 5a) reveals a statistically significant decrease in $EnR_{abs}$ (up to ~45%) corresponding to the photochemical age ($t_e$) period from about 25 to 60 hours. The changes in $EnR_{abs}$ corresponding to both fresher and older BB aerosol are not statistically significant. This trend indicates that if AAOD were simulated for the given situation without taking into account any BB aging processes (except for coagulation) and were fitted to the AAOD observations corresponding to BB aerosol aged less than 40 h, the simulated AAOD corresponding to the aged BB plumes ($t_e > 60$ h) would be over-
estimated by ~30%. This rough estimate is in tentative agreement with the results from the two-region analysis discussed above (see Fig. 4), taking into account the ranges of the BB aerosol age in the source and the receptor region (see Fig. 6) and the uncertainty of the trends.

Variations in EnR for AOD ($EnR_{ext}$) are more pronounced (see Fig. 5b). Specifically, $EnR_{ext}$ increases by more than a factor of two during the initial 30 hours of the daytime evolution (since $t_e$ of about 6 hours) but then decreases by ~40%. This non-monotonic behavior is not contradictory to the relatively minor overestimation of AOD in the receptor region in our simulations for the 'bb_trc' scenario, taking into account that the major part of the increase of $EnR_{ext}$ occurs inside the source region, which includes BB aerosol with photochemical ages less than 40 h. Interestingly, the decreasing part of the trend in $EnR_{ext}$ almost coincides in time with the decreasing part of the trend in $EnR_{abs}$ (cf. Figs. 5a and 5b). This observation suggests that the reductions in both $EnR_{abs}$ and $EnR_{ext}$ may be driven by the same processes.

It is noteworthy that a similar increase (by a factor of two) in EnR for AOD as a result of atmospheric evolution of aerosol from Siberian fires was reported in our previous study (Konovalov et al., 2017a), despite the differences in the region, period and method of analysis compared to those in the present study. Furthermore, the previous analysis indicated the presence of a decreasing part in the dependence of $EnR_{ext}$ on the BB aerosol photochemical age, but the statistical significance of that feature was not evaluated. The quantitative difference between the BB aerosol photochemical ages corresponding to the $EnR_{ext}$ maximum (~30 h in this study and ~15 h in Konovalov et al., 2017a) may partly be due to the different definitions of the photochemical age. To the best of our knowledge, this is the first study reporting simultaneous decreases in both the AAOD and AOD (both corrected for dilution) in strongly aged BB plumes.

It seems reasonable to expect that the major increase in $EnR_{ext}$ in the absence of a similarly strong increase in $EnR_{abs}$ is likely to signify an increase in SSA. Figures 5c and 5d indicate that there is indeed a significant increase of SSA (at 388 nm) during the first 20-30 hours of the daytime evolution. According to the SSA estimates inferred from both the OMI AOD and MODIS AAOD data (Fig. 5c) under the assumption that the Ångström exponent is constant (see Sect. 2.2) and equal (in the given case) to 1.0, SSA increased with a decreasing rate from ~0.89 to almost 0.95 during the first 25 h and then continued to increase much more slowly, reaching a maximum after 67 h of the BB aerosol evolution. This assumption, however, may be too strong, because whatever processes might significantly affect AAOD and AOD will also likely affect the size distribution of the particles and the Ångström exponent. As an alternative, we also estimated variable values of the Ångström exponent for the 388 and 550 nm wavelengths by using the corresponding AOD from our simulations for the 'bb_vbs' scenario. The dependence of the derived SSA estimates on the photochemical age is found to be qualitatively similar to the dependence shown in Fig. 5c, although the amplitude of the SSA changes in the test case is smaller than in the base case (see Fig. S5). Importantly, a growing dependence on $t_e$ is also found for the direct SSA retrievals (see Fig. 5d). In this case, SSA increases from about 0.9 to almost 0.94, and the increase is statistically significant. These direct SSA retrievals are, however, very sparse, resulting in a large uncertainty of the derived trend, and have not been selected consistently with the AAOD and AOD data (see Sect. 2.5). Although the inferred SSA increase associated with the BB aerosol aging may look relatively small, it can be regarded as an indication of major changes in the optical properties of BB aerosol. For example, an increase of SSA from 0.91 to 0.95 under a constant value of AAOD would result in an enhancement of the aerosol scattering of the

solar radiation by almost a factor of two. Such enhancement can likely have serious implications for the radiative effects of BB aerosol.

Note that the freshest BB aerosol considered in our analysis has already been exposed to atmospheric processing for several hours. So, on the one hand, in the context of most BB aerosol aging experiments in smog chambers (e.g., Hennigan et al., 2011; Tiitta et al., 2016; Ciarelli et al., 2017) such an aerosol would be considered as already aged. On the other hand, these experiments do not usually examine changes in BB aerosol optical properties, especially at the long time scales addressed in this study. Therefore, a comparison of the results of our analysis with available results of laboratory experiments is not straightforward, and it goes beyond the scope of this study. However, the increase of SSA in our analysis is qualitatively consistent with the persistent enhancements of SSA of BB aerosol particles after 24 hours of atmospheric aging according to a recent analysis of AERONET data (Shi et al., 2019). Significant increases of the BB aerosol mass scattering efficiency and SSA as a result of the BB aerosol aging were observed by Kleinman et al. (2020) in near-field observations of BB plumes, consistent with an earlier observation (Akagi et al., 2012) of a major increase in aerosol light scattering in BB plumes. To the best of our knowledge, the simultaneous increase of both SSA and AOD (corrected for dilution) is reported here for the first time.

## 3.3 Interpretation of the inferred changes in the BB aerosol optical properties

In this section, we employ our simulations with the CHIMERE CTM to interpret the qualitative features of the BB aerosol evolution inferred from the satellite observations. It should be noted that our interpretation is not unambiguous, especially with respect to the quantitative aspects of the effects considered. However, to the best of our knowledge, this interpretation is the first attempt to reconcile the major changes in both absorption and scattering characteristics of BB aerosol due to multi-day atmospheric aging with the available knowledge on the atmospheric transformations of BB aerosol by using numerical simulations with a CTM. The analysis presented below may have implications for developing adequate and robust parameterizations of BB aerosol aging processes in chemistry transport and climate models.

Figure 7 shows the nonlinear approximations for the AAOD and AOD enhancement ratios calculated using the simulations for the 'bb_vbs" scenario. The corresponding approximations based on the analysis of the satellite data are also shown for comparison. Evidently, based on the criteria given by Eqs. (8) and (10), $EnR_{abs}$ and $EnR_{ext}$ from the simulations are overall consistent with their counterparts from the observations (see Fig. 5a, b). This fact gives credence to the representation of the BB aerosol evolution in our model.

Further insights into the processes leading to the nonlinear trends in $EnR_{ext}$ and $EnR_{abs}$ according to our simulations are provided by the results presented in Figs. 8-10. Specifically, Fig. 8 illustrates the evolution of the chemical composition of BB aerosol in the simulations for the 'bb_vbs' scenario, presenting the fractional contributions of the model species in the mass columnar concentration of BB aerosol (Fig. 8a) and the normalized EnRs for the columnar concentrations of organic species in the particulate phase (Fig. 8b). These EnRs were evaluated similarly to Eq. (1) as the normalized ratio of the columnar

concentration of the given component originating from fires to the columnar concentration of BB BC (which was considered as a passive tracer). Figure 8b also presents the evolution of EnR for OA in the sixth section (310-630 nm) of the particle size distribution assumed in our simulations. Figure 9 shows the evolution of the mass absorption and mass scattering efficiencies, and Fig. 10 demonstrates several sensitivity tests (explained below) aimed at a better understanding of the factors governing the evolution of $EnR_{abs}$. Additionally, several characteristics that can affect the gas-particle partitioning and oxidation processes and AAOD and AOD enhancement ratios are discussed in Sect. S4 (see also Fig. S6).

According to Fig. 8a, the BB aerosol composition is dominated by SOA species from the very beginning of the BB aerosol aging period addressed in our analysis (that is, from $t_e$ bigger than about 6 h). This is not quite surprising, as the typical lifetime of NTVOCs, which is the major source of SOA in our simulations, is just ~2 hours (with $k_{OH}$ of $4\times10$ cm$^3$ molec$^{-1}$ s$^{-1}$ and OH concentration of ~$4\times10^6$ cm$^{-3}$, see Fig. S6b). The same lifetime is characteristic of the gaseous fractions of POA (but note that the gaseous fraction of LV-POA is relatively very small) and HV-SOA. During the next 30 h or so, the remains of MV-POA and HV-SOA are evaporated and converted into LV-SOA and MV-SOA. At this stage, the SOA fraction remains nearly constant (~78 %), as the effects of evaporation and fragmentation are overall counterbalanced by the effects of condensation and functionalization. It is worth noticing, however, that while the MV-SOA fraction also remains nearly constant, the LV-SOA fraction increases by more than 20 %, and the ratio of the LV-SOA and LV-POA fractions increases quite significantly by about 45 %. After about 40 h, the main processes governing the evolution of SOA are the evaporation (facilitated by dilution, see Fig. S6c) of MV-SOA, its subsequent conversion into LV-SOA, and relatively slow evaporation and gas-phase fragmentation of the latter. Consequently, the MV-SOA fraction shrinks from 44 % to merely 5 %, whereas the LV-SOA fraction increases twofold. The contribution of the inorganic fraction to the BB aerosol is initially small (16 %) but it increases up to 37 % (see Fig. 8a) following the increase of the relative humidity (see Fig. S6a) from about 40 to 65 %. The predominant inorganic compound in particles is water, whose fraction remains below 15 % before 60 h and eventually rises to 26 %. Similar to non-absorbing organic compounds, inorganic ions and water increase the scattering cross-section of BB aerosol particles, thereby increasing AOD and SSA, although the contribution of a unit mass of water to the scattering efficiency is considerably smaller in our simulations than that of the organic matter because the real component of the refractive index for water (1.33) is substantially smaller than for the organic species (1.63). It is noteworthy that the substantial increase in the water uptake occurs only after 60 h, and therefore it could not contribute significantly to the strong increase in $EnR_{ext}$ before 30 h. The water uptake by the organic shell of the particles can also contribute to the lensing effect, thereby increasing AAOD, but as argued in this section below, the lensing effect does not play a key role in the evolution of $EnR_{abs}$.

Consistent with the results shown in Fig. 8a, the EnRs for both MV-POA and HV-SOA are rapidly decreasing and become negligible after about 60 h (see Fig. 8b). In contrast, the EnR for LV-SOA exhibits a growing tendency during the whole period of evolution, increasing by ~70 %, although becoming nearly stable after about 70 h. A growing tendency is also manifested in the evolution of the EnR for MV-SOA and total OA, but only during the initial 25 hours, after which the EnRs for both MV-SOA and total OA starts to decrease. Since the BB OA concentration is typically much higher than 1 μg m$^{-3}$

(see Fig. S6c), LV-POA evaporates slowly, with its EnR decreasing by 30 % during the whole analysis period. The evaporation of LV-POA can be mainly driven by the loss of its gas phase-fraction due to reaction (R2).

It may be puzzling why $EnR_{ext}$ is increasing twofold in the period until 30 h (see Fig. 7b), whereas no similar strong increase is demonstrated by the EnRs for any of the organic species. However, Fig. 8b also shows that the EnR for OA in particles with sizes from 310 to 630 nm increases more than three times during the same period when $EnR_{ext}$ increases twofold. According to Mie theory, this section corresponds to the particle diameters with the maximum scattering and extinction efficiencies at 550 nm. Hence, it is not surprising that the gain of mass by particles in this section of the accumulation mode results in an enhancement of AOD. Furthermore, Fig. 9 shows that the mass scattering efficiency (which is the main contributor to the mass extinction efficiency in our case) at the 550 nm wavelength increases by a comparable amount (almost by 70 %) in the same initial period of evolution. A similar, although relatively smaller, increase (~ 25 %) takes place in the mass scattering efficiency at 388 nm. In contrast, the mass absorption efficiency at 388 nm (also shown in Fig. 9) gradually decreases during the whole period of the evolution. Therefore, according to our simulation, the changes in the mass scattering and absorption efficiencies are the key drivers for the increases in both EnR for AOD at 550 nm and SSA at 388 nm. Our further analysis indicates that the changes in the mass absorption and scattering efficiencies are indeed associated with changes in the size distribution of BB particles. Specifically, the particle size distribution shifts toward larger particles (see Fig. S7) as the BB aerosol ages. Such a shift is apparently due to a complex interplay between the evaporation process (affecting predominantly smaller particles having a larger surface-to-volume ratio) and the dominating condensation process (which can significantly affect also bigger particles, especially in the situation when smaller particles are partially evaporated). In the diluting plumes, condensation of organic material onto the particles occurs primarily as a result of oxidation processes driving the SOA formation. The growth of particles can also be caused by the uptake of water by both organic and inorganic components of the aerosol.

To get further insights into the role of the SOA formation in the analyzed evolution of the optical properties of BB aerosol, we also performed simulations for a test scenario ('bb_poa') which has been briefly introduced in Sect. 2.3. In these simulations, the POA emissions had to be strongly (by a factor of 4.3) increased to keep the agreement between the observed and simulated AOD. The 'bb_poa' simulations were performed by assuming that POA is hydrophilic with κ of 0.2 (whereas it was assumed to be hydrophobic in the 'bb_vbs' scenario). The corresponding results are presented in Sect. S5 and Fig. S8. Importantly, we found that neither $EnR_{ext}$ nor the mass scattering efficiency considerably increase during the period until 30 h, in striking contrast to the behavior of the same characteristics in the base case simulation. Changes in $EnR_{ext}$, which are due to the combined effect of evaporation of MV-POA and the uptake of water, are small (less than 15%) and negative during the whole period of evolution. There are decreasing trends in the mass absorption efficiency and $EnR_{abs}$ and a gradual increase in SSA, which are apparently due to the limited lifetime of BrC in LV-POA. Overall, the simulations for the 'bb_poa' scenario confirm that the oxidation processes are the main driving force behind the major features in the retrieved evolution of AOD and SSA.

As could be expected, the simulated evolution of $EnR_{abs}$ is also found to be driven by condensation and evaporation processes, but the impact of these processes on $EnR_{abs}$ is mediated by several additional factors. In particular, one should take into account that AAOD is determined by BC, POA, and SOA species, each of which is characterized by a different value of the imaginary refractive index, and therefore equal relative changes in the mass concentrations of these components will have different impacts on AAOD. Furthermore, changes in the POA and SOA mass concentrations can affect AAOD through the lensing effect (Lack et al., 2010). To examine these factors, we ran the OPTSIM module for several limiting cases representing different optical properties of the individual components of BB aerosol simulated within the 'bb_vbs' scenario. The main results are presented in Fig. 10, where the evolution of $EnR_{abs}$ simulated under the different assumptions is compared with the corresponding trends derived from the OMI observations. Based on these results, we also evaluated the relative contribution of several factors to the BB aerosol absorption (see Sect. S6 and Fig. S9).

Specifically, Fig. 10a shows the trend in $EnR_{abs}$ for the sensitivity test where the refractive index of the shell of BB particles approaches that of ambient air. This is a relatively trivial case because the size distribution of BC particles should not be affected by evaporation and condensation of the POA and SOA species, and therefore AAOD associated with such particles is expected to behave almost as an inert tracer. And indeed, $EnR_{abs}$ computed for this case is found to be nearly constant (Fig. 10a). A minor increase in the $EnR_{abs}$ is likely due to different effects of dry deposition and coagulation on the size distribution of BB aerosol particles in the 'bb_vbs' and 'bb_trc' scenarios. Despite looking trivial, this test confirms the integrity of our simulations and indicates that the evolution of AAOD in our simulations is almost fully determined by transformations of the organic fraction of BB aerosol. Note that in this case, the average AAOD in the selected dataset is much lower than in the base case: we estimated that "pure" BC (without the lensing effect) accounts on average for only 31% of the total absorption (see Sect. S6).

Figure 10b presents results for the opposite sensitivity test, where the contribution of BC to the BB aerosol absorption was disregarded. In other words, AAOD was computed under the assumption that BC is not absorbing. In this case, the evolution of $EnR_{abs}$ is similar to that in the base case (see Fig. 7a), except that the amplitude of a variation of the nonlinear approximation is much larger in the test case. This test case further confirms a pivotal role played by OA in the simulated evolution of AAOD. As could be expected, constant BC absorption (see Fig. 10a) dampens, to a significant extent, the AAOD changes caused by the OA.

In principle, one would expect also that variations of the thickness of the organic shell, which are associated with the evolution of $EnR_{ext}$, lead to changes in $EnR_{abs}$ due to the lensing effect associated with the non-absorbing fraction of OA. To test this possibility, we set the imaginary refractive index for both POA and SOA to zero. The results of the corresponding computations are shown in Fig. 10c. According to these results, the lensing effect does not significantly contribute to the $EnR_{abs}$ variations in the base case. This can be due to the saturation of the absorption enhancement by a clear coating (Wu et al., 2018) and also because the dilution-corrected variations of OA concentration in our simulations are relatively small (see Fig. 8b).

The test computations shown in Figs. 10a-10c clearly indicate that the key factor responsible for the changes of $EnR_{abs}$ in our simulation is the variable BrC absorption. Further insights into the mechanism of these changes are provided by a computation under the assumption that SOA is non-absorbing (while POA is still absorbing). The results of these computations are shown in Fig. 10d. Similar to the base case, $EnR_{abs}$ is decreasing with $t_e$, but at a much lower rate. Hence, we can conclude that the evolution of the aerosol absorption properties in our simulations is mainly driven by the formation and transformations of the SOA species.

Taking the entirety of the results of our analysis into account, we suggest the following qualitative interpretation of the main processes driving the evolution of $EnR_{abs}$ and $EnR_{ext}$ in our simulations and, ultimately, in the real atmosphere. First, fast oxidation of NTVOCs results in the production of HV-SOA and MV-SOA. HV-SOA is mostly oxidized into MV-SOA and then LV-SOA during the initial period of a few hours, which is not represented in the satellite observations considered here. Oxidation of POA provides an additional source of both MV-SOA and LV-SOA. Evaporation (and subsequent transformation) of POA and HV-SOA and formation and condensation of LV-SOA and MV-SOA leads to a major increase of $EnR_{ext}$, mainly as a result of an increase in the mass scattering efficiency at 550 nm due to a shift of the size distribution towards bigger particles. In contrast, the mass absorption efficiency at 388 nm is not strongly affected by the indicated processes, and there is only a minor increase in $EnR_{abs}$, which is caused by the formation of MV-SOA (as LV-SOA is assumed to be non-absorbing). As the sources of MV-SOA are depleted, its concentration decreases due to the oxidation reaction (R5). Since the fragmentation pathway is assumed to be dominating in this reaction, the depletion of MV-SOA is not compensated by the production of LV-SOA; so the total concentration of the SOA species also decreases. This process can explain the decreasing stage in both the $EnR_{abs}$ and $EnR_{ext}$ evolution. The decrease of the total SOA concentration – and the decrease of $EnR_{abs}$ and $EnR_{ext}$ – slows down as the MV-SOA concentration decreases. Eventually, the BB aerosol consists predominantly of the lower volatility fractions of POA and SOA. This agrees with observations of low volatility of aged BB aerosols in field campaigns (Clarke and Kapustin, 2010; Thornberry et al., 2010; Andreae et al., 2018).

**3.4 Discussion of the uncertainties**

In this section, we discuss possible sources of systematic errors in our retrieval and interpretation of the evolution of the optical properties of BB aerosol, which are associated both with our analysis of satellite data and CTM simulations. Random uncertainties (due to the spatial and temporal variability of satellite and model data corresponding to the same photochemical age of BB aerosol) in the retrieved tendencies are taken into account in the confidence intervals as explained in Sect. 2.1.

To examine the main potential sources of systematic errors in our analysis of the satellite data, we performed several sensitivity tests. These tests and the corresponding results are presented in the Supplementary Material, Sect. S7 and Figs. S10-S12.

First, we tried to ensure that the major features in the retrieved evolution of the BB aerosol absorption properties are not sensitive to possible biases in the simulated vertical profiles of BB aerosol, having in mind that such biases can affect the

processing of the OMAERUV retrievals which are provided for five different altitudes of the aerosol center of mass (see Sect. 2.2). For this reason, we repeated our analysis with the 'final' OMAERUV retrievals, that is, without involving the simulated heights of the BB plumes. We found that the trends in $EnR_{abs}$ and SSA, which were obtained with the 'final' retrievals (see Fig. S10a, b) are only slightly different from those for the base case (see Fig. 5a, d, respectively).

Second, we examined to what extent our analysis is sensitive to the satellite and modeled data for the CO columns involved in the estimation of the correction factor $f_c$ (see Eqs. 2 and 3) intended to compensate for possible model transport errors but possibly having biases of its own (see Sect. S7). To this end, we repeated our analysis of the satellite data with $f_c$ equal to 1. The derived trends (see Fig. S11) retained the main qualitative features of the tendencies obtained for the base case (see Fig. 5a, b). It is noteworthy that the tendency in EnR for AOD has wider confidence intervals corresponding to photochemi-

cal ages larger than 70 h in the test case than in the base case, thereby indicating that the application of the CO columns is indeed helpful in reducing the uncertainties associated with model transport and BB emission errors in the simulations of the strongly aged BB plumes. This sensitivity test, therefore, provides strong evidence that the main derived qualitative features of the evolution of the BB aerosol optical characteristics are not an artifact of the model errors in the transport and BB emissions.

Third, we evaluated the sensitivity of the inferred tendencies in $EnR_{ext}$ and $EnR_{abs}$ to the assumptions regarding the background AOD and AAOD values (see Eq. 1) by neglecting the background AOD and assuming a constant value of 0.018 (based on the daily Level-3 OMAERUV data) for the background AAOD. This test indicated (see Fig. S12) that neither the increasing part in the trend in $EnR_{ext}$ nor the decreasing parts in the trends in both $EnR_{ext}$ and $EnR_{abs}$ is due to possible biases in the background AOD and AAOD values. Overall, these tests confirm that our major qualitative findings obtained as a

result of the analysis of the satellite data are not artifacts of the analysis procedure or model errors.

Potential systematic errors in the inferred tendencies in $EnR_{ext}$ and $EnR_{abs}$ can also be associated with biases in the satellite retrievals. However, as noted above (see Sect. 2.2), previous validation studies and analyses involving both the MODIS AOD and OMI AAOD retrievals for Siberian BB plumes did not reveal indications that these data are significantly biased. Accordingly, we have reason to assume that the corresponding systematic errors in the inferred tendencies are relatively

small in comparison to random errors and are covered by the estimated confidence intervals. Nonetheless, the lack of in situ measurements in Siberia does not allow us to properly evaluate and definitely rule out any effects of potential biases in the satellite retrievals on the results of our analysis. Validation of the satellite retrievals, especially ones distinguishing between fresh and aged BB aerosol, requires further dedicated studies.

As pointed out in the introduction and Sect. 2.4, our interpretation of the inferred evolution of the BB aerosol optical proper-

ties is based on the use of a highly simplified representation of the evolution of the organic fraction of BB aerosol. Accordingly, our analysis of the simulated data (see Sect. 3.3) was intended to identify only basic possible mechanisms underlying the main qualitative feature of the inferred tendencies and does not allow us to make any quantitative assessments. Although – based on our knowledge – we cannot propose any plausible alternative to oxidation processes associated with SOA for-

mation and transformation as a driving force behind the major changes in the enhancement ratios for both AOD and AAOD, our interpretation involves considerable uncertainties with respect to many important features of these processes.

For example, the evolution of the mass scattering efficiency ($\alpha_s$) is likely to depend on the initial size distribution of the BB aerosol particles, which may be very variable (Reid et al., 2005a). A smaller increase in $\alpha_s$ would need to be compensated by a larger enhancement of the OA concentration (and vice versa). This could be achieved in our simulations by assuming a larger weight for the functionalization (or, alternatively, fragmentation) pathway of the oxidation reactions. The evolution of POA and SOA can also depend on the phase state and viscosity of the particles. While we assume in our simulations that particles are liquid and inviscid, the critical temperature for the transition between the liquid and glassy states of the particles is estimated to be in the range of ambient temperatures in the boundary layer over Siberia (Shiraiwa et al., 2017), being a complex function of ambient relative humidity and particle composition. Slow diffusion of organic molecules within highly viscous glassy or semi-solid particles could limit gas-particle interactions, thereby slowing down the POA and SOA evaporation (e.g., Kim et al., 2019; Vaden et al., 2011) but also effectively exposing larger amounts of SOA species to gas-phase photochemical processing. The gas-particle partitioning and evolution of organic species forming BB aerosol can also depend on the mixing state of POA and SOA species. While there is experimental evidence supporting our simplifying assumption that POA and SOA species form a well-mixed solution (Asa-Awuku et al., 2009), there is also evidence (Song et al., 2007) that POA and SOA form external mixtures. If POA and SOA species do not mix in the real BB aerosol and form two separate organic phases (as assumed, e.g., in Shrivastava et al., 2015), then our model is likely prone to overestimate the concentrations of both POA and SOA species in particles. Taking the phase state transition processes, particle viscosity and possible scenarios concerning mixing of organic solutions into account would result in significant extensions of our simplified representation of BB aerosol aging. These extensions, which cannot be sufficiently constrained by available observations in Siberia, would probably require changes in the optimal values of the adjustable parameters and some other modifications (e.g., changes in the assumed volatility distribution or enthalpies of evaporation). However, we see no evidence in the literature that a more complex representation of BB aerosol could produce qualitative changes in the simulated dynamics of BB aerosol, and thus an application of such a representation would not likely invalidate the proposed qualitative interpretation of the "observed" changes in the BB aerosol optical properties.

One more potentially important factor that can affect the BB aerosol evolution in the real atmosphere but is not taken into account in our simulations is the photolysis of SOA species, especially in particles. Based on a very limited amount of experimental data on the quantum yields for photolytic reactions of organic species in the particle phase, Hodzic et al. (2015) estimated the in-particle photolysis rate of SOA as 0.04 % of the photolysis rate of $NO_2$ and found – as a result of a global model simulation – that the SOA photolysis can remove aerosol from troposphere on time scales of several days. Although this mechanism of SOA removal is very uncertain, we cannot rule out that it contributed significantly to the SOA depletion in the analyzed situation. If that was the case, then the SOA fragmentation rate in our simulations is likely overestimated. However, it should be noted that if the in-particle SOA photolysis were a primary driver for the decrease of AOD in our

analysis (see Fig. 5b), then the rate of the decrease in $EnR_{ext}$ would not be expected to strongly decelerate after about 70 h, as it does according to Fig. 5b (presuming that SOA and linked water are actually predominant components of aged BB aerosol). Hence, the brief inspection of our results suggests that the in-particle SOA photolysis could hardly be a principal cause for the decreasing part of the trend in $EnR_{ext}$.

The evolution of the imaginary refractive index of BB aerosol is also represented in our simulations in a very simplified way. Although we differentiate between the several model species with respect to these properties, both primary and secondary real organic compounds are likely to feature a much wider spectrum of absorptive properties. Nonetheless, our interpretation will likely hold, if the low-volatility SOA species formed in the real atmosphere from VOCs and POA as a result of a long chain of oxidation processes are much less absorptive than relatively fresh (and more volatile) SOA species. This assumption

is supported by field observations (Forrister et al., 2015; Selimovic et al., 2019) and numerous laboratory experiments (e.g., Browne et al., 2019; Fan et al., 2019, Wong et al., 2017; 2019) indicating eventual bleaching of BrC in BB aerosol as a result of its exposure to the atmospheric oxidation processes and UV radiation. One more important simplifying assumption involved in our simulations concerns the evolution of the optical properties of the SOA species: as noted above, we assume that the imaginary refractive index for a given "virtual" SOA species, which represents the molecular weight and volatility of

its real counterparts, does not change as long as this species exists. This assumption is not true in a general case, because the destruction of chromophores by UV irradiation or as aqueous heterogeneous oxidation is not necessarily associated with significant changes in the volatility or molecular weight of the affected compound. However, on the one hand, there is experimental evidence noted above (Fleming et al., 2020) that direct photodegradation of BrC in BB aerosol is a slow process with a characteristic time scale of at least several days – even though this observation, strictly speaking, applies only to fresh

primary BB aerosol. On the other hand, aqueous heterogeneous oxidation can hardly be a significant process in our situation, where water uptake by particles – as discussed above – is typically small, although the effect of ambient relative humidity on the atmospheric evolution of BrC in BB aerosol has yet to be investigated.

Overall, this discussion suggests that although our model representation of BB aerosol evolution involves strong assumptions (which yet need to be verified in future research), our qualitative interpretation of the inferred major changes of the optical

properties of BB aerosol in Siberia is sufficiently robust and realistic. In turn, the applicability of this interpretation corroborates the reliability of our major findings from the analysis of the satellite observations.

## 4 Conclusions

We have presented an analytical framework designed to advance the knowledge of changes in the optical properties of BB aerosol due to its atmospheric aging by using retrievals of AAOD, AOD, and SSA from satellite observations of BB plumes.

This framework includes a method to evaluate nonlinear trends in the optical properties of BB aerosol due to the oxidation and gas-particle partitioning processes at the temporal scale (typically, several days) associated with long-range transport of

BB plumes. It also involves using adjustable simulations of the sources and atmospheric evolution of BB aerosol with a chemistry transport model to reproduce and interpret the inferred trends.

We used this framework to get insights into the evolution of BB aerosol optical properties during a pronounced episode of a large-scale outflow of BB smoke plumes from Siberia towards Europe that occurred in July 2016. The analysis was based on
the use of OMI AAOD and SSA retrievals combined with MODIS AOD retrievals. These retrievals were used together with BB aerosol simulations and the CO columns derived from IASI observations to evaluate the enhancement ratios (EnR) for AAOD and AOD, which allows isolating the effects of oxidation and gas-particle partitioning processes from those of other processes, including transport, deposition, and wet scavenging. The simulations were performed with the CHIMERE chemistry transport model combined with the OPTSIM module enabling evaluation of the BB aerosol optical properties under the
assumption of a core-shell morphology of the particles. The OA oxidation and gas-particle partitioning processes were represented in the simulation in the framework of a highly simplified but adjustable VBS scheme. Importantly, the evolution of BrC was simulated consistently with the evolution of the POA and SOA species, based on the assumptions that the POA species are much more absorptive in the near-UV wavelength range than the SOA species and that the imaginary refractive index for a given SOA "virtual" species is constant. The EnR estimates and SSA data were analyzed as a function of model-
based estimates of the BB aerosol photochemical age, which – in the situation considered – is found to be strongly associated with the geographical location of a BB plume.

We found that, while the EnR for AAOD does not change significantly during the first 20-30 hours of daytime evolution, the EnR for AOD strongly (by more than a factor of 2) increases during the same period. The increase in EnR for AOD is accompanied by a statistically significant increase in SSA. Note that the first few hours of the atmospheric evolution of BB
aerosol are not covered by our analysis. Further atmospheric processing of BB aerosol (up to a photochemical age of about 100 h) is found to be associated with statistically significant decreases (of about 45 %) of EnRs for both AAOD and AOD and insignificant changes of SSA. Note that while a similar increase in EnR for AOD was reported previously (Konovalov et al., 2017a), our findings concerning the simultaneous changes in both the absorption and extinction characteristics of BB aerosol are novel.

By adjusting the imaginary refractive indexes for the SOA species, a few parameters of the simplified VBS scheme, and the emission factors, our simulations were brought into close agreement with the observed AAOD and AOD values and with the trends in EnRs for both AAOD and AOD. The analysis of our simulation results suggests that the upward trend in EnR for AOD (which is associated with the initial increase in SSA) could be due to SOA formation and POA evaporation leading to a major increase in the mass scattering efficiency of BB aerosol through modulation of the particle size distribution. Evapo-
ration of the semi-volatile SOA is indicated as a likely reason for the subsequent decrease of the EnR for both AAOD and AOD. We suggest that despite a highly simplified character of our representation of BB aerosol our simulations, our conclusions regarding the basic factors behind the "observed" BB aerosol evolution are sufficiently robust. These factors need to be adequately taken into account in chemistry transport and climate models to ensure reliable simulations of the BB aerosol

optical properties and accurate estimates of associated radiative effects, specifically in Northern Eurasia. The proposed simplified representation of BB aerosol evolution can contribute to achieving such a goal, suggesting a reasonable compromise between over-complexity unconstrained by available observations and over-simplicity leading to major biases in the simulation results.

Overall, this study has demonstrated that the presented analytical framework can help identify and interpret manifestations of the BB aerosol aging processes far beyond the time scales that can currently be addressed in aerosol chamber experiments. Although the application of the framework in the present study has been limited to a concrete episode of long-range transport of BB plumes, the proposed methods are sufficiently general and can be used in many other studies of BB aerosol and applied to different sets of satellite data, including, for example, multi-spectral retrievals of aerosol absorptive properties from

MISR (Multi-Angle Imaging Spectrometer) measurements (Junghenn Noyes et al., 2020). The framework was also shown to enable validation and optimization of model representations of BB aerosol evolution. Therefore, its future applications can help address current challenges associated with the representation of the evolution and optical properties of BB aerosol and its components in regional and global models (Shrivastava et al., 2017; Samset et al., 2018; Tsigaridis and Kanakidou, 2018; Konovalov et al., 2019).

***Data availability.*** The OMAERUV data product (Torres, 2006), the MOD04/MYD04 (Levy and Hsu, 2015) and MYD14/MOD14 datasets (Giglio and Justice, 2015a, b) are available through the NASA Earth Data Search (https://searchearthdata.nasa.gov/, last access: 19 April 2020). The CO column amounts retrieved from the IASI measurements (Clerbaux et al., 2009) are from the ESPRI data center (http://cds-espri.ipsl.fr/etherTypo/index.php?id=1707&L=1, last access: 19 April 2020). The CHIMERE chemistry transport model (CHIMERE-2017) is available at

http://www.lmd.polytechnique.fr/chimere/, last access: 20 April 2020, and the OPTSIM software is available at https://www.lmd.polytechnique.fr/optsim/, last access: 20 April 2020.

***Competing interests.*** The authors declare that they have no conflict of interest.

***Author contributions.*** IBK and MB designed the study. IBK also designed the method to analyze satellite observations, contributed to the analysis of satellite and model data, and prepared the manuscript. MB also contributed to the discussion of

the results and to the preparation of the manuscript. NAG contributed to the analysis of satellite observations and conducted part of the numerical experiments. MOA contributed to the discussion of the results and the preparation of the manuscript.

***Financial support.*** The analysis and simulations of the BB aerosol evolution were supported by the Russian Science Foundation (grant agreement no. № 19-77-20109). The development and validation of the simplified VBS scheme were performed with support from the Russian Foundation for Basic Research (grant no. 18-05-00911).

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

**Table 1.** Parameters of the species representing the evolution of BB OA in the framework of the simplified VBS scheme

| Species name | Volatility, $C^*$ ($\mu g\ m^{-3}$) | Molar mass | Enthalpy of evaporation, $\Delta H_{vap}$ ($kJ\ mol^{-1}$) |
|---|---|---|---|
| LV-POA | $10^0$ | 216 | 85 |
| MV-POA | $10^2$ | 216 | 77 |
| LV-SOA | $10^0$ | 144 | 85 |
| MV-SOA | $10^2$ | 135 | 77 |
| HV-SOA | $10^3$ | 131 | 73 |
| NTVOCs | $\infty$ | 113 | - |

**Table 2.** Reactions representing the evolution of BB OA in the simplified VBS scheme and their OH reaction rates ($k_{OH}$)

| | Reaction | $k_{OH}$ ($cm^3\ molec^{-1}\ s^{-1}$) |
|---|---|---|
| (R1) | MV-POA + OH → 1.3 LV-SOA + OH | $4\times10^{-11}$ |
| (R2) | LV-POA + OH → 1.3 LV-SOA + OH | $4\times10^{-11}$ |
| (R3) | NTVOC + OH → $\xi_1^{(a)}$ 0.33 HV-SOA + (1-$\xi_1$) 0.30 LV-SOA + OH | $4\times10^{-11}$ |
| (R4) | HV-SOA + OH → MV-SOA + OH | $4\times10^{-11}$ |
| (R5) | MV-SOA+ OH → $\xi_2^{(a)}$ LV-SOA + OH | $1\times10^{-11}$ |
| (R6) | LV-SOA+ OH → $\xi_2$ LV-SOA + OH | $1\times10^{-11}$ |

[a] $\xi_1$ and $\xi_2$ are adjustable parameters, which have been evaluated as: $\xi_1$=0.85, $\xi_2$=0.15

**Table 3.** Key features of the configuration of the simulations with the CHIMERE model

| Simulation scenario | Key features |
|---|---|
| 'bb_vbs' | - POA and SOA processing is represented using a simplified 6-component VBS scheme (see Tables 1 and 2) with two adjustable parameters<br>- Gas-particle partitioning of the POA and SOA species is calculated assuming these species to form an ideal inviscid liquid solution;<br>- BB POA and BC emissions are derived from the FRP measurements and constrained by the AOD and AAOD satellite observations with the injection heights parameterized[a] as a function of FRP; the mass emission ratio of NTVOCs to POAs is an adjustable parameter estimated as 14.2;<br>- Spherical core/shell structure of particles, with BC forming the core and the other (homogeneously mixed) components forming the shell, is assumed to calculate the BB aerosol optical properties using the OPTSIM module[b]<br>- Light absorption by LV-POA, HV-SOA, and MV-SOA with $k_{poa}$[c] $=0.03\times\exp(-t_e/82)$ (where $t_e$ is the BB aerosol photochemical age [hours]) and $k_{soa}$[c] estimated as $9\times10^{-3}$ is taken into account, while MV-POA and LV-SOA are treated as non-absorbing species<br>- Uptake of water by SOA is computed[d] assuming the hygroscopicity parameter $\kappa_{org}$ of 0.2, whereas BC and POA are assumed to be hydrophobic<br>- Photochemical age of BB aerosol is estimated using two model tracers, one of which is inert and another is decaying with a constant rate<br>- Biogenic and anthropogenic emissions of aerosols and gases are turned off<br>- Zero concentration are taken for the boundary conditions |
| 'bb_trc' | - SOA formation is turned off and the organic fraction of BB aerosol is treated as being chemically inert, refractory, non-absorbing, and hydrophobic<br>- Other features and assumptions are the same as in the 'bb_vbs' scenario |
| 'bgr' | - BB emissions are turned off but both the biogenic and anthropogenic emissions of aerosols and gases are taken into account using the HTAP[e] and MEGAN[f] emission inventories, respectively<br>- SOA formation from both the anthropogenic and biogenic precursors is taken into account using simplified representations[g] provided in the standard version of CHIMERE<br>- AOD is calculated with the in-built FAST-JX module[h] of CHIMERE with the standard settings[i]<br>- Boundary conditions for aerosol and gaseous components are specified using the monthly climatology from the LMDZ-INCA global model[j] |
| 'bb_poa' (a test scenario) | - SOA formation is turned off but POA is treated in the same way as in the 'bb_vbs' scenario except that POA is assumed to be hydrophilic ($\kappa_{org}$ =0.2)<br>- Other features and assumptions are the same as in the 'bb_vbs' scenario |
| All scenarios | - Simulations are performed on a $1°\times 1°$ model grid with 12 non-equidistant layers extending up to 200 hPa pressure level<br>- Gas-phase chemical processes are simulated using the MELCHIOR2 chemical mechanism (in-build into CHIMERE)<br>- Dry deposition of gases, wet scavenging of aerosol and gases are taken into account using parameterizations implemented in the standard version of CHIMERE[i]<br>- Meteorological fields are computed "off-line" with the WRF (version 3.9) model[k] |

[a] according to Sofiev et al. (2012); [b] Stromatas et al. (2012); [c] $k_{poa}$ and $k_{soa}$ are the imaginary parts of the refractive indexes (at 388 nm) for the POA and SOA species, respectively; [d] according to Petter and Kreidenweis et al. (2007); [e] Janssens-Maenhout et al. (2015); [f] Guenther et al. (2006); [g] Kroll et al. (2006), Zhang et al. (2007), Bessagnet et al. (2008); [h] Wild et al., (2000); [i] Mailler et al. (2017); [j] Folberth et al. (2006); [k] Skamarock et al., (2008)

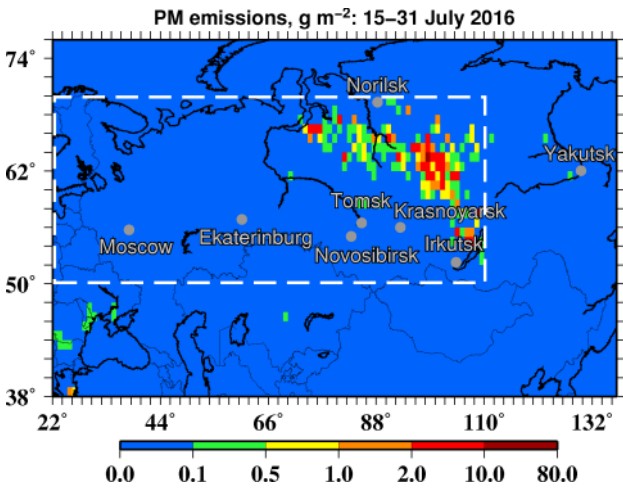

**Figure 1.** Spatial distribution of the total emissions (g m$^{-2}$) of particulate matter from fires in the period from 15 to 31 July 2016. The emissions were computed using the MODIS FRP data (see Sect. 2.3) and are shown over the CHIMERE domain specified in this study. A dashed white rectangle indicates the study region.

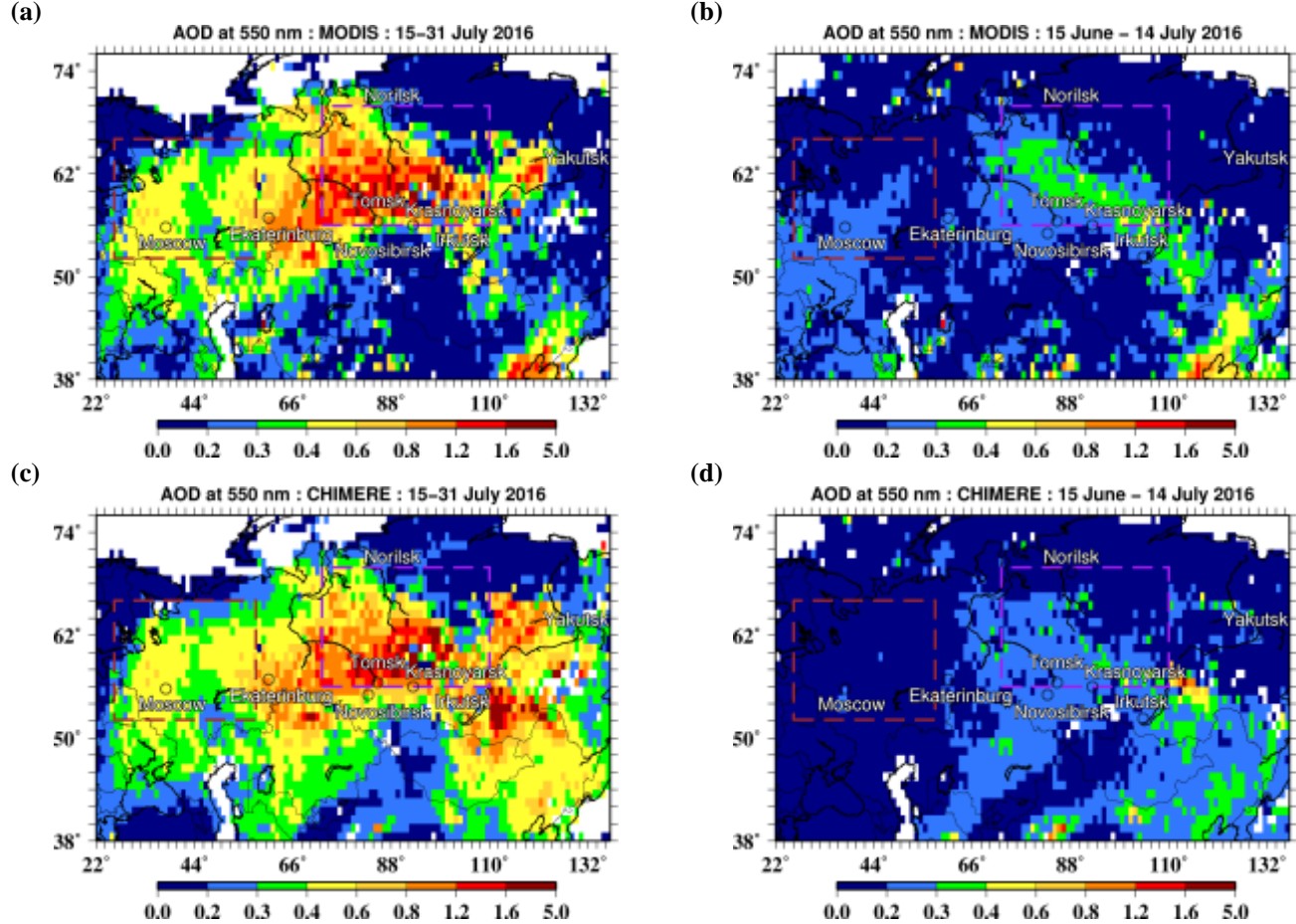

**Figure 2.** Spatial distributions of the temporal averages of AOD (at 550 nm) in the periods (a, c) from 15 to 31 July 2016 and (b, d) from 15 June to 14 July 2016 according to (a, b) the MODIS observations and (c, d) the combined CHIMERE simulations for the 'bb_vbs' and 'bgr' scenarios. The distributions represent the ancillary sets of AOD data that were selected irrespective of the availability of the corresponding AAOD retrievals. The rectangles depict the "source" (purple lines) and "receptor" (dark red lines) regions covering parts of Siberia and (mostly) the European territory of Russia, respectively.

**(a)**                                                    **(b)**

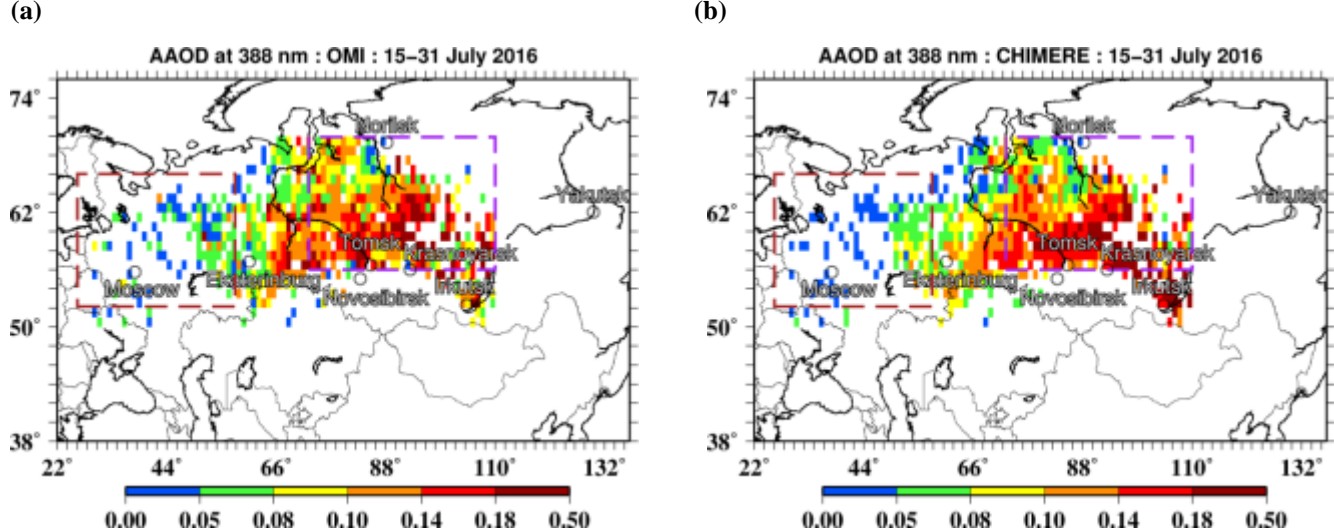

**Figure 3.** Spatial distributions of temporal averages of AAOD (at 388 nm) in the period from 15 to 31 July 2016 according to (a) the OMI observations and (b) CHIMERE simulations for the 'bb_vbs' scenario. The AAOD data represent only BB aerosol in the study region according to the selection criterion specified in the OMAERUV data product. The rectangles depict the source and receptor regions as in Fig. 2.

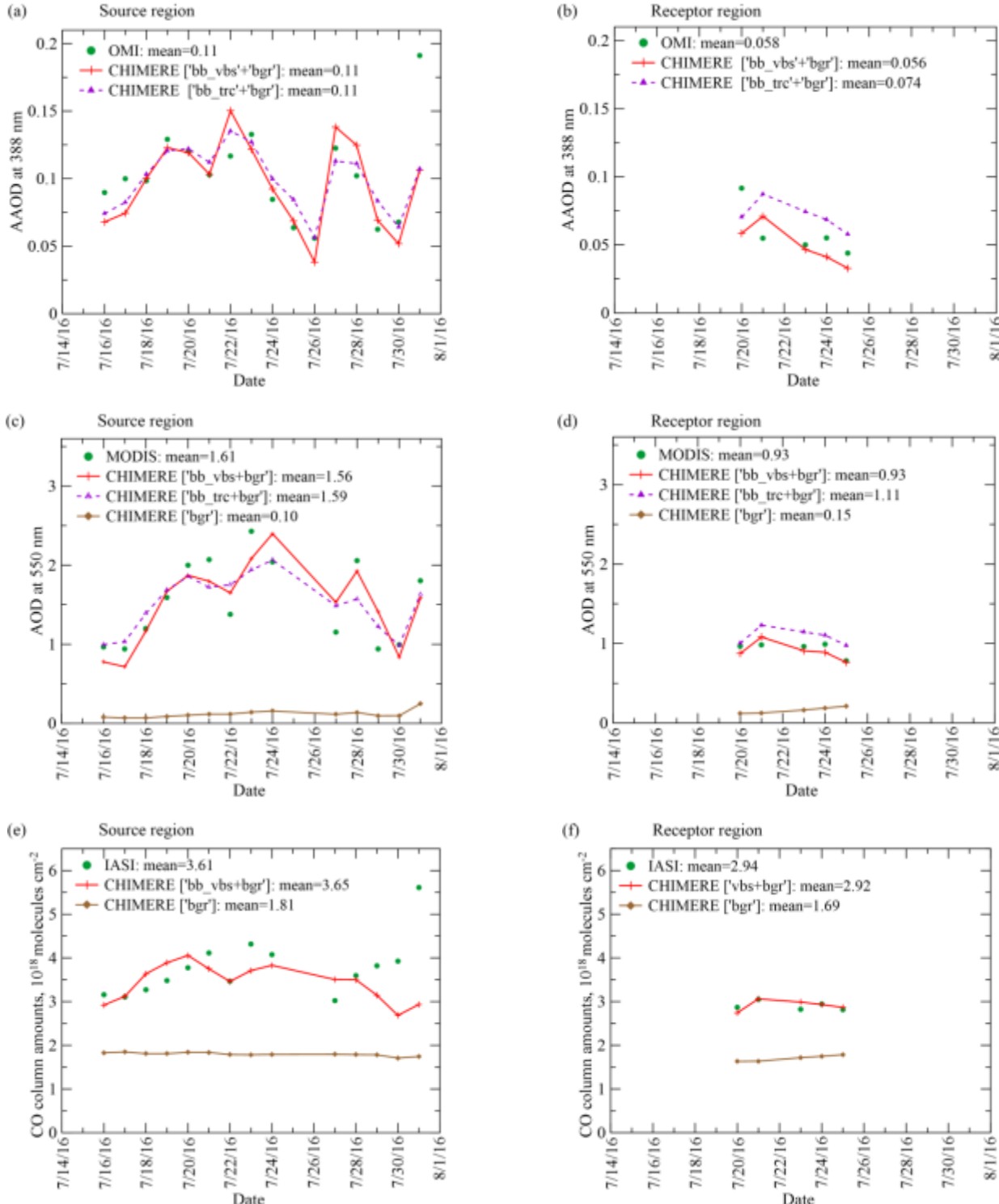

**Figure 4.** Time series of the daily (a, b) AAOD, (c, d) AOD, and (e, f) CO values averaged over the study region according to the satellite

observations and the CHIMERE simulations under the different model scenarios for the (a, c, e) "source" and (b, d, f) "receptor" regions indicated in Fig. 2. All the data were selected consistently. Note that the simulations for the 'bgr' (background) scenario are shown after applying the de-biasing procedure (see Sect. 2.5). The background AAOD is not evaluated in this study and therefore not shown. Taking into account that the time series are very short, the correlation coefficient has not been evaluated (as its values are not sufficiently robust).

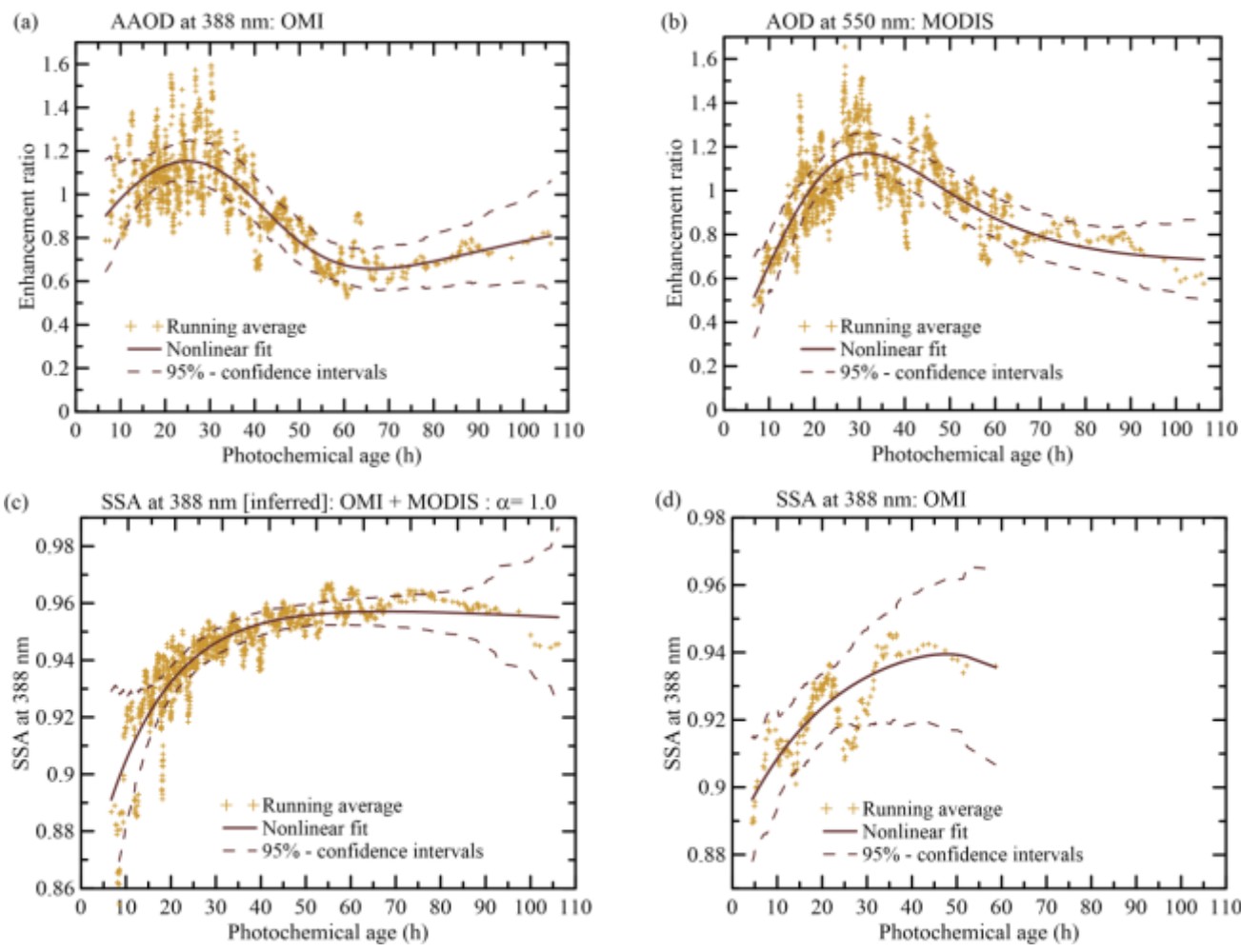

**Figure 5.** Nonlinear approximations (see Eq. 5) of the dependencies of (a, b) EnRs for AAOD (388 nm) and AOD (550 nm) and (c, d) SSA (388 nm) on the photochemical age of BB aerosol. Also shown are the running averages over each consecutive 15 data points (for EnRs or SSA) arranged with respect to the photochemical age as well as the 95 % confidence intervals for the approximations. The SSA values approximated in panel (c) are inferred from the OMI AAOD and MODIS AOD observations, while those presented in panel (d) are provided directly in the OMAERUV data product.

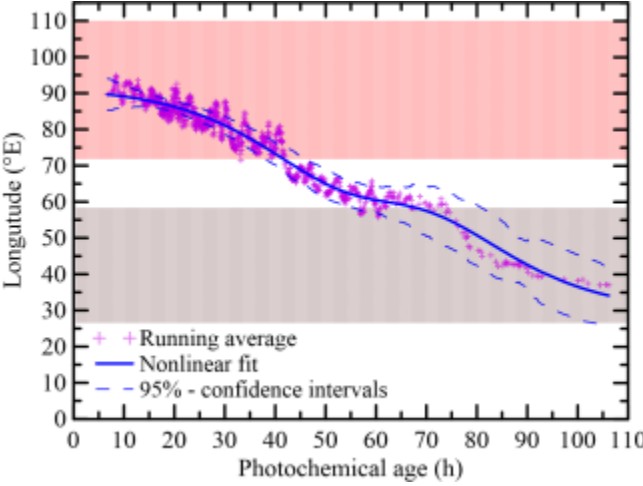

**Figure 6.** Dependence of the longitude of the instant location of a given BB plume on its photochemical age. The shaded areas indicate the ranges of longitudes of the source (red shade) and receptor (brown shade) regions shown in Fig. 2.

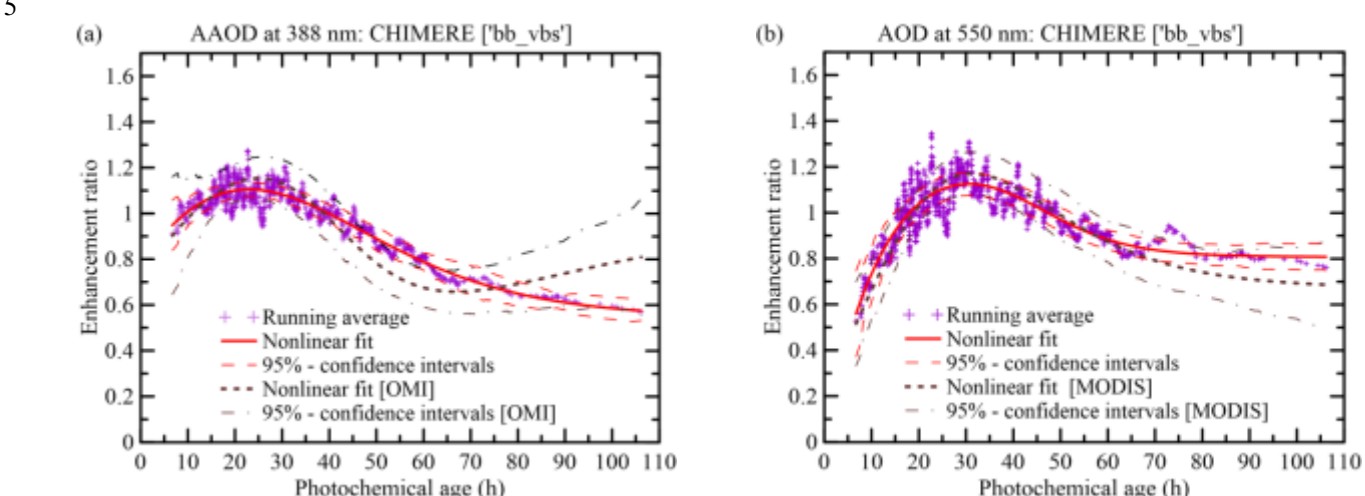

**Figure 7.** The same as in Figs. 5a, b but obtained using the CHIMERE simulations for the 'bb_vbs' scenario. The dependencies from Figs. 5a, b, and their confidence intervals are also shown for comparison.

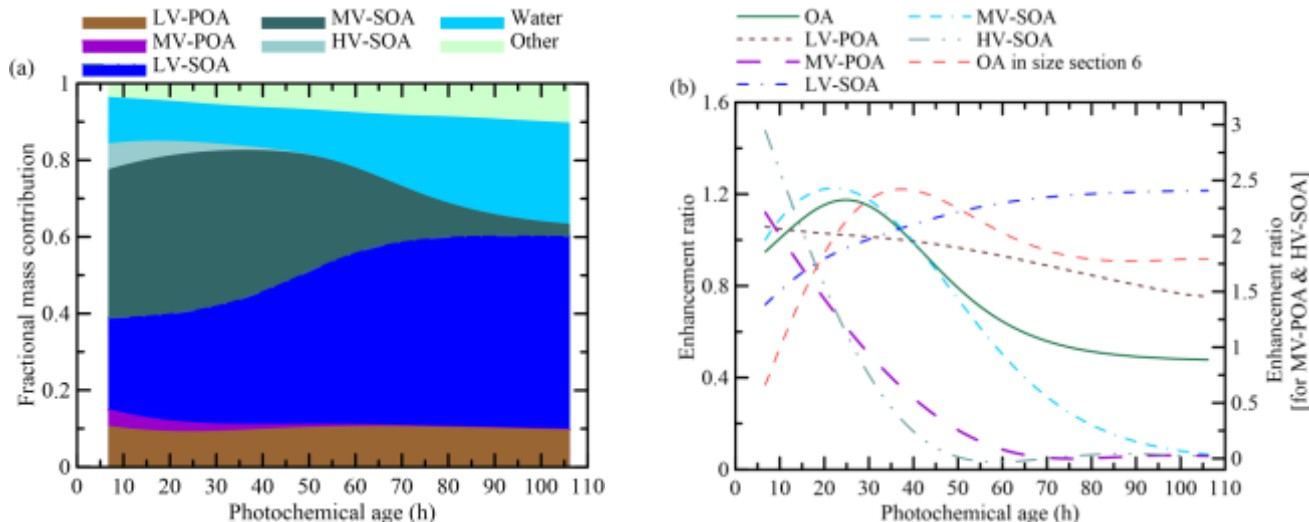

**Figure 8.** Evolution of the key model species determining the composition of BB aerosol in the CHIMERE simulation for the 'bb_vbs' scenario: (a) fractional contributions of the model species in the mass columnar concentration of BB aerosol, (b) the normalized EnRs of columnar concentrations of organic species in the particulate phase in the simulations for the 'bb_vbs' scenario with respect to OA concentration in the simulations for the 'bb_trc' scenario, along with similar enhancement ratios for the total columnar OA concentration and OA concentration in the 6$^{th}$ bin (310-630 nm) of the particle size distribution in CHIMERE. Note that the EnRs for MV-POA and HV-SOA are presented using the right ordinate axis.

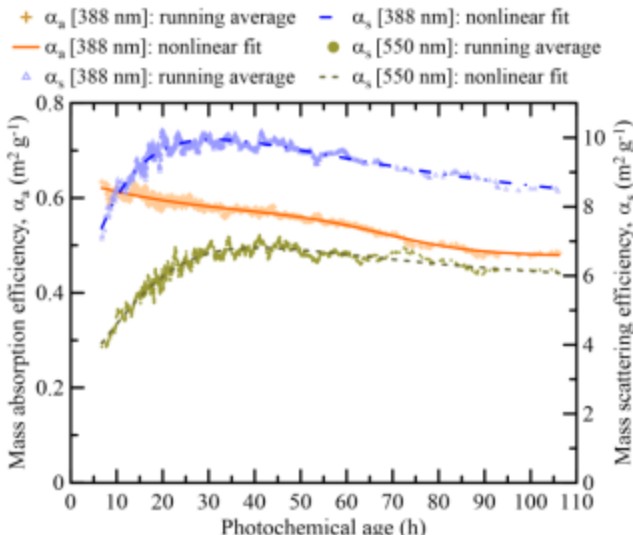

**Figure 9.** Evolution of the mass absorption efficiency at 388 nm (left axis) and mass scattering efficiency at 388 and 550 nm (right axis) according to the simulations with the CHIMERE CTM and OPTSIM module for the 'bb_vbs' scenario.

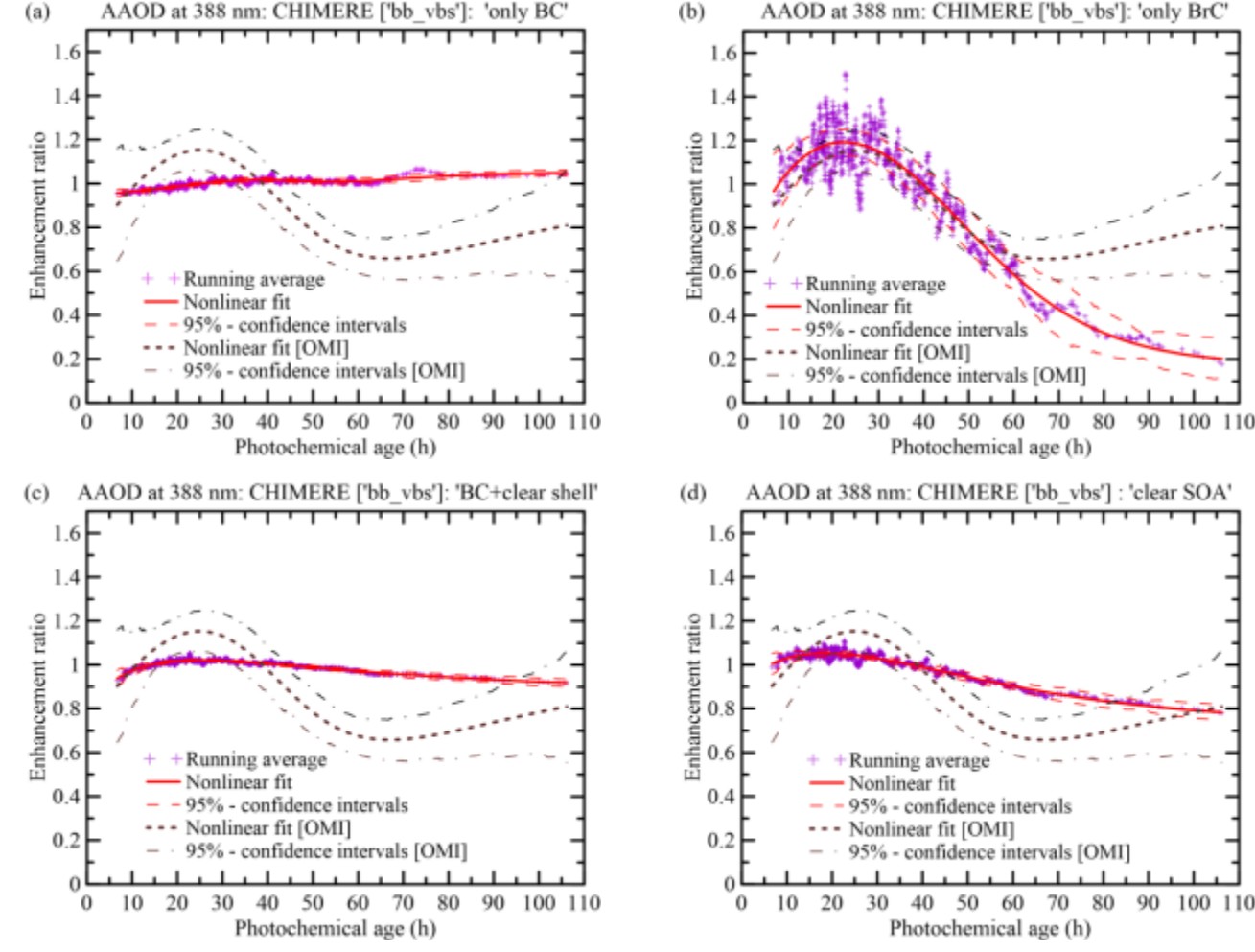

**Figure 10.** Evolution of EnR$_{abs}$ according to model results for several limiting cases involving different specifications of the optical properties of the individual components of BB aerosol: (a) absorption is determined only by BC, (b) absorption is determined only by BrC, (c) absorption is determined only by the BC core surrounded by the non-absorbing shell, (d) absorption is determined only by both the BC core and OA shell, but SOA is non-absorbing. The trends in EnR$_{abs}$ according to the analysis of satellite observations (Figs. 5a) and its confidence intervals are also shown for comparison. The corresponding model results for the base case where the BC core is surrounded by the absorbing shell are shown in Fig. 7a.