# Peer review of "Insights into the aging of biomass burning aerosol from satellite observations and 3D atmospheric modeling: Evolution of the aerosol optical properties in Siberian wildfire plumes"

_Atmospheric Chemistry and Physics, 2020_

## Referee Comment (RC1) · Anonymous Referee #1 · 22 Aug 2020

This is a detailed study combining satellite observations and modeling of biomass burning SOA. It provides several insights on evolution of BBOA and its optical properties based on analyses of a field dataset of a fire event.

Following are some key comments/suggestions for improvement: 1. How is plume rise included in Chimere? What is the maximum vertical height of injection?

2. A lot of model measurement evaluation is based on satellite data that have their own

challenges as described in this study. Are there any in-situ measurements of organic aerosols at source/receptor sites? These would be valuable to constrain surface and vertical profiles of OA simulated by Chimere.

3. Biomass burning emissions inventories like GFED often severely under-predict POA emissions from biomass burning. Can the authors provide any discussions/evaluations of primary BBOA emissions used in the model? Along the same lines, if inventories are missing a large BB-POA source, one could simply increase POA by a factor of 3 to 5, turn off BB-SOA formation and just age POA. This may result in different size evolutions (e.g. SSA, scattering efficiency), AAOD and AOD evolution. Would be interesting to see how this could affect the results and interpretations of this study.

4.Page 18: Its confusing as written: " We selected only those simulations that have the same ranks as the subset of CO observations matching (as described above) the OMI AAOD data". Please elaborate. How did the authors calculate ranks? Did they consider different simulations driven by different renalyses to minimue plume dispersion errors?

5. Table2: In equation R3 how were coefficients 0.33 and 0.30 determined? Also in R5 and R6 85% of SOA is lost by fragmentation. Is the fragmented SOA recirculated or does it react to yield some MV-SOA/LV-SOA?

6. Figure 2: Why does Chimere have too high AOD compared to MODIS is sSouth-Eastern part of domain?

7. Page 22 Before section 3.2: Can the auhtors comment on relative importance of various processes affecting decrease of AOD to AAOD from source to receptor, e.g. fragmentation, evaporation etc.?

8. Figure 10: Seems we need another case showing BC core with absorbing OA shell and lensing effects.

9. How does particle water affect calculated optical properties, especially if water is on the shell?

10. It would be inutitive to see a map of particle water over the domain, especially given discussion of potential importance of heterogeneous oxidation of biomass burning OA. I would think that even with hygroscopicity of 0.2 water content of BBOA will be large due to its high concentration.

11. Can the authors comment on role of photolysis of biomass burning SOA as its loss process?

12. Page 28: If SOA is glassy, it may not mix with POA. How are the authors treating absorptive gas-particle partitioning of POA-SOA mixtures. One could envision treating them as two separate solutions from Raoult's law perspective.

13. Would the increase of brown carbon with BC to OA ratio imply LV-SOA is more absorbing (brown) than MV-SOA? I recall this was implied in Saleh et al. study referenced by the authors. But this is contrary to the author's hypothesis that LV-SOA is much less absorbing than MV-SOA. This may need some discussions.

14. In Conclusions, the authors say 100h processing decreases ENRs for AOD and AAOD by 45% but not SSA. Why? One would expected SSA to change as the size distribution shifts with changes in evaporation, fragmentation etc. Would be nice to show change in size distributions simulated by Chimere with different processing times.
* * *

---

## Referee Comment (RC2) · Anonymous Referee #2 · 7 Oct 2020

This study analyses changes in optical properties of Siberian biomass burning aerosols during their atmospheric transport using a combination of satellite observations and chemistry-transport model simulations (the CHIMERE model) over Eastern Europe. Similarly to their previous study Konovalov et al., 2017 (https://doi.org/10.5194/acp-17-4513-2017), authors use the enhancement ratios in AOD, AAOD and SSA due to the formation of organic aerosols from BB emissions relative to the corresponding enhancement of an inert aerosol tracer to investigate processes that occur during atmo-

spheric aging of BB plumes. I have several major concerns with the assumptions used in this study as well as the interpretation of the results that need to be addressed prior to publication.

Major concerns:

1) The representation of organic aerosol chemistry and processes within the CHIMERE model is expected to play a key role in the interpretation of the satellite observations, and in the conclusions of this paper. My concern is that the parameterizations used in this study are either somewhat outdated i.e. for biogenic and anthropogenic precursors, or have not been previously evaluated i.e. biomass burning precursors. For instance, the VBS parameterization used in this study for BB precursors was derived from the VBS proposed by Ciarelli et al., 2017 that provided a hybrid volatility basis-set model for aging of wood-burning emissions. It seems that organic compounds were lumped over several volatility bins, and given different properties and aging reactions (Table 2), and this was done without any constraint from experimental data. It is critical for this paper to demonstrate that the derived simplified mechanism provides accurate results. Authors should provide a box model simulation comparing their simplified VBS parameterization with the original one for various aging experiments, as well as comparing it with previously published experimental measurements (e.g. total yields) and/or other VBS parameterizations used for BB precursors (e.g. Shrivastava et al. 2017, Majdi et al., 2017).

The term "mechanistic (p4, p13, p30)" should not be used here to refer to the representation of BB organic aerosols in the CHIMERE model given that there is not process level representation of the underlying chemistry and optical properties.

2) The term "BB aerosol photochemical age" is misused in this study. As defined on page 7, this term does not account for the photochemical reactions or the chemical aging of the BB plume. It only accounts for the sunlight exposure of the plume, and should be referred to as "hours of sunlight exposure". This needs to be corrected

throughout the manuscript and for corresponding figures (e.g. Figure 5).

Also, it is unclear how the transport time from the source region of a given BB plume was determined for the satellite data, and for the model. Please add this explanation to the methods section.

3) This study uses a large number of assumptions, e.g. parameterizations of aging of organic compounds, their optical and absorptive properties, fire emissions, averaging in time and space to match satellite measurements, etc. Please make a table that summarizes all the assumptions used in this manuscript, and quantify the associated sensitivity of the conclusions to this assumption. This is needed to show that the conclusions of this study are robust.

4) Does the proposed method allow separating between the changes in AOD due to oxidation and gas-particle partitioning vs. those due to dry and wet removal of organic gases and particles and subsequent evaporation/condensation. This needs to be clearly explained and justified.

5) Can the CHIMERE model capture the emissions and transport of the smoke plume during the studied period before all the corrections have been applied to the model? In particular, I am concerned about the coarse vertical resolution with only 12 levels up to 200hPa. What is the uncertainty in the transport and vertical distribution of smoke associated with this poor model resolution?

In addition, BB emissions were estimated using the satellite FRPs, and emission factors. What is the total amount of OA, BC, CO emitted by these fires during the period of interest, and how does this emission estimate compares with other publicly available emission inventories e.g. GFED or FINN. By how much were these emissions adjusted to match the satellite AOD data?

6) The analysis performed in this study are following very closely the approach used in the previous study by Konovalov et al., 2017 (except for the estimate of the photochemical age, and the study of a different Siberian fire even). The originality and significance of the present study needs to be well justified with regard to the previous one in the discussion section.

Minor concerns:

The introduction is quite long and dense. Please try to shorten by avoiding the redundancies. Also the description of the modeling approach should be moved into the methodology section (p4 line 19 to p6 line5).

p1 line: 13: please remove "including the Arctic". p1 line 13: change "Atmospheric evolution" to "changes that occur in". p3 line 20 remove "recalcitrant" p5 line 1: take a step further instead of forward? p5 line 21 – remove "clockwise, and counterclockwise". p9 line 21: Remove the parenthesis after tabs. p10 line 17: remove "numerous" p10 line 22: provide a reference for the melchior2 chemical mechanism, and for Fast-JX. p12 line 10: provide a reference for LMDZ-INCA boundary conditions. p17 line 11: What is the model resolution used in this study? And for this regridding? p22 line 5-8: these account for different airmasses?

Figure 4, should these AODs be compared quantitatively given all the adjustments that are applied to the emissions (p19 e.g. equation 9)?

[Figure]

---

## Referee Comment (RC3) · Anonymous Referee #3 · 8 Oct 2020

This paper uses satellite retrievals of OMI AAOD, OMI SSA, MODIS AOD and IASI CO and the CHIMERE CTM to understand the aerosol evolution of a large smoke plume transported from Siberia to European Russia during July 2016, focusing on the different contributions of BC, BrC, POA and SOA. The motivation, methods and data are described with a high amount of detail. The model adjustment process for the VBS scheme in particular is described transparently and precisely. There is an interesting diagnosis of the contributions of POA and SOAs to AOD, and helpful sensitivity tests to

understand the importance of different contributing processes (e.g. wrt lensing effects, role of BC) to aerosol enhancement. The big picture interpretation of OA enhancement is nicely summarized (P27L14). Overall, this is a focused and very well-written paper.

My only main question was about model/satellite data harmonization. At P17L9, the approach to harmonize the model and satellite data, in terms of temporal and spatial sampling is described in detail. One important aspect, which I did not see described: as part of the satellite/model harmonization, how was the model data filtered for retrieval quality, by which, masking out of retrievals under cloudy conditions is meant? At P5L23, the predominantly cloudless conditions during this period are mentioned, but a quick inspection of the MODIS true color imagery and retrieved AOD show a mix of cloudy and clear sky conditions during the second half of July 2016. Under cloudy conditions, much of the AOD is masked out, as is the case for the OMI AOD and AAOD retrievals (for L3, at least, and presumably for many of the L2 pixels and individual CO retrievals). Is CTM filtered accordingly to not introduce a discrepancy due to the inclusion of simulated data under cloudy conditions where the retrievals fail? This could even be as basic as an ad-hoc threshold of cloud fraction or cloud optical depth, depending on what is available, so that the CTM data are 'biased' toward clear sky conditions in the same way as the satellite data. Or are the CTM cloud fields in sufficient enough agreement with those seen by the satellites that the retrieval co-location sampling handles this?

Minor comments P1L12: In the abstract, consider a basic description of importance of how the VBS scheme improved agreement between the model and satellite data (i.e. in Fig 4 b,d). This point is worth mentioning.

P21L6: In Figure 2b and 2d, I would suggest that the AOD scale range from, say, 0-2, to get a better sense of the AOD enhancement over the receptor region centered on Moscow.

P21L9: by correlation, do you mean spatial pattern correlation or something else?

Minor editorial point throughout: use the word 'rather' fewer times.

---

## Author Response (AR1)

Dear Prof. Sergey A. Nizkorodov,

Thank you very much for the handling of the reviewing process for our manuscript. We have carefully addressed all of the comments by the anonymous referees # 1 and 3 in the revised manuscript. We also did our best to address the comments by anonymous referee # 2, although we found that most of them are either not fair or already addressed in the reviewed manuscript. Our point-by-point responses to the referees' comments are provided below. All the changes and corrections are highlighted in the revised manuscript and in the Supplementary Material which are also provided below. We hope that you will find the revisions satisfactory and sufficient.

Respectfully,

Igor Konovalov

on behalf of all the authors

**Authors' response to the comments of Anonymous Referee # 1**

We thank the Referee for the constructive comments. We tried to address them in the best way possible in the revised manuscript. Our point-by-point responses to the Referee's comments are provided below.

Referee's comment: *1. How is plume rise included in Chimere? What is the maximum vertical height of injection?*

As noted in the reviewed manuscript (page 11, lines 1, 2, 10), the fire emissions in our simulations were specified using a methodology that included calculations of the emission heights using the parameterization proposed by Sofiev et al. (2012) and was detailed in our previous papers. In response to the Referee's comment, a concise description of this parameterization and the information on its validation are provided in Sect. 2.3 of the revised manuscript. In particular, we note that the maximum injection heights of biomass burning (BB) emissions were calculated for each FRP pixel as a function of the observed FRP, the boundary layer height, and the Brunt–Väisälä frequency. In other words, the maximum vertical height of injection was not assumed to be constant but varied both in time and space. We also explain that the emissions for a given model layer were computed proportionally to the weighted number of pixels yielding the maximum injection height corresponding to this layer, with the weight of each pixel evaluated proportionally to the corresponding FRP. The average heights of the BB plumes considered in our analysis are reported in Figs. S8b and S6b in the Supplementary Material for the reviewed and revised versions of our manuscript, respectively.

Referee's comment: *2. A lot of model measurement evaluation is based on satellite data that have their own challenges as described in this study. Are there any in-situ measurements of organic aerosols at source/receptor sites? These would be valuable to constrain surface and vertical profiles of OA simulated by Chimere.*

Indeed, the satellite data which were used in our analysis are certainly not perfect. However, as pointed out in the reviewed manuscript (page 9, lines 6-12), the OMI (OMAERUV) AAOD retrievals were evaluated, both directly and indirectly (through the modeled BB emissions constrained with these retrievals), using AERONET, aircraft, and in situ data in Siberia as part of our previous study (Konovalov et al., 2018) which revealed no serious biases in these retrievals. The uncertainties in the MODIS AOD data are well documented and are relatively small (Levy et al., 2010). Accordingly, we see no evidence that the strong changes in the BB aerosol optical properties, which are identified in our analysis, can be an artifact of biases in the satellite data. Random uncertainties in the satellite data are, of course, quite considerable, but they are taken into account in our estimates of the confidence intervals for the derived tendencies in the analyzed characteristics.

Regrettably, we do not have any measurements of organic aerosols in the study region in the analyzed period at our disposal. But even if such measurements were available, they would hardly be sufficiently informative for our study: as we argue in the introduction, field studies show a very diverse picture of atmospheric transformations of BB aerosol, reflecting big diversity in fuel type and burning conditions, and so the sparse and often contradictory results of these studies can hardly provide consistent observational constraints for model representations of the effects of atmospheric aging of BB aerosol originating from multiple fires from a big region, specifically Siberia.

Taking into account the Referee's comment, we extended the discussion of possible uncertainties in our analysis, including those associated with the analysis of satellite data and the modeling, and provided a caveat concerning potential biases in the satellite data. This discussion is presented in the newly-introduced section (Sect. 3.4) of the revised manuscript.

Referee's comment: *3. Biomass burning emissions inventories like GFED often severely under-predict POA emissions from biomass burning. Can the authors provide any discussions/evaluations of primary BBOA emissions used in the model? Along the same lines, if inventories are missing a large BB-POA source, one could simply increase POA by a factor of 3 to 5, turn off BB-SOA formation and just age POA. This may result in different size evolutions (e.g. SSA, scattering efficiency), AAOD and AOD evolution. Would be interesting to see how this could affect the results and interpretations of this study.*

We thank the Referee for this insightful comment. Indeed, there is evidence (partly based on multiple studies that reported underestimation of AOD by models) that GFED and other inventories strongly underestimate POA (or OC) emissions from biomass burning. At the same time, the underestimation of AOD by a model can, at least partly, be caused by missing SOA sources in the model representation of organic aerosol, as argued, in particular, in one of our previous papers (Konovalov et al., 2015). So, in our understanding, there is a general problem concerning the evaluation of POA (OC) emissions provided by inventories. The results of our study have some implications in the context of this problem, even though the evaluation of the BB emission inventories is not its primary objective.

To address the Referee's comment, we provided a comparison of our "top-down" estimates of OC and BC emissions in the study region in July 2016 (Sect. 2.6 of the revised manuscript) with the bottom-up estimates obtained from the GFED4.1s inventory (van der Werf et al., 2017). While the top-down OC emission estimate for the 'bb_vbs' simulation scenario is found to agree well with the GFED data, the top-down estimate obtained using the 'bb_trc' simulation (which does not take into account the SOA formation) is almost a factor of 3 bigger than the corresponding bottom-up estimate (similar to what was found in our previous study (Konovalov et al., 2018)). As a caveat, we note that the good agreement of the bottom-up and top-down estimates of the OC emissions does not necessarily indicate that the emission inventory is accurate: ideally, the consistency of top-down and bottom-up estimates of the BB OC emissions should be examined by taking into account the partitioning of the measured emissions between gases and particles, but the corresponding data are not provided as part of emission inventories.

Following the referee's instructions, we also performed an additional simulation (referred to as 'bb_poa') in which the SOA formation was turned off but POA emissions were strongly increased (specifically, by a factor of 4.3) to retain the consistency between the simulated and observed AOD in the source region. The results of this simulation are summarized in Sect. 3.3 and presented in more detail in the Supplementary Material (Sect. S5). Consistently with our conclusions about the major role of SOA formation in the evolution of BB aerosol during the analyzed event, the simulations for the 'bb_poa' scenario are not found to exhibit any qualitative nonlinear features of the behavior of SSA and EnR for AOD, which were identified in the analysis of the satellite observations and were reproduced in the simulations for the 'bb_vbs' scenario. We consider this result as evidence that the proposed joint analysis of satellite and model data can be used to distinguish between the two possible reasons for the underestimation of AOD by a model, such as (1) an underestimation of OC emissions in the emission inventory and (2) missing SOA sources in the model.

*Referee's comment: 4. Page 18: Its confusing as written: " We selected only those simulations that have the same ranks as the subset of CO observations matching (as described above) the OMI AAOD data". Please elaborate. How did the authors calculate ranks? Did they consider different simulations driven by different renalyses to minimize plume dispersion errors?*

We are sorry that the sentence indicated by the referee was confusing, and we tried to explain our procedure more clearly in the revised manuscript. As it is common in statistics (https://en.wikipedia.org/wiki/Ranking), the ranking of a set of data points was done in our case by assigning the order number to values arranged in ascending order. We'd like to note that while the selection procedure based on the ranking of the CO columns retrieved from satellite observations and simulated with CHIMERE was introduced to ensure more accurate quantitative results of our analysis, our main conclusions concerning the qualitative features of the BB aerosol evolution do not depend on the application of this procedure.

Unfortunately, we could not perform simulations based on different reanalyses in the framework of this study due to computational limitations. We are also not sure that the use of different reanalyses could result in a major reduction of the differences between the simulated and observed data, since these differences appear to be more due to uncertainties in the temporal and spatial variability of fire emissions rather than due to model transport errors.

*Referee's comment: 5. Table2: In equation R3 how were coefficients 0.33 and 0.30 determined? Also in R5 and R6 85% of SOA is lost by fragmentation. Is the fragmented SOA recirculated or does it react to yield some MV-SOA/LV-SOA?*

The coefficients 0.33 and 0.3 were defined to retain the same total SOA yields for HV-SOA and LV-SOA from the oxidation of NTVOCs (~ 43 g mol$^{-1}$) as in Ciarelli et al. (2017) (page 14, lines 16, 17 of the reviewed manuscript), taking into account that the assumed molecular weights of HV-SOA and LV-SOA are 131 and 144 g mol$^{-1}$. In the revised manuscript (Sect. 2.4), we additionally explain that we took into account the ratio of the molar masses of these species (131/144 $\cong$ 0.30/0.33). The products of the fragmentation pathway of reactions (R5) and (R6) were considered as volatile and did not contribute to the SOA formation (page 14, line 27 of the reviewed manuscript).

Referee's comment: *6. Figure 2: Why does Chimere have too high AOD compared to MODIS is South-Eastern part of domain?*

We addressed this question in Sect. 3.1 of the revised manuscript. We suppose that the simulated AOD is biased high outside of the study region in the south-eastern part of the model domain as a result of spatial variations in any parameters determining (or affecting) the relationship between the real biomass burning rate and the calculated aerosol emissions. Such parameters include, in particular, the emission factors, volatility distributions for the POA emissions, and empirical factor relating FRP to the BB rate. The variations in these parameters may, in turn, be due to the inhomogeneity in the spatial distribution of the vegetation species across the study region and model domain. It is noteworthy that the high bias in AOD is not mirrored in the spatial distribution of the simulated CO columns (see Figs. S1 and S2 in the Supplementary Material of the reviewed and revised versions of the manuscript, respectively).

Referee's comment: *7. Page 22 Before section 3.2: Can the authors comment on relative importance of various processes affecting decrease of AOD to AAOD from source to receptor, e.g. fragmentation, evaporation etc.?*

Our in-detail interpretation of the processes affecting the changes in AOD to AAOD during the transport of BB plumes from the source to receptor regions is provided in Sect. 3.3 of both the reviewed and revised versions of the manuscript. In the revised manuscript, we provided an additional remark in Sect. 3.1 that the additional changes (that cannot be explained by dilution) in AOD to AAOD are primarily caused by the losses of the medium-volatility fraction of SOA due to fragmentation.

Referee's comment: *8. Figure 10: Seems we need another case showing BC core with absorbing OA shell and lensing effects.*

Figure 10 shows our computations only for several test cases. The respective results for the base case (where, as suggested by the referee, the BC core is surrounded by the absorbing OA shell and the lensing effects are taken into account) are shown in Fig. 7a. In the revised manuscript, a corresponding explanatory remark is introduced in the caption of Fig. 10.

Referee's comment: *9. How does particle water affect calculated optical properties, especially if water is on the shell?*

Since the relative humidity (RH) characterizing ambient conditions for the BB plumes analyzed in our study typically remained below 70 % (see Figs. S8a and S6a in the reviewed and revised versions of the manuscript, respectively), we did not expect, based on the common knowledge (e.g., Reid et al., 2005), that the water uptake by particle could significantly affect BB aerosol optical properties and their evolution. Indeed, according to our simulations, the water mass fraction stays, on the average, below 15% as long the photochemical age is less than 60 h, eventually increasing up to 26% toward the end of the evolution period considered (see Fig. 8a in the revised manuscript). Possible effects of the water uptake on our simulations are discussed in Sect. 3.3 of the revised manuscript. Specifically, we argue that water and inorganic ions increase the scattering cross-section of BB aerosol particles, thereby increasing AOD and SSA, although the contribution of a unit mass of water to the scattering efficiency is expected to be considerably smaller than that of the organic matter because the real component of the refractive index for water is substantially smaller than for the organic species. We also point out that a substantial increase in the water uptake occurs in our simulations only after 60 h, and therefore it could not contribute significantly to the strong increase in EnR for AOD before 30 h. The water uptake by the organic shell of the particles can also contribute to the lensing effect, thereby increasing AAOD, but the evolution of EnR for AAOD in our case was found to be determined by other factors anyway, as illustrated in Fig. 10. In addition to this discussion, we explicitly estimated the effects of water on AOD, SSA, and the scattering efficiency in the 'bb_poa' test simulation mentioned above. These estimates, which are discussed in Sect. S5 of the revised Supplementary Material and are presented in Fig. S8, confirm that the water uptake of particles was not among the key factors that drove the evolution of the optical properties of BB aerosol during the analyzed episode.

5   Referee's comment: *10. It would be intuitive to see a map of particle water over the domain, especially given discussion of potential importance of heterogeneous oxidation of biomass burning OA. I would think that even with hygroscopicity of 0.2 water content of BBOA will be large due to its high concentration.*

The calculated mean water content in the BB particles is shown in Fig. 8a of the revised manuscript as a function of the photochemical age. As noted above, it turned out to be relatively small, since the ambient relative humidity in the
10 center of mass of the BB plumes typically did not exceed 70 %. We presumed that Fig. 8a would be more informative than a map suggested by the referee, because the calculated mean water fraction takes into account the spatial inhomogeneity of the aerosol mass loadings in the atmospheric columns, giving larger weights to the high loadings. As the manuscript and Supplementary Material are already lengthy, we opted to not provide one more figure (which would also entail additional discussion). Note that Fig. 8a provided in the reviewed manuscript was meant to implicitly show
15 the contribution of water among "other" components. However, thanks to the Referee's comment we noticed that the model output data for these components were not processed properly. We apologize for this oversight. A corrected and improved version of Fig. 8a, in which the contribution of water is shown explicitly, is provided in the revised manuscript.

  Referee's comment: *11. Can the authors comment on role of photolysis of biomass burning SOA as its loss process?*

20 A corresponding comment is provided in Sect. 3.4 of the revised manuscript. In particular, we mention the available global-scale estimates (Hodzic et al., 2015) of the SOA lifetime with respect to the in-particle photolysis, and also note that if the in-particle SOA photolysis were a primary driver for the decrease of AOD in our analysis, then the rate of the decrease in EnR$_{ext}$ would not be expected to strongly decelerate after about 70 h, as it does according to Fig. 5b.

  Referee's comment: *12. Page 28: If SOA is glassy, it may not mix with POA. How are the authors treating absorptive*
25 *gas-particle partitioning of POA-SOA mixtures. One could envision treating them as two separate solutions from Raoult's law perspective.*

We made it clear in the revised manuscript (Sect. 2.4) that following Ciarelli et al. (2017) we assumed, for definiteness, that all organic species within particles form a well-mixed liquid and inviscid solution. Possible implications of this assumption – which cannot be validated or invalidated using available observations in Siberia – are briefly dis-
30 cussed in Sect. 3.4 of the revised manuscript. In particular, we note that if POA and SOA species do not mix in reality but form two separate solutions (as assumed, e.g., in Shrivastava et al., 2015), then our model is likely prone to overestimation of their concentrations in particles. It is, however, hard to see how these potential biases can invalidate our qualitative interpretation of the major changes in the BB aerosol optical properties. We plan to address the possible effects of viscosity and non-ideality of organic solutions on the Siberian BB aerosol evolution in our future studies.

35   Referee's comment: *13. Would the increase of brown carbon with BC to OA ratio imply LV-SOA is more absorbing (brown) than MV-SOA? I recall this was implied in Saleh et al. study referenced by the authors. But this is contrary to the author's hypothesis that LV-SOA is much less absorbing than MV-SOA. This may need some discussions.*

This point is briefly discussed in Sect. 2.4 of the revised manuscript. Specifically, we note that our assumption that LV-SOA is non-absorbing does not contradict the experimental findings indicating high absorptivity of low-volatility
40 organic compounds (Saleh et al., 2014), because the contributions of POA and SOA species to absorption were not isolated in these experiments, and the effect of SOA addition was found there to be comparable to measurement uncertainties.

Referee's comment: *14. In Conclusions, the authors say 100h processing decreases ENRs for AOD and AAOD by 45% but not SSA. Why? One would expected SSA to change as the size distribution shifts with changes in evaporation, fragmentation etc. Would be nice to show change in size distributions simulated by Chimere with different processing times.*

SSA was evaluated in the standard way as 1-AAOD/AOD according to Eq. (6). So, as both AOD and AAOD decrease (due to various processes, including those indicated by the referee) to about the same extent, SSA is not significantly affected. The average size distributions of BB aerosol particles for different processing times are shown in Figs. S9 and S7 of the Supplementary Material of the reviewed and revised versions of the manuscript, respectively. We are sorry if we did not understand this comment of the Referee properly.

We are not aware of any state-of-the-art parameterizations for biogenic and anthropogenic precursors that have been
30 shown to allow improving 3D-model simulations of biogenic and anthropogenic aerosol in Siberia compared to simulations based on relatively simple parameterizations similar to those implemented in the standard version of CHIMERE. At the same time, we believe that our analysis could not be significantly affected by possible uncertainties in the modeled concentrations of anthropogenic and biogenic aerosol. This is specifically indicated by the facts that (1) the AOD level during the analyzed episode exceeded the background AOD level on the average by at least a factor of
35 6 (see Fig. 4 c, d in the reviewed manuscript) and (2) that the AOD simulations performed without fire emissions fit well the AOD observations in the periods when fire emissions were small (see Fig. S2). The sensitivity of the inferred evolution of the enhancement ratios for both AAOD and AOD to the assumed background conditions was examined in the supplementary section S3 of the reviewed manuscript. Regrettably, we see no evidence that the results of this analysis were taken into account by the referee.

40 Concerning the modeling of biomass burning precursors of aerosol, we argued in the introduction of the reviewed manuscript (page 3, lines 26-28 and above), that "… the sparse and often contradictory results of field and laboratory studies available so far can hardly provide consistent observational constraints for representations of the effects of atmospheric aging of BB aerosol in chemistry-transport and climate models". We also noted that "using the different VBS oxidation schemes partly constrained by laboratory measurements or atmospheric observations to simulate the

multi-day BB aerosol evolution under fixed ambient conditions has been found to result in major quantitative and even qualitative differences between the simulations (Konovalov et al., 2019)" and that "it is not given that any of the available schemes can adequately describe the BB aerosol evolution specifically in Siberia" (page 13, lines 5-8). So, instead of using data of specific laboratory experiments (which are always representative of particular fuel types and a particular range of burning and ambient conditions), we constrained our simulations by satellite observations which are representative of numerous BB plumes in the real atmosphere. These constraints are shown to successfully take the role of the more traditional but sparse and limited constraints provided by "experimental" data from chamber experiments. The BB aerosol evolution simulated with the proposed VBS scheme is demonstrated to be consistent with the satellite observations as evidenced by results shown in Fig. 7 of the reviewed manuscript. Regrettably, we have no evidence that these results were taken into account by the Referee.

Referee's comment: *Authors should provide a box model simulation comparing their simplified VBS parameterization with the original one for various aging experiments, as well as comparing it with previously published experimental measurements (e.g. total yields) and/or other VBS parameterizations used for BB precursors (e.g. Shrivastava et al. 2017, Majdi et al., 2017).*

Regrettably, we do not see how the suggested comprehensive comparison of box model simulations involving different parameterizations could help us to "demonstrate that the derived simplified mechanism provides accurate results". Indeed, to the best of our knowledge, neither of the alternative VBS parameterizations proposed in the literature so far has been shown to enable an adequate model representation of the atmospheric evolution of BB aerosol in Siberia. Specifically, the original VBS scheme proposed by Ciarelli et al. (2017) was evaluated only against the measurements of organic aerosol (OA) produced from several burns of a particular type of fuel (beech logs) in a residential wood burner. Hence, there is no evidence that the original scheme is applicable to simulations of BB aerosol in Siberia.

In this situation, any difference or resemblance of results obtained with different mechanisms could hardly be indicative of any shortcomings in either of them. Furthermore, a numerical analysis employing a microphysical box model and several VBS schemes of various complexities have already been performed in Konovalov et al. (2019) for a wide range of conditions representative of BB plumes in Siberia. The main conclusions of the previous box model analysis, which are relevant for the present study, are summarized in the introduction and Sect. 2.4 of the reviewed manuscript, and we tried to improve this summary in the revised manuscript.

That said, we agree that a comparison of box model simulations with our simplified parameterization to similar simulations with the original (C17) scheme can be useful as it can provide evidence on whether or not our VBS parameterization enables a realistic representation of the initial stage of the BB aerosol evolution, which is poorly represented in the satellite observations used as constraints for our model. To this end, using the microphysical box model described in Konovalov et al. (2019) with both the VBS scheme used in this study and the original C17 scheme, we simulated the BB OA evolution under the conditions of chamber experiments reported in Ciarelli et al. (2017). The results of these simulations are mentioned in Sect. 2.4 of the revised manuscript and are presented in more detail in the newly introduced supplementary section S2. Briefly, we found that the BB OA concentration initially increases more rapidly in the simulation with our scheme than in the simulation with the original scheme, and so the BB OA concentration predicted by our scheme after about 10 hours of evolution is about 40% larger than the corresponding concentration predicted by the original scheme. Nonetheless, taking into account the range of the experimental variability of BB OA concentrations, the BB OA evolution simulated with our VBS scheme does not look unrealistic. A major qualitative difference between the two simulations is that the original scheme demonstrates a monotonically saturating increase of BB OA concentration, whereas the simplified scheme yields a non-monotonic behavior of BB OA concentration (a rapid increase followed by a gradual decrease due to fragmentation of SOA). Accordingly, a smaller concentration (also by about 40 %) is found in the simulation with our scheme after 110 hours of evolution. These two types of behavior of BB OA were earlier identified and explained in Konovalov et al. (2019).

Referee's comment: *The term "mechanistic (p4, p13, p30)" should not be used here to refer to the representation of BB organic aerosols in the CHIMERE model given that there is not process level representation of the underlying chemistry and optical properties.*

The meaning of the term "mechanistic model" is somewhat uncertain and varies across different branches of science. Here we specifically meant that our scheme is expected to take into account basic physical "mechanisms" (such as oxidation/evaporation, a decrease or increase of the volatility of semi-volatile organic species, destruction of chromophores) driving the BB aerosol evolution rather than individual chemical processes (such as e.g., the oxidation chain of naphthalene). To avoid misunderstanding, we are not using this term in the revised manuscript, following the Referee's instruction.

Referee's comment: *2) The term "BB aerosol photochemical age" is misused in this study. As defined on page 7, this term does not account for the photochemical reactions or the chemical aging of the BB plume. It only accounts for the sunlight exposure of the plume, and should be referred to as "hours of sunlight exposure". This needs to be corrected throughout the manuscript and for corresponding figures (e.g. Figure 5).*

We thank the Referee for this concrete comment. Since we do not have any observable photochemical "clock" at our disposal and cannot be sure that the model predicts OH concentration within the plumes accurately, we approximately quantify the exposure of BB aerosol to photochemical processes by assuming that OH concentration in the BB plumes is constant during daytime and is zero during the nighttime. Such an approximate estimate of the BB aerosol photochemical age accounts for the well-known fact that the photochemical processing of organic aerosol is typically much faster during the daytime than during the night time (with exceptions for highly unsaturated forested emissions which are efficiently oxidized also by $NO_3$ and $O_3$). In the revised manuscript, the corresponding definition in Sect. 2.1 is corrected accordingly, and an additional caveat is provided. The term "hours of sunlight exposure" is rather cumbersome and is difficult to use throughout the text, although we agree that it is more accurate (as it was noted in the reviewed manuscript).

Referee's comment: *Also, it is unclear how the transport time from the source region of a given BB plume was determined for the satellite data, and for the model. Please add this explanation to the methods section.*

We presume that the Referee refers to Sect. 3.2 where we identified a strong association between the photochemical age of BB aerosol and the geographical location (specifically, the longitude) of the BB plumes transported westward. This association, which is illustrated in Fig. 6 of the discussion paper, allowed us to estimate the ranges of the photochemical age of BB aerosol in the source and receptor regions. This analysis does not involve any other definitions apart from those already introduced in Sects. 2.1 and 2.3 (the photochemical age or hours of sunlight exposure of BB aerosol) and in Sect. 2.5 (coordinates of the source and receptor regions).

Referee's comment: *3) This study uses a large number of assumptions, e.g. parameterizations of aging of organic compounds, their optical and absorptive properties, fire emissions, averaging in time and space to match satellite measurements, etc. Please make a table that summarizes all the assumptions used in this manuscript, and quantify the associated sensitivity of the conclusions to this assumption. This is needed to show that the conclusions of this study are robust.*

A table summarizing distinctive features and parameterizations of our modeling system is provided in the revised manuscript (Table 3). Furthermore, to address the Referee's comment, we have presented the discussion of the uncertainties in a more focused way in a new section (Sect. 3.4). This new section brings together the relevant content of Sects. 3.2, S3, and 3.3 of the reviewed manuscript, and also it includes some additional discussion.

We dedicated considerable efforts to ensure that the conclusions of our analysis are robust, and these efforts were comprehensively described in the reviewed manuscript. In particular, we quantified the uncertainties in the derived trends of the optical characteristics using a statistical (bootstrapping) method as described in Sect. 2.1. The obtained confidence intervals were shown in all our figures presenting the evolution of the optical characteristics according to

both the satellite data and model results. Furthermore, a special section (Sect. S3) was dedicated to the analysis of test cases addressing possible biases associated with the processing of the satellite data in our study. The results of this analysis allowed us to conclude that "the tests overall confirm the robustness of our major findings". Regrettably, we see no evidence that these efforts were noticed and appreciated by the Referee.

5   While the main results of our analysis of the satellite data are quantitative, our simulations involving the VBS parameterization are mainly used to get insights into qualitative patterns of the BB aerosol transformations, rather than for quantitative characterization of any processes. This feature of our modeling analysis is emphasized in the introduction (page 5, lines 3-5) of the reviewed manuscript and rephrased in the next sections (page 11, lines 1, 2; page 24, lines 12-18; page 28, lines 32, 33). To "quantify" the associated sensitivity of the qualitative conclusions to any assumptions involved in our study is logically not feasible. Furthermore, we emphasized that our interpretation of the detected changes in AAOD, AOD, and SSA is not necessarily unique (page 27, line 30). At the same time, a discussion of several major factors which can potentially affect our conclusions (page 28) allowed us to conclude that "that although our model representation of BB aerosol evolution involves strong assumptions (which yet need to be verified in future research), our qualitative interpretation of the inferred major changes of the optical properties of BB aerosol in Siberia is sufficiently robust and realistic". We regret that this discussion also apparently went unnoticed by the Referee.

Referee's comment: *4) Does the proposed method allow separating between the changes in AOD due to oxidation and gas-particle partitioning vs. those due to dry and wet removal of organic gases and particles and subsequent evaporation/condensation. This needs to be clearly explained and justified.*

As explained in Sect. 2.1 (page 6, lines 14-16) of the reviewed manuscript, "the analysis of EnRs is expected to reveal
20   the differences between the dynamics of AAOD or AOD in the real BB plumes and in a hypothetical simulation in which BB aerosol is assumed to consist of only non-volatile material and SOA formation processes are disregarded". This explanation implies that, by design, our method does not allow separating between the processes indicated by the Referee. We noticed, however, that the simplified formulation of the idea of our analysis in the reviewed manuscript (page 6, lines 11-14) could be somewhat confusing. In the revised manuscript, we clarified that the objective of our
25   analysis was to identify the differences between changes in AAOD or AOD due to aging of BB aerosol in the real BB plumes (including, first of all, the changes associated with oxidation and condensation/evaporation and also the changes that can be indirectly induced by the dry and wet deposition of organic gases and particles) and those in a hypothetical simulation in which the organic fraction of BB aerosol is composed of only non-volatile, inert and hydrophobic material and the SOA formation is negligible.

30   Referee's comment: *5) Can the CHIMERE model capture the emissions and transport of the smoke plume during the studied period before all the corrections have been applied to the model? In particular, I am concerned about the coarse vertical resolution with only 12 levels up to 200hPa. What is the uncertainty in the transport and vertical distribution of smoke associated with this poor model resolution?*

The performance of our model in capturing the emissions and transport of the smoke plume during the studied period
35   was validated by comparing our simulations and satellite observations of CO columns, but maybe the Referee did not notice this. This comparison was introduced in Sect. 3.1 and was presented in more detail in the supplementary section S2 of the reviewed manuscript (see also Sect. S3 of the revised manuscript). The spatial and temporal distributions of CO are driven predominantly by emissions and transport since the effects of other processes on the time scales considered are small. The spatial distributions of the CO columns during the analysis period according to the IASI observa-
40   tions and the CHIMERE simulations were shown in Fig. S1, while the corresponding temporal variations of CO in the source and receptor regions were compared in Figs. S3 and 4 (e, f) of the reviewed version (see Figs. S3 and S4 in the revised version). The satellite retrievals of CO columns are known to be sensitive to the CO vertical distribution (e.g., George et al., 2009), and this sensitivity has been taken into account in our simulations by applying the averaging kernels (page 17, lines 29, 30 of the reviewed manuscript). Hence, the comparison is indicative of the performance of
45   CHIMERE in capturing not only the spatial distributions of emissions and horizontal transport of BB plumes but also

the vertical distribution of CO. As evident in Fig. S1 (in the reviewed version), our model captures the observed spatial distribution of CO columns quite adequately, including the large-scale smoke plume extending from Siberia into the European territory of Russia. Fig. S3 further demonstrates a good quantitative agreement between the temporal variations of CO columns over both source and receptor regions (the correlation coefficient exceeds 0.9 in both cases). Finally, very small differences between the regional averages of observed and modeled CO columns (Fig. 4 e, f) indicate that the overall uncertainty in our simulations of the BB fraction of the CO columns over the receptor region (that is, within the strongly aged BB plume), including its part associated with a limited vertical resolution of our simulations is far less than the relative magnitudes of the variations in AAOD and AOD identified in our analysis. We would like to note that the vertical discretization of the simulations in our study was defined by taking into account not only available computational resources but also probable large random uncertainties associated with the vertical distribution of BB emissions (Sofiev et al., 2012). Given these uncertainties, a higher vertical discretization of our simulations would not necessarily result in smaller errors in the modeled vertical distribution of the smoke. We extended the discussion of our simulations of CO columns in the revised manuscript (specifically, in Sect. 3.1), taking into account the Referee's comment.

Referee's comment: *In addition, BB emissions were estimated using the satellite FRPs, and emission factors. What is the total amount of OA, BC, CO emitted by these fires during the period of interest, and how does this emission estimate compares with other publicly available emission inventories e.g. GFED or FINN. By how much were these emissions adjusted to match the satellite AOD data?*

Estimation of emissions of aerosol and trace gases from fires is a subject of numerous dedicated studies (including our previous study (Konovalov et al., 2018)) but is not the focus of this one. Nonetheless, to address the Referee's comment, we have provided our top-down estimates for BB emissions of BC and OC in the study region and compared them with the corresponding estimates from the GFED4.1s inventory in Sect. 2.6 of the revised manuscript.

Referee's comment: *6) The analysis performed in this study are following very closely the approach used in the previous study by Konovalov et al., 2017 (except for the estimate of the photochemical age, and the study of a different Siberian fire even). The originality and significance of the present study needs to be well justified with regard to the previous one in the discussion section.*

We regret that the originality and significance of the present study went unnoticed by the Referee. As stated in the introduction of the reviewed manuscript (page 4, lines 6, 7), "this study substantially extends the scope of the previous one by analyzing satellite observations of both absorption and extinction characteristics of BB aerosol". We further noted (page 5, lines 26, 27) that "to the best of our knowledge, this is the first study attempting to constrain simulations of the aging behavior of BB aerosol with satellite observations of both absorption and extinction AODs". Hence, we believe that our study is highly original and that its main original features are clearly explained in the introduction of both the reviewed and revised versions of the manuscript.

The significance of our study is primarily associated with its general objective, such as "to find a way to infer statistically reliable information on the impact of aging processes on the optical properties of BB aerosol from available satellite measurements" (page 3, lines 28-30). We argue that achieving this objective is important because laboratory and field studies do not "provide consistent observational constraints for representations of the effects of atmospheric aging of BB aerosol in chemistry-transport and climate models" (page 3, lines 27, 28). The significance of the study is also emphasized in the last paragraph of the conclusions. Specifically, we note that "the presented analytical framework can be helpful in identifying and interpreting manifestations of the BB aerosol aging processes far beyond the time scales that can currently be addressed in aerosol chamber experiments" and that the proposed methods of the analysis of satellite data are sufficiently general and can be applied to different satellite observations.

The original results of our analysis include an identification of the statistically significant downward parts of the trends of the enhancement ratios for both AOD and AAOD during the multi-day aging of BB plumes as well the simultaneous increase of SSA and the enhancement ratio for AOD during the initial 20-30 hours of the evolution under daytime

conditions. These are, to the best of our knowledge, unique results in the literature, and at least highly original. The interpretation of the detected changes using a VBS parameterization is certainly also a major novel point. These new results are discussed in the context of other studies (including our previous ones) in Sect. 3.2 and 3.3. Regrettably, we do not see any indications that the scientific content of these sections was examined and taken into account by the Referee.

Finally, we would like to note that although our idea to analyze satellite observations using modeled tracers was indeed introduced earlier, the concrete methods proposed in the reviewed manuscript are highly original. In particular, unlike Konovalov et al. (2017) we use a general algorithm for the nonlinear trend analysis. Differences between the analysis procedures used in Konovalov et al. (2017) and this study are discussed in the Supplementary Material, Sect. S1. The use of an adaptive VBS parameterization and explicit modeling of the optical properties of aerosol are also important novel methodological features of this study.

Taking into account the Referee's comment, we made an additional effort to emphasize the novel points of our study throughout the revised manuscript. We also tried to better explain the differences between this study and the study by Konovalov et al. (2017) in the introduction and conclusions of the revised manuscript.

15 Referee's comment: *The introduction is quite long and dense. Please try to shorten by avoiding the redundancies. Also the description of the modeling approach should be moved into the methodology section (p4 line 19 to p6 line5).*

The introduction is indeed relatively lengthy, but this is unavoidable because our study addresses not just one but several directions of research (such as modeling of BB organic aerosol, investigations of brown carbon, applications of satellite measurements of atmospheric aerosol, the role of aerosol from Siberian fires for the environment and climate).

20 We tried but could not find any redundancies. However, following the recommendation of the referee, we have removed, for the most part, the description of our modeling approach from the introduction. In this way, the introduction has been noticeably shortened.

Referee's comment: *p1 line: 13: please remove "including the Arctic". p1 line 13: change "Atmospheric evolution" to "changes that occur in". p3 line 20 remove "recalcitrant" p5 line 1: take a step further instead of forward? p5 line 21*

25 *– remove "clockwise, and counterclockwise". p9 line 21: Remove the parenthesis after tabs. p10 line 17: remove "numerous"*

All the above textual changes suggested by the referee are done in the revised manuscript.

Referee's comment: *p10 line 22: provide a reference for the melchior2 chemical mechanism, and for Fast-JX. p12 line 10: provide a reference for LMDZ-INCA boundary conditions.*

30 The requested references are provided in the revised manuscript.

Referee's comment: *p17 line 11: What is the model resolution used in this study? And for this regridding?*

The horizontal resolution of the model grid was 1 by 1 degree (page 11, line 19 of the reviewed manuscript). By saying that satellite data were projected on the model grid we implied that the regridding was done with the same spatial resolution (page 17, line 11), that is, 1 by 1 degree.

35 Referee's comment*: p22 line 5-8: these account for different airmasses?*

Yes, the conclusions in Sect 3.1 account for different airmasses, since the averaging of the observed characteristics over the big regions and period of several days allows us to suppress the variability associated with individual airmasses. A corresponding remark is added to the first paragraph in Sect. 3.1.

Referee's comment*: Figure 4, should these AODs be compared quantitatively given all the adjustments that are ap-*

40 *plied to the emissions (p19 e.g. equation 9)?*

As explained in the reviewed manuscript (page 21, lines 3-5), the goal of the comparison presented in Fig. 4 and Sect. 3.1 was to "examine the aging-changes in the optical properties of BB aerosol by considering the satellite and simulated data for the "source" and "receptor" regions. In this context, while the adjustments applied to the BB emissions allowed us to bring the simulations close to observations in the source region, they obviously could not ensure the agreement of the simulations with the observations in the receptor region where the optical properties of BB aerosol could be affected by aging processes. Hence, we consider the discrepancies between the satellite observations and simulated data for the 'bb_trc' scenario in the receptor region as an indication for the BB aerosol aging, and we do not see any reason why these discrepancies should not be evaluated quantitatively. The comparison presented in Fig. 4 was, in our opinion, sufficiently discussed in Sect. 3.1 of the reviewed manuscript.

**Authors' response to the comments of Anonymous Referee # 3**

We thank the Referee very much for the positive evaluation of our manuscript and for the useful comments, which are addressed in the revised manuscript. Our point-by-point responses to the Referee's comments are provided below.

Referee's comment: *At P17L9, the approach to harmonize the model and satellite data, in terms of temporal and spatial sampling is described in detail. One important aspect, which I did not see described: as part of the satellite/model harmonization, how was the model data filtered for retrieval quality, by which, masking out of retrievals under cloudy conditions is meant? At P5L23, the predominantly cloudless conditions during this period are mentioned, but a quick inspection of the MODIS true color imagery and retrieved AOD show a mix of cloudy and clear sky conditions during the second half of July 2016. Under cloudy conditions, much of the AOD is masked out, as is the case for the OMI AOD and AAOD retrievals (for L3, at least, and presumably for many of the L2 pixels and individual CO retrievals). Is CTM filtered accordingly to not introduce a discrepancy due to the inclusion of simulated data under cloudy conditions where the retrievals fail? This could even be as basic as an ad-hoc threshold of cloud fraction or cloud optical depth, depending on what is available, so that the CTM data are 'biased' toward clear sky conditions in the same way as the satellite data. Or are the CTM cloud fields in sufficient enough agreement with those seen by the satellites that the retrieval co-location sampling handles this?*

First of all, we would like to clarify our remark concerning "predominantly cloudless conditions" during the BB aerosol outflow event. According to the Level-3 AIRS retrievals (see Fig. C1 below), the cloud fraction was indeed typically less 0.5 over the region (40-80°E, 55-65°N) and period (17-27 July) most affected by the outflow of BB plumes from Siberian fires. Furthermore, the cloud fraction was smaller and clouds were less dense than typical for a summer period in the same region (cf. panels (a) and (c) with panels (b) and (d) in Fig. C1). That said, we recognize that our remark could be confusing, particularly because the cloud fractions inferred from different observations can differ significantly. The corresponding sentence is corrected in the revised manuscript.

(a)                                          (b)

[Figure]

(c)                                          (d)

**Figure C1.** Time-averaged maps of (a, b) the cloud fraction and (c, d) the cloud optical depth according to the data retrieved from the AIRS and MODIS observations, respectively, for the periods (a, c) 17 - 27 July 2016 and (b, d) 15 June – 14 July 2016. The data are shown only for the study region. Source: Giovanni, https://giovanni.gsfc.nasa.gov/giovanni/, last access: 1 November 2020.

Our satellite/model harmonization procedure did not involve any filtering of the model data with respect to the cloud coverage. However, since each available retrieval falling into a given grid cell was matched with the corresponding simulations in time and since grid cells/hours corresponding to the continuous cloud coverage are barely represented in the satellite observations, both the satellite data and simulations are "automatically" weighed towards the clear sky conditions. Furthermore, as part of a preliminary analysis, we examined whether or not the lack of satellite retrievals for cloudy scenes could introduce significant systematic discrepancies between the simulated and modeled data fields by plotting the relationship between the difference of the observed and retrieved values of AOD in a given grid cell in the study region and period and the number of retrievals per grid cell. This analysis is illustrated in Fig. C2 provided below. If there were strong systematic discrepancies between the observations and simulations due to the lack of observations for cloudy scenes, then the difference of the observed and retrieved values would probably depend on the number of retrievals per grid cell (since this number is inversely proportional to the fraction of cloudy scenes). However, we did not observe such a dependence and therefore concluded that biases introduced into our analysis by cloudy scenes in the satellite observations are not significant. We included a concise discussion of this point in Sect. 2.5 of the revised manuscript.

[Figure]

**Figure C2.** The differences between the simulated AOD values (for the 'bb_vbs' scenario) and their counterparts retrieved from the MODIS observations as a function of the number of AOD retrievals per grid cell. Both the simulated and retrieved data were gridded on a 1°×1° model grid with the hourly temporal resolution and were not subject to any pre-selection procedure. The red line shows the best linear fit to the data. Only each 10[th] data point is shown in the plot to preserve its readability. Note that the simulations shown here do not include the background part, which is the reason for a small negative bias in the simulated values.

Referee's comment: *P1L12: In the abstract, consider a basic description of importance of how the VBS scheme improved agreement between the model and satellite data (i.e. in Fig 4 b,d). This point is worth mentioning.*

We thank the referee for this useful suggestion. The corresponding remark is included in the abstract of the revised manuscript.

Referee's comment: *P21L6: In Figure 2b and 2d, I would suggest that the AOD scale range from, say, 0-2, to get a better sense of the AOD enhancement over the receptor region centered on Moscow.*

We have re-drawn Fig. 2 using the scale suggested by the referee. The AOD enhancement associated with anthropogenic emissions from Moscow (if this is what the referee meant) is now clearly seen in the simulations (Fig. 2d), but

the corresponding enhancement in the observations is more smeared (Fig. 2c). These spatial differences between the observed and simulated AOD values are typically less than 0.1 and are likely due to either a minor positive bias in the MODIS retrievals or underestimation of the AOD associated with biogenic aerosol by the model. These minor differences could not significantly affect the results of our analysis. In addition, we would like to note that the receptor region is somewhat shifted from the center of the plume northward in order to exclude the impact of minor local fires (in the south-western part of the domain, see Fig. 1) on the analysis results.

Referee's comment: *P21L9: by correlation, do you mean spatial pattern correlation or something else?*

We meant the correlation between the daily values of the spatially averaged retrievals and simulations. This is clarified in the revised manuscript.

Referee's comment: *Minor editorial point throughout: use the word 'rather' fewer times.*

We thank the referee for this useful suggestion. We tried to avoid using the word 'rather' in the revised manuscript unless it is necessary.

[revised manuscript text omitted]

Here we present the simulations performed with a microphysical dynamic (box) model of organic aerosol. The model is described in detail in Konovalov et al. (2019), where five different VBS schemes were used to simulate the BB OA evolution in an isolated plume under the prescribed ambient conditions. In the present study, box model simulations were performed using our simplified adaptive VBS scheme as well as the original 1.5-dimensional (C17) one (Ciarelli et al., 2017) from which our scheme was derived. Air temperature and the initial BB OA mass concentration were chosen to be representative of the conditions of the four "high-temperature" chamber experiments reported by Ciarelli et al. (2017). These experiments were conducted at a temperature of 288.15 K, with the initial BB OA mass concentration varying in the narrow range from 17.55 to 22.63 $\mu g \ m^{-3}$. In our model runs, we set the temperature accordingly at 288.15 K and assumed the fixed value of 19 $\mu g \ m^{-3}$ for the initial BB OA mass concentration. Note that the "low-temperature" experiments conducted by Ciarelli et al. (2017) at 263.15 K are not considered here because according to our simulations (see Sect. S4 below), the typical ambient temperature at the locations of the center of mass of the analyzed BB plumes was significantly higher than 263.15 K (ranging from about 280 to 285 K). Dilution and the background OA concentration, which were taken into account in the analysis reported in Konovalov et al. (2019), were disregarded in the simulations presented here, which are intended to replicate the conditions of chamber experiments. The parameters of our simplified VBS scheme (the reaction rates, enthalpies of evaporation, stoichiometric coefficients, and the ratio of the initial mass concentration of NTVOCs to the sum of the initial mass concentrations of the POA species) were chosen to be the same as those reported in Sect. 2.4 and Tables 1 and 2. As explained in Sects. 2.4 and 2.6, a few parameters (specifically those defining HV-SOA and LV-SOA yields, the product yields from the oxidation reactions of MV-SOA and LV-SOA, and the emission ratio of mass concentration of NTVOCs to the sum of mass concentrations of the POA species) were optimized in this study using satellite observations. The parameters of the C17 scheme were adopted from Ciarelli et al. (2017, see Table 1 therein). Note that these parameter values were partly constrained by data from both the low-temperature and high-temperature experiments (see Ciarelli et al., 2017 for details). The model runs were performed for a period of 110 hours. This period corresponds to the estimated maximum duration of the exposure of smoke plumes to sunlight in the analyzed region and period (see Sect 3.2). The OH concentration was fixed at $5 \times 10^6$ molec $cm^{-3}$.

The results of our simulations are presented in Fig. S1. Specifically, the figure shows the time series of the total BB OA mass concentration and also illustrates changes in the fractional composition of BB OA. The calculations can be compared to measurements of the BB OA mass concentrations at the high-temperature experiments. The BB OA concentration initially enhances more rapidly in the simulation with our scheme (Fig. S1a) than in the simulation with the original scheme (Fig. S1b). As a result, the BB OA concentration predicted by our scheme after about 10 hours of evolution is about 40 % larger than the corresponding concentration predicted by the original C17 scheme. It is also about 15% larger than the maximum concentration obtained in the chamber experiments. However, it should be noted that these particular experiments are not representative of the wide range of burning conditions and large variety of fuels in the Siberian forest. Moreover, there was significant variability in BB OA concentrations even under the controlled conditions of the chamber experiments (Fig. S1). Taking these observations into account, we consider the comparison presented in Fig. S1a as additional evidence that the simulations of the BB OA evolutions with our simplified scheme are sufficiently reasonable. A comparison of the results obtained with the simplified and original schemes (cf. Fig. S1a and Fig. S1b) reveals that the simulations are qualitatively different. Indeed, while the original scheme demonstrates a monotonic saturating increase of BB OA concentration, the simplified scheme yields a non-monotonic behavior of BB OA concentration (a rapid increase followed by a gradual decrease due to fragmentation of SOA). These two types of behavior of BB OA were earlier identified and discussed in Konovalov et al. (2019). However, the available chamber experiments, which are representative of only relatively small OH exposures, do not allow us to exclude any of these types of behavior of BB OA aerosol.

Since the definitions of surrogate organic species in our VBS scheme and C17 schemes are considerably different, a direct comparison of the simulated compositions of OA is not feasible. Nonetheless, the comparison of Fig. S1a and S1b reveals some obvious similarities between the simulations with the two schemes. In particular, there is a decreasing trend of POA in

the both simulations; there is also a steep increase in the concentration of the secondary compounds in the medium volatility range during the first several hours and eventual disappearance of these compounds. In both simulations, the low –volatility components ($C^* < 10^0 \, \mu g \, m^{-3}$) provide predominant contributions to the BB OA composition at the end of the evolution period. Overall, this comparison indicates that the BB OA behavior predicted by our scheme is not strikingly different from that predicted by the C17 scheme and is physically reasonable. The major differences are associated with the dynamics of the medium-volatility components.

[revised manuscript text omitted]

The 'bb_poa' simulation test scenario that was introduced in Sect. 2.3 and discussed in Sect. 3.3 addresses a hypothetical situation where the organic components of BB aerosol are not affected by any oxidation reactions. For this scenario, we also postulated that the POA species are hygroscopic instead of being hydrophobic as in the bb_vbs scenario to compensate for the water uptake by the SOA species. The simulations were performed using the same model configuration as for the 'bb_vbs' scenario

except that all the reaction rates reported in Table 2 were set to zero, the hygroscopicity parameter $\kappa_{org}$ for both LV-POA and MV-POA was assumed to be 0.2, and the POA emissions were increased by a factor of 4.3. Using the simulated concentrations for this scenario, we also performed an additional calculation with the OPTSIM module for a test case in which the contribution of water to the BB aerosol composition was neglected. The main simulation results are presented in Fig. S8.

According to these results, EnRs for both AAOD and AOD (see Fig. S8a, b) demonstrate monotonous decreasing dependencies, in striking contrast to the non-monotonous dependencies obtained from satellite observations (cf. with Fig. 5a, b) and simulations for the 'bb_vbs' scenario (cf. with Fig. 7a, b). The decrease in $EnR_{abs}$ is substantial (~50 %), while that in $EnR_{ext}$ is small (~10 %). Consistently with the monotonous changes in AOD and AAOD, SSA gradually increases (Fig. S8c), instead of demonstrating a hyperbolic (saturable) dependence on the BB aerosol photochemical age as in our analysis of the satellite data (see Figs. 5c). The corresponding values of the mass absorption and scattering efficiencies ($\alpha_a$ and $\alpha_s$, respectively) indicate (see Fig S8d) that the decrease in $\alpha_a$ (which is expected to occur due to the limited lifetime of BrC in the primary aerosol, see Sect. 2.4) is the main factor responsible for the strong decrease in EnR for AAOD. The main process underlying the decrease in EnR for AOD is apparently the evaporation of MV-POA, since in contrast to $\alpha_a$, $\alpha_s$ increases as a result of the BB evolution (by 44 %). The increase in $\alpha_s$ can be due to changes in the particle size distribution upon the evaporation of MV-POA, as well as due to an increase in the water uptake as a result of the rise in RH (see Fig. S6a). The test OPTSIM calculation, in which the contribution of water to BB aerosol was disregarded and $\alpha_s$ is found to increase by 27 % (see a brown line in Fig. S8d), confirms that the increase in the water uptake is important but not the major factor underlying changes in $\alpha_s$. It may be noteworthy that the initial – for the period considered – value of $\alpha_s$ in the 'bb_poa' scenario (6.4 m$^2$ g$^{-1}$) is found to be larger than that in the 'bb_vbs' scenario (4.0 m$^2$ g$^{-1}$, see Fig. 9). A more detailed examination of the simulation data reveals that this difference is associated with the differences in the corresponding particle volume size distributions. Specifically, the 'bb_vbs' simulation allocates a much larger fraction of the particle mass to the particles with diameters around 100 nm or less, which have a relatively small scattering efficiency, than the 'bb_poa' simulation. In the 'bb_vbs' simulation, these particles consist predominantly of the SOA species and are, therefore, formed as a result of oxidation of NTVOC and POA. Further SOA formation results in the growth of these particles, pushing the size distribution toward larger values as illustrated in Fig. S7. The difference between values of the absorption efficiency for the 'bb_vbs' and 'bb_poa' scenarios is determined by the difference in the absorptive properties of the POA and SOA species and is not of interest in the context of this study.

Overall, the simulation results presented in Fig. S8 indicate that the evaporation of POA and the water uptake by BB aerosol particles cannot explain a prominent increase in EnR for AOD, which is found both in the analysis of satellite data and in our simulations for the base case ('bb_vbs') scenario. Accordingly, the simulation for the 'bb_poa' scenario confirms that the strong increase in AOD is primarily a result of oxidation processes leading to the SOA formation. The analysis presented here may also have a wider implication when considered together with the OC emission estimates discussed in Sect. 2.6. Specifically, it provides further evidence that if a given model strongly underestimates AOD (with respect to satellite observations), this fact does necessarily mean that the underestimation is due to a negative bias in BB emission inventory data. Instead, it may be due to insufficiently strong SOA formation in the model. The proposed joint analysis of satellite and model data allows distinguishing between the two possible reasons for the underestimation of AOD by the model. If AOD is underestimated as a result of missing SOA formation processes, the adjustment of the POA emissions in the model would not enable it to reproduce the major features of the retrieved evolution of the BB aerosol optical properties, as demonstrated in our analysis presented in this section.

[revised manuscript text omitted]

**Supplementary figures**

[Figure]

**Figure S1.** The dynamics of the mass concentration of BB OA and several groups of its components according to the simulations performed with a box model using (a) the simplified VBS parameterization introduced in this paper (see Sect. 2.4) and (b) 1.5-dimensional VBS scheme proposed by Ciarelli et al. (2017). The crosses, triangles, and dots depict the BB OA mass concentration measurements (Ciarelli et al., 2017) originally reported for the OH exposures of 0, $30\times10^6$, and $50\times10^6$ molec cm$^{-3}$ h, respectively, and corresponding to 0, 5, and 10 hours of the exposure of BB aerosol to oxidation processes under the assumption that OH concentration equals to $5\times10^6$ molec cm$^{-3}$.

**(a)**

[Figure]

**(b)**

[Figure]

**(c)**

[Figure]

**(d)**

[revised manuscript text omitted]